

# Quantum hypothesis testing in many-body systems

**Jan de Boer[1⋆], Victor Godet[1†], Jani Kastikainen[2,4‡] and Esko Keski-Vakkuri[2,3‘∘]**

**1** Institute for Theoretical Physics, University of Amsterdam,
PO Box 94485, 1090 GL Amsterdam, The Netherlands
**2** Department of Physics, P.O.Box 64, FIN-00014 University of Helsinki, Finland
**3** Helsinki Institute of Physics, P.O.Box 64, FIN-00014 University of Helsinki, Finland
**4** APC, AstroParticule et Cosmologie, Université de Paris, CNRS/IN2P3, CEA/IRFU,
Observatoire de Paris, 10, rue Alice Domon et Léonie Duquet, 75205 Paris Cedex 13, France

⋆ j.deboer@uva.nl, † v.z.godet@uva.nl,
‡ jani.kastikainen@helsinki.fi, ∘ esko.keski-vakkuri@helsinki.fi

## Abstract

One of the key tasks in physics is to perform measurements in order to determine the state of a system. Often, measurements are aimed at determining the values of physical parameters, but one can also ask simpler questions, such as "is the system in state A or state B?". In quantum mechanics, the latter type of measurements can be studied and optimized using the framework of quantum hypothesis testing. In many cases one can explicitly find the optimal measurement in the limit where one has simultaneous access to a large number $n$ of identical copies of the system, and estimate the expected error as $n$ becomes large. Interestingly, error estimates turn out to involve various quantum information theoretic quantities such as relative entropy, thereby giving these quantities operational meaning. In this paper we consider the application of quantum hypothesis testing to quantum many-body systems and quantum field theory. We review some of the necessary background material, and study in some detail the situation where the two states one wants to distinguish are parametrically close. The relevant error estimates involve quantities such as the variance of relative entropy, for which we prove a new inequality. We explore the optimal measurement strategy for spin chains and two-dimensional conformal field theory, focusing on the task of distinguishing reduced density matrices of subsystems. The optimal strategy turns out to be somewhat cumbersome to implement in practice, and we discuss a possible alternative strategy and the corresponding errors.

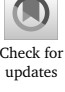

# 1   Introduction

The purpose of this work is to i) introduce and review quantum hypothesis testing for readers with a background in quantum field theory and many-body theory, ii) develop some new results in a perturbative setup, and then iii) apply the tools to distinguish in particular two reduced density matrices in a subsystem of a quantum many-body system.

We begin with some background motivation. An elementary quantum task is to distin-

guish between two quantum states. Recently there has been much effort to study this question in quantum field theory and many-body theory, and to develop methods to compute various quantum information theoretic distinguishing measures analytically. A particularly interesting case is a large or infinite system in two different global states viewed from a small subsystem. The problem is then to distinguish the two reduced density matrices (RDMs) resulting from a partial trace over the complement of the subsystem. For this problem, critical systems modeled by conformal field theories have offered a fruitful arena for analytic progress. Additional motivation for studying conformal field theories comes from the connections between quantum information and gravity. In this context, a famous issue is the state of Hawking radiation escaping from an evaporating black hole: how can one detect in subsystems the subtle quantum correlations between radiated quanta at different times, to distinguish a conjectured pure state of radiation from something resembling thermal radiation?

In quantum field theory and many-body theory, there has been much progress in studying well-known distinguishing measures both analytically and numerically. For example, in the context of conformal field theory and critical lattice models, there are studies of fidelity $F(\rho, \sigma)$ [1, 2], relative entropy $S(\rho \| \sigma)$ [2–7], generalized divergences [8–13] and trace distance $D(\rho, \sigma) = \frac{1}{2} \| \rho - \sigma \|$ [14, 15]. In this work, our focus is instead to distinguish two states by measurements. We begin with three remarks: i) a rigorous framework for the task is *quantum hypothesis testing*, ii) many results obtained for relative entropy and generalized divergences can be embedded in this framework, giving them an operational interpretation, and iii) hypothesis testing also suggests an optimal measurement protocol to minimize the error in distinguishing two states. We are thus lead to study how quantum hypothesis testing can be implemented in many-body theory and quantum field theory.

Quantum hypothesis testing builds on the classical theory of hypothesis testing, which is a cornerstone of statistical analysis and the scientific method. Borrowing terminology from the classical theory, one may want to test whether the system is in a state $\rho$ called the *null hypothesis*, thought of as the "background", or in another quantum state $\sigma$ called the *alternative hypothesis*, which is the "signal" that one desires to detect. The framework of quantum hypothesis testing then provides rigorous estimates for the probabilities of the errors of mistaking the two states in an asymptotic limit of many measurements[1]. Here, it is important that by "many measurements" we mean simultaneous measurements on many copies of the system, as opposed to performing a sequence of individual measurements on independent single copies of the system. The error probability estimates involve various quantum information theoretic quantities, which depend on the details of the quantum hypothesis testing protocol. For example, for the case of so-called asymmetric testing, the error estimate involves the relative entropy as well as the relative entropy variance between the two states; both measures can be obtained from generalized divergences. Quantum hypothesis testing has numerous applications in quantum information science, such as quantum illumination [18–20], entanglement-assisted communication [21], and the analysis of environment-parametrized quantum channels [22, 23], to name a few. In particular, there are rigorous studies of particular quantum hypothesis testing protocols to distinguish states in spin chains, see *e.g.* [17, 24, 25].

Here, we are interested in connecting various mathematical results about hypothesis testing to implementations and applications of hypothesis testing in models at criticality with an emphasis on distinguishing reduced density matrices of subsystems associated to different global states. For example subsystems of free fermion chains have been extensively studied in the context of entanglement, because subsystem reduced density matrices are determined analytically by two-point functions [26–29]. The analytic tractability allows one to study for example entanglement spectra [30, 31] and entanglement entropies of subsystems [32] (see

---

[1]The asymptotic limit is an idealization, in practice one is limited to a finite number of samples. We leave this "finite blocklength" case [16, 17] to a further investigation.

also [33, 34] for reviews). Distance measures such as relative entropy and Rényi divergences have also been explored [35, 36].

We now summarize the main results of this work, which is divided in two parts. In the first part of this paper, we consider quantum hypothesis testing for general systems and develop a perturbative approach to hypothesis testing. Many applications often involve a setup where the two global states are parametrically close, as functions of one parameter (such as the ambient temperature). In that case it is natural to use a perturbative expansion to approximate two neighboring states. After giving a general review of quantum hypothesis testing in section 2, we study error probability estimates combined with a perturbative approach in section 3. The relevant error estimates involve the perturbative expansions of relative entropy and relative entropy variance, with leading terms appearing at second order. To examine the behavior of the error estimate, we study the relative size of these leading terms. In doing so, we find a universal result, a lower bound for the ratio of the two terms, applicable for any system in the perturbative setting. The result also allows us to develop a new joint perturbative bound on the two types of errors.

In section 4, we discuss and compare different types of measurements. We argue that independent (*i.e.* factorized) measurements perform poorly in general. We review the optimal measurement described in [37], which saturates the theoretical error bound. This measurement turns out to be rather difficult to describe explicitly. As an alternative, we consider a simpler but suboptimal measurement, the likelihood ratio (or Neyman-Pearson) test, which is easier to describe and performs rather well.

In the second part of this work, we implement these measurement protocols in quantum systems of increasing complexity: a single qubit, Gaussian fermion chains and finally two-dimensional conformal field theories.

We consider the qubit in section 5 and we construct the optimal measurement. Surprisingly, an explicit description is difficult as it leads to a challenging combinatorial problem, involving Krawtchouk polynomials and related to the Terwilliger algebra of the Hamming cube. This motivates the simpler likelihood ratio test, which can be described explicitly, and implemented with a quantum circuit given in Figure 5. Using numerical methods, we study the optimal measurement and compare it to the likelihood ratio test.

In section 6, we move on to spinless fermion chains with quadratic Hamiltonians. Motivated by hypothesis testing, we derive formulas for the relative entropy and the relative entropy variance in subsystems of free fermions (with only hopping interactions) at different temperatures. Then we present a prescription to compute overlaps between eigenstates of two different modular Hamiltonians of the same subsystem. The main technical tool is a generalization of Wick's theorem to correlators that involve Bogoliubov transformations [38, 39]. The resulting overlaps allow the construction of the optimal measurement that distinguishes two thermal states by a local measurement. We find that in the simplest single fermion subsystem, the likelihood ratio test is optimal for distinguishing any two reduced density matrices, whereas for a two-fermion subsystem, it is not sufficient in general. In the XY model at finite temperature, for a two-fermion subsystem, the likelihood ratio test is again optimal.

We finally consider two-dimensional CFTs in section 7. We focus on states for which the modular Hamiltonian can be written as an integral of the stress tensor [40]. We construct optimal measurement protocols for subregions, using techniques of boundary CFT [41] to compute the necessary ingredients. This general framework can be applied to distinguish two thermal states from a subregion, and we study explicitly the case of the free fermion. We explain how to implement the optimal measurement, which is difficult to describe explicitly, and the simpler likelihood ratio test. We also consider the detection of a primary excitation on top of the vacuum, for which the likelihood ratio test can be implemented with a relatively simple procedure: by measuring one-point functions of the lightest operator interacting with

the primary excitation.

We conclude with a discussion and some open questions, and summarize various useful properties and technical results in the appendices.

After the completion of this paper, related work studying various properties and applications of relative entropy variance (there called "variance of relative surprisal") from an information theoretic point of view appeared in [43].

## 2 Review of quantum hypothesis testing

In this section, we give a brief review of quantum hypothesis testing, to provide background for readers unfamiliar with this theory. In (binary) hypothesis testing, we have to choose between two hypotheses, the *null hypothesis $H_0$* and the *alternative hypothesis $H_1$*.

In the classical theory, the two hypotheses are associated with two probability distributions $p(X), q(X)$ over the space $\Omega$, and the problem is to discriminate between the two by a test $T : \Omega \to I$. If $I = [0, 1]$, the test is randomized, if $I = \{0, 1\}$, the test is deterministic. The probability of detection for the hypothesis $H_1$ is then the expectation value $\mathbf{E}_q[T] = \sum_{x \in \Omega} Q(x) T(x)$. If the test is deterministic, it is often expressed as an indicator function $T = \mathbf{1}_H = \mathbf{1}\{x \in H\}$ over an acceptance subset $H \subset \Omega$.

In the quantum theory, $H_0$ and $H_1$ are two quantum states $\rho$ and $\sigma$, and the test becomes an operator $T = E_1$. More precisely the decision is made by measuring observables $E_0 = A$ and $E_1 = 1 - A$ which form a positive operator-valued measure (POVM), *i.e.* $0 \le E_i \le 1$ and $\sum_{i=0,1} E_i = 1$. In making a measurement, the probabilities of identifying the two states correctly are $\mathrm{Tr}(\rho E_0)$ and $\mathrm{Tr}(\sigma E_1)$, the latter being the probability of detection of the hypothesis $H_1$. There are two ways to make errors, which are called of type I or type II. Type I error (false positive) corresponds to identifying $H_1$ while in fact $H_0$ is true. Type II error (false negative, missed detection) corresponds of choosing $H_0$ while $H_1$ is true. The probabilities of the two errors are given by

$$\alpha = \mathrm{Tr}\,\rho(1 - A) \qquad \text{(type I)}, \qquad\qquad (2.1)$$
$$\beta = \mathrm{Tr}\,\sigma A \qquad \text{(type II)}.$$

The objective of hypothesis testing is to find the best measurement which jointly minimizes the two errors. In this work we focus on the independent and identically distributed (i.i.d.) setting, and consider a joint measurement $A^{(n)}$ on $n$ identical copies of the system, to discriminate between the states $\rho^{\otimes n}$ and $\sigma^{\otimes n}$. The error probabilities then become $n$-dependent, $\alpha_n$ and $\beta_n$, given by

$$\alpha_n = \mathrm{Tr}\,\rho^{\otimes n}(1 - A^{(n)}) \qquad \text{(type I)}, \qquad\qquad (2.2)$$
$$\beta_n = \mathrm{Tr}\,\sigma^{\otimes n} A^{(n)} \qquad \text{(type II)}.$$

Quantum hypothesis testing addresses the question of the optimality of a measurement $A^{(n)}$. The notion of optimality depends on the error optimization strategy. Symmetric testing optimizes the sum of the two errors, while asymmetric testing optimizes the type II error under the condition that the type I error remains bounded.[2] We review these two cases below.

---

[2] A third strategy assumes a given exponential decay rate for the type I error.

## 2.1 Symmetric testing

In symmetric hypothesis testing, we treat the two types of errors equally and define the symmetric error[3]

$$P_n = \frac{1}{2}(\alpha_n + \beta_n) \,. \tag{2.3}$$

The optimal measurement is obtained by minimizing $P_n$ over all possible measurements $A^{(n)}$, where $A^{(n)}$ is a Hermitian operator satisfying $0 \leq A^{(n)} \leq 1$. We can define the minimum error as

$$P_n^* = \frac{1}{2} \inf_{A^{(n)}} \text{Tr}\left(\rho^{\otimes n}(1 - A^{(n)}) + \sigma^{\otimes n} A^{(n)}\right) \,. \tag{2.4}$$

The asymptotic behavior of this quantity is given by the quantum Chernoff bound [42], which says that

$$\lim_{n \to +\infty} \left(-\frac{1}{n} \log P_n^*\right) = -\log Q(\rho, \sigma) \,, \tag{2.5}$$

where the quantum Chernoff distance is defined as

$$-\log Q(\rho, \sigma) \equiv \max_{0 \leq s \leq 1} \left[-\log Q_s(\rho, \sigma)\right] \,, \qquad Q_s(\rho, \sigma) = \text{Tr}\,\rho^s \sigma^{1-s} \,. \tag{2.6}$$

We can see that $-\log Q_s(\rho, \sigma)$ are proportional to the relative Rényi entropies defined by Petz [44]. As a result, symmetric hypothesis testing gives an operational meaning to these quantities. More precisely, their maximum for $0 \leq s \leq 1$ gives the asymptotic exponent of the symmetric error

$$P_n^* \underset{n \to \infty}{\sim} e^{-n(-\log Q)} \,. \tag{2.7}$$

It is also interesting that $Q(\rho, \sigma)$ is related to other information quantities [42]. We have

$$0 \leq 1 - Q \leq \mathcal{T} \leq \sqrt{1 - Q^2} \,, \tag{2.8}$$

where $\mathcal{T} = \frac{1}{2}\|\rho - \sigma\|_1$ is the trace norm distance and

$$Q \leq Q_{s=1/2} = \text{Tr}\,\rho^{1/2}\sigma^{1/2} \leq F(\rho, \sigma) \,, \tag{2.9}$$

where $F(\rho, \sigma) = \|\rho^{1/2}\sigma^{1/2}\|_1$ is the Uhlmann fidelity. If one of the states is pure, we have $Q = \text{Tr}\,\rho\,\sigma$. $Q$ also satisfies the data-processing inequality (B.17).

## 2.2 Asymmetric testing

In this work, we will be interested in the asymmetric treatment of the two types of errors, which is the setting which gives an operational meaning to the relative entropy. In asymmetric testing, we require that the type I error is bounded, $\alpha_n \leq \varepsilon$, and examine the asymptotic behavior of the type II error $\beta_n$[4]. More precisely, we estimate the asymptotic behavior of the quantity

$$\beta_n^*(\varepsilon) \equiv \inf_{A^{(n)}} \{\beta_n \mid \alpha_n \leq \varepsilon\} \,, \tag{2.10}$$

where the infimum is taken over Hermitian operators $A^{(n)}$ satisfying $0 \leq A^{(n)} \leq 1$.

The asymptotic behavior of this quantity is given by the *quantum Stein's lemma* [45, 46] which is the statement

$$\lim_{n \to \infty} \left(-\frac{1}{n} \log \beta_n^*(\varepsilon)\right) = S(\rho\|\sigma) \,, \tag{2.11}$$

---

[3]It is also possible to consider a more general combination of the form $P_n = \kappa\alpha_n + (1-\kappa)\beta_n$ and $0 < \kappa < 1$, with no change to the discussion [42].

[4]The asymmetric case means that the probability of missed detection (type II error) is seen as more significant than a false positive (type I error).

for any $0 < \varepsilon < 1$. The relative entropy $S(\rho \| \sigma)$ is defined as

$$S(\rho \| \sigma) = \begin{cases} \mathrm{Tr}\left[\rho(\log \rho - \log \sigma)\right], & \mathrm{supp}(\rho) \subseteq \mathrm{supp}(\sigma) \\ +\infty & \mathrm{otherwise} \end{cases} . \tag{2.12}$$

The quantum Stein's lemma shows that the type II error decays exponentially at large $n$ with exponent given by the relative entropy,

$$\beta_n^*(\varepsilon) \underset{n \to \infty}{\sim} e^{-nS(\rho \| \sigma)} . \tag{2.13}$$

The asymptotic formula (2.11) was improved in [37, 47] to subleading order.[5] The refined quantum Stein's lemma says that

$$-\frac{1}{n} \log \beta_n^*(\varepsilon) = S(\rho \| \sigma) + \frac{1}{\sqrt{n}} \sqrt{V(\rho \| \sigma)} \Phi^{-1}(\varepsilon) + \mathcal{O}\left(\frac{\log n}{n}\right) \tag{2.14}$$

and involves the *relative entropy variance*[6] defined as

$$V(\rho \| \sigma) \equiv \mathrm{Tr}\left[\rho(\log \rho - \log \sigma)^2\right] - S(\rho \| \sigma)^2 , \tag{2.15}$$

and the inverse $\Phi^{-1}$ of the cumulative distribution function of the normal distribution,

$$\Phi(x) \equiv \frac{1}{\sqrt{2\pi}} \int_{-\infty}^{x} dt \, e^{-t^2/2} . \tag{2.16}$$

In analogy with the quantum Chernoff distance, one can also define [48] the *quantum hypothesis testing relative entropy*

$$D_H^\varepsilon(\rho \| \sigma) \equiv -\log \beta_n^*(\varepsilon) , \tag{2.17}$$

for $0 < \varepsilon < 1$. This quantity is another generalized divergence, satisfying the data-processing inequality [47]. In the rest of this work we will be focusing on asymmetric testing and the refinement of the quantum Stein's lemma (2.14).

The refined quantum Stein's lemma should be understood as a refined estimate of the asymptotic error of an optimal measurement. Following [37], it is useful to define the quantity

$$\alpha_n(E_1, E_2) = \inf_{A^{(n)}} \left\{ \alpha_n \mid \beta_n \leq \exp(-(E_1 n + E_2 \sqrt{n} + o(\sqrt{n}))) \right\} . \tag{2.18}$$

This is the best type I error if we require that the type II error exponentially decays with leading exponent $E_1$ and subleading exponent $E_2$. It is similar to $\beta_n^*(\varepsilon)$ in that it measures the interdependence between the type II and type I errors. It is shown in [37] that an equivalent way to formulate the refined quantum Stein's lemma is to say that

$$\lim_{n \to +\infty} \alpha_n(E_1, E_2) = \begin{cases} 0 & \text{if } E_1 < S(\rho \| \sigma) \\ \Phi\left(\dfrac{E_2}{\sqrt{V(\rho \| \sigma)}}\right) & \text{if } E_1 = S(\rho \| \sigma) \\ 1 & \text{if } E_1 > S(\rho \| \sigma) \end{cases} . \tag{2.19}$$

We see that the relative entropy $S(\rho \| \sigma)$ acts as a threshold value for the leading exponent $E_1$. Above the threshold, the type I error becomes uncontrolled and goes to one, while below the

---

[5]See also [25] for a generalization to beyond i.i.d. setting and additional discussion.

[6]The nomenclature varies, other names are "quantum relative variance", "quantum information variance", *etc,*

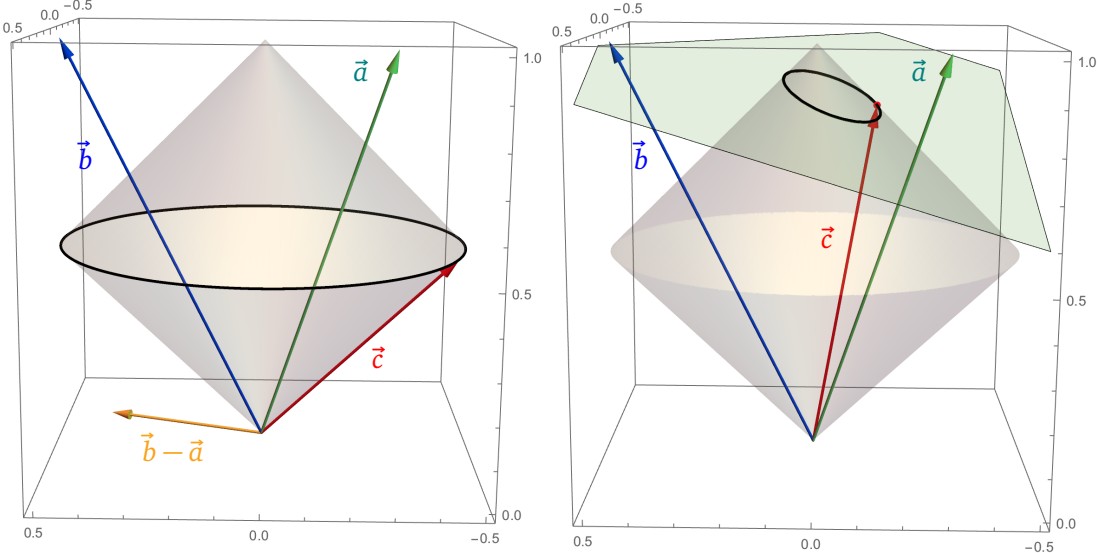

Figure 1: Geometrical problem for the one-shot optimal measurement of a qubit. We optimize over a vector $\vec{c}$ in $\mathbb{R}^4$ and plot here the coordinates $(c_1, c_2, c_4)$ (suppressing $c_3$). The condition $0 \leq A \leq 1$ restricts $\vec{c}$ to lie in the gray diamond. **Left:** Symmetric testing. This corresponds to minimizing the product $(\vec{b} - \vec{a}) \cdot \vec{c}$. The optimal vector $\vec{c}$ is the point on the black circle that is most opposite to $\vec{b} - \vec{a}$. **Right:** Asymmetric testing. This corresponds to minimizing $\beta = \vec{b} \cdot \vec{c}$ under the condition $\alpha = 1 - \vec{a} \cdot \vec{c} \leq \varepsilon$, which restricts $\vec{c}$ to be above the green plane. The intersection of this plane and the boundary of the diamond and is the black circle, on which the optimal $\vec{c}$ must lie. In both cases, we show the optimal solution in red. The values chosen for these plots are $\vec{a} = (-0.3, 0.3, 0, 1), \vec{b} = (0.5, 0, 0, 1)$ and $\varepsilon = 0.1$.

threshold, it can be made to vanish. The refined asymptotics become relevant when we are exactly on the threshold. On the threshold, we define

$$\alpha_n^*(E_2) = \alpha_n(S(\rho \| \sigma), E_2) , \tag{2.20}$$

and we have

$$\lim_{n \to +\infty} \alpha_n^*(E_2) = \Phi\left(\frac{E_2}{\sqrt{V(\rho \| \sigma)}}\right) , \tag{2.21}$$

which varies smoothly from 0 to 1 when $E_2$ ranges from $-\infty$ to $+\infty$.

## 2.3 Single qubit example

We now consider a toy version of our problem: what would be the optimal measurement for a single qubit? This example gives a nice illustration of quantum hypothesis testing. Here, we only take a single copy of the system: we describe the "one-shot" measurement. As we will see, it can be formulated as a constrained optimization problem which has a simple geometrical interpretation.

We have a qubit in the two possible states $\rho$ and $\sigma$ and we would like to find the best Hermitian operator $A$ with $0 \leq A \leq 1$ to distinguish between these two states. In symmetric testing, we are minimizing the error $\frac{1}{2}(\alpha + \beta)$. In the asymmetric case, we are minimizing the type II error $\beta$ under the condition that the type I error $\alpha$ is less than a given $\varepsilon$.

This can be formulated geometrically using a parametrization in terms of Pauli matrices. Defining the four-vector of $2 \times 2$ matrices $\vec{\sigma} = (\sigma^1, \sigma^2, \sigma^3, 1)$, we write

$$\rho = \frac{1}{2}\vec{a} \cdot \vec{\sigma}\,, \qquad \sigma = \frac{1}{2}\vec{b} \cdot \vec{\sigma}, \qquad \vec{a}, \vec{b} \in \mathbb{R}^4\,, \tag{2.22}$$

in terms of two four-vectors $\vec{a}, \vec{b}$. From $\operatorname{Tr}\rho = \operatorname{Tr}\sigma = 1$, we have that $a_4 = b_4 = 1$. We parametrize the Hermitian operator $A$ using a four-vector $\vec{c}$ as

$$A = \vec{c} \cdot \vec{\sigma}, \qquad \vec{c} \in \mathbb{R}^4\,. \tag{2.23}$$

The type I and type II errors take the form

$$\begin{aligned} \alpha &= 1 - \vec{a} \cdot \vec{c}\,, \\ \beta &= \vec{b} \cdot \vec{c}\,. \end{aligned} \tag{2.24}$$

The condition $0 \leq A \leq 1$ gives $0 \leq c_4 \leq 1$ and

$$\sqrt{c_1^2 + c_2^2 + c_3^2} \leq \min(c_4, 1 - c_4)\,. \tag{2.25}$$

This defines a diamond in $\mathbb{R}^4$ depicted in gray in Figure 1. Then, we have two different optimization problems corresponding to symmetric or asymmetric testing.

**Symmetric testing.** This is depicted in the left of Figure 1. Here, we have to find the vector $\vec{c}$ that minimizes $(\vec{b} - \vec{a}) \cdot \vec{c}$ under the condition that $\vec{c}$ lies inside the gray diamond. We can see that the optimal $\vec{c}$ lies on the circle corresponding to $c_4 = \frac{1}{2}$ and $c_1^2 + c_2^2 + c_3^2 = \frac{1}{2}$ (depicted in black). We can write down the solution explicitly as

$$c_4 = \frac{1}{2}\,, \qquad c_i = \frac{1}{2|\vec{b} - \vec{a}|}(a_i - b_i), \qquad i = 1, 2, 3\,, \tag{2.26}$$

which is shown in red.

**Asymmetric testing.** This is depicted in the right of Figure 1. In this case, we have to find the vector $\vec{c}$ that minimizes $\beta = \vec{b} \cdot \vec{c}$ under two conditions: the requirement $0 \leq A \leq 1$ forces $\vec{c}$ to lie inside the gray diamond and the constraint $\alpha \leq \varepsilon$ implies that $\vec{c}$ must lie above the green plane. The optimal $\vec{c}$ is inside the intersection region where these two inequalities are saturated (shown in black) and is shown in red. It is also possible to write down explicit expressions for the optimal vector $\vec{c}$ by solving the quadratic equations that define it.

# 3 Perturbative hypothesis testing

In this section, we study quantum hypothesis testing in a pertubative regime. We consider the case where the alternative hypothesis and the null hypothesis states belong to a one-parameter family, and are perturbatively close. This setting is natural in many applications. We will derive a new joint bound on the type I and type II errors, and a universal lower bound on the ratio of the relative entropy variance to the relative entropy, for systems with a finite dimensional Hilbert space.

We are interested in a one-parameter family of states, with the two states related by the series expansion[7]

$$\rho = \sigma + \lambda \rho^{(1)} + \frac{\lambda^2}{2}\rho^{(2)} + \mathcal{O}(\lambda^3), \tag{3.1}$$

---

[7]It would be more natural to expand the hypothesis state $\sigma$ over the null hypothesis $\rho$, $\sigma = \rho - \lambda \rho^{(1)} + \cdots$, our convention is chosen to make it more convenient to use some previous results from the literature. The two conventions are related by a trivial relabeling.

where $\lambda$ is a small parameter. This setting is natural in many applications of hypothesis testing. For example, consider the analysis of environment-parametrized quantum channels [20], where a system is interacting with an environment whose state is dependent on a parameter with unknown value. As concrete examples, [20] studied thermal and amplifier channels, where the environment is a thermal state parametrized by the temperature. The problem then is to distinguish two channels with two nearby temperatures, differing by a small parameter $\lambda$.

Another motivation is to consider CFT reduced density matrices in subsystems in the limit where the subsystem size is perturbatively small. An example could be the eigenstate thermalization hypothesis, in which expectation values of reduced density matrices of high energy eigenstates appear close to thermal, and it is of interest to study how the system responds to changes in the ratio of the subsystem size to the global system size. Another setting is to study global thermal states reduced to a subsystem, and consider the dimensionless ratio of the subsystem size to the thermal wavelength as a parameter to vary. We study optimal measurements for such subsystems in section 7.

## 3.1 A perturbative bound on errors

The quantum Stein's lemma was derived by first proving a bound [45] and then showing that it can be achieved [46]. For the first part, the following bound was used:

$$(1 - \alpha_n)(-\log \beta_n) \leq nS(\rho \| \sigma) + \log 2 \,, \tag{3.2}$$

which holds for a general measurement $A^{(n)}$ and any $n$. This can be seen as a bound on how good a measurement can be. It characterizes the trade-off between the two types of errors: $\alpha_n$ and $\beta_n$ cannot be made arbitrarily small at the same time.

The bound (3.2) can be seen as a "first order in $n$" bound that holds for a general measurement. We will now derive a "second order in $n$" bound that holds for a restricted set of measurements that are optimal at first order in $n$. This consists of all the measurements with errors satisfying the two conditions

$$\alpha_n \leq \varepsilon, \qquad \beta_n \leq e^{-nS(\rho\|\sigma) - \sqrt{n}E_2} \,, \qquad n \to +\infty \,, \tag{3.3}$$

for some fixed choice of $\varepsilon$ and $E_2$. The refinement of the Stein's lemma implies that

$$\Phi\left(\frac{E_2}{\sqrt{V(\rho\|\sigma)}}\right) \leq \varepsilon \,, \tag{3.4}$$

with saturation for the optimal measurement. In the notation of section 2.2, we have $\alpha_n \geq \alpha_n^*(E_2)$ and $\beta_n \geq \beta_n^*(\varepsilon)$, which implies that

$$\log \alpha_n \log \beta_n \leq \log \alpha_n^*(E_2) \log \beta_n^*(\varepsilon) \,. \tag{3.5}$$

We can then use the asymptotic estimate

$$\log \alpha_n^*(E_2) \log \beta_n^*(\varepsilon) \underset{n \to +\infty}{\sim} -nS(\rho\|\sigma) \log\left[\Phi\left(\frac{E_2}{\sqrt{V(\rho\|\sigma)}}\right)\right] \,, \tag{3.6}$$

to obtain the bound

$$\log \alpha_n \log \beta_n \leq -nS(\rho\|\sigma) \log\left[\Phi\left(\frac{E_2}{\sqrt{V(\rho\|\sigma)}}\right)\right] \,, \qquad n \to +\infty \,. \tag{3.7}$$

This is a bound on the measurements satisfying (3.3) and can be interpreted as a second order in $n$ refinement of (3.2). It also characterizes the trade-off between the two types of errors, implying that we cannot make both $\alpha_n$ and $\beta_n$ too small. Note that this also gives a bound on the LHS of (3.2) since we have $(1 - \alpha_n)(-\log \beta_n) \leq \log \alpha_n \log \beta_n$. It becomes stronger than (3.2) for $E_2 \geq \Phi^{-1}(1/e)\sqrt{V(\rho\|\sigma)} \approx -0.34\sqrt{V(\rho\|\sigma)}$.

We now consider measurements satisfying (3.3) in the perturbative regime (3.1), taking $\varepsilon$ and $E_2$ to be independent of $\lambda$, and we consider the perturbative version of the upper bound (3.7). As will be shown in the next subsection, the leading terms of both the relative entropy and the relative entropy variance are quadratic in $\lambda$:

$$S(\rho\|\sigma) = \frac{\lambda^2}{2}S^{(2)}(\rho\|\sigma) + \mathcal{O}(\lambda^3), \qquad V(\rho\|\sigma) = \frac{\lambda^2}{2}V^{(2)}(\rho\|\sigma) + \mathcal{O}(\lambda^3) . \tag{3.8}$$

In the perturbative limit, we see that at leading order

$$\alpha_n^*(E_2) = \frac{\lambda}{2}\sqrt{\frac{V^{(2)}(\rho\|\sigma)}{\pi E_2^2}}\exp\left(-\frac{E_2^2}{\lambda^2 V^{(2)}(\rho\|\sigma)}\right) , \tag{3.9}$$

where we have restricted to $E_2 < 0$ for $\alpha_n^*(E_2)$ to be close to zero rather than close to one. Note that $\alpha_n^*(E_2)$ is non-perturbative in $\lambda$, which is a consequence of the fact that the variance becomes small in the perturbative $\lambda \to 0$ limit. Because the estimate for $\alpha_n$ is obtained using the central limit theorem, it has an error of order $n^{-1/2}$. As a result, we can trust the above result only in the regime where $n$ is non-perturbatively large:

$$n \gg e^{c/\lambda^2} , \tag{3.10}$$

where $c$ is some positive constant. We can now consider the perturbative $\lambda \to 0$ limit of (3.6) and we find

$$\log \alpha_n^*(E_2) \log \beta_n^*(\varepsilon) \underset{n \to +\infty}{\sim} nE_2^2 \frac{S^{(2)}(\rho\|\sigma)}{V^{(2)}(\rho\|\sigma)} + \mathcal{O}(\lambda) . \tag{3.11}$$

Interestingly, this gives a finite answer in the $\lambda \to 0$ limit. This implies the bound

$$\log \alpha_n \log \beta_n \leq nE_2^2 \frac{S^{(2)}(\rho\|\sigma)}{V^{(2)}(\rho\|\sigma)} + \mathcal{O}(\lambda) , \qquad n \to +\infty , \tag{3.12}$$

which holds on all measurements satisfying the conditions (3.3).

In the next subsection, we will obtain a general lower bound $V^{(2)}(\rho\|\sigma) \geq 2S^{(2)}(\rho\|\sigma)$ which is saturated when $\rho$ and $\sigma$ commute at first order in $\lambda$. This implies that the above bound becomes

$$\log \alpha_n \log \beta_n \leq \frac{nE_2^2}{2} . \tag{3.13}$$

It is interesting to note that this bound is universal in the sense that it is independent on the state. It is saturated for the optimal measurement if and only if $\rho$ and $\sigma$ commute at first order in $\lambda$.

## 3.2 Lower bound for the ratio

We will now prove a lower bound on the ratio $V(\rho\|\sigma)/S(\rho\|\sigma)$ in the perturbative regime (3.1). The relative entropy has the perturbative expansion

$$S(\rho\|\sigma) = \frac{\lambda^2}{2}S^{(2)}(\rho\|\sigma) + \mathcal{O}(\lambda^3) \tag{3.14}$$

with no linear term, because $S(\rho\|\sigma) \geq 0$ with saturation at $\lambda = 0$. The perturbative relative entropy $S^{(2)}(\rho\|\sigma)$ is given by [49][8]

$$S^{(2)}(\rho\|\sigma) = \mathrm{Tr}\left(\rho^{(1)}\mathcal{L}\right), \tag{3.15}$$

where $\mathcal{L}$ is the logarithmic derivative

$$\mathcal{L} = \frac{d}{d\lambda}\log\left(\sigma + \lambda\rho^{(1)}\right)\Big|_{\lambda=0} = \sigma^{-1}\left(\rho^{(1)} + \frac{1}{2}\left[\log\sigma,\rho^{(1)}\right] + \frac{1}{12}\left[\log\sigma,\left[\log\sigma,\rho^{(1)}\right]\right] + \dots\right). \tag{3.16}$$

Relative entropy variance has a similar expansion and the linear term vanishes again, since $V(\rho\|\sigma) \geq 0$ with saturation at $\lambda = 0$. Then,

$$V(\rho\|\sigma) = \frac{\lambda^2}{2}V^{(2)}(\rho\|\sigma) + \mathcal{O}(\lambda^3) \tag{3.17}$$

where the perturbative variance is given by[9]

$$V^{(2)}(\rho\|\sigma) = 2\,\mathrm{Tr}(\sigma\mathcal{L}^2). \tag{3.18}$$

Since perturbative relative entropy and variance have the same behaviours for small $\lambda$, their ratio is finite in the limit $\lambda \to 0$:

$$\lim_{\lambda\to 0}\frac{V(\rho\|\sigma)}{S(\rho\|\sigma)} = \frac{V^{(2)}(\rho\|\sigma)}{S^{(2)}(\rho\|\sigma)}. \tag{3.19}$$

Our main result is the following *universal lower bound* for this ratio:

**Theorem 1.** *Let $\rho(\lambda)$ be a one-parameter family of density matrices over a finite dimensional Hilbert space. Given the expansion $\rho = \sigma + \lambda\rho^{(1)} + \frac{\lambda^2}{2}\rho^{(2)} + \mathcal{O}(\lambda^3)$, the ratio obeys the lower bound*

$$\frac{V^{(2)}(\rho\|\sigma)}{S^{(2)}(\rho\|\sigma)} \geq 2, \tag{3.20}$$

*with an equality if and only if $\left[\sigma,\rho^{(1)}\right] = 0$.*

To prove the theorem, we need an expression for $\mathcal{L}$ in the eigenbasis of $\sigma$. Let the eigenvalues of $\sigma$ be $\lambda_i$. Then a generic function $f(\sigma + X)$ has the following expansion in the eigenbasis of $\sigma$:

$$f(\sigma + X)_{ij} = f(\lambda_i)\delta_{ij} + \frac{f(\lambda_i) - f(\lambda_j)}{\lambda_i - \lambda_j}X_{ij} + \mathcal{O}(X^2). \tag{3.21}$$

Applying this to to $\log\left(\sigma + \lambda\rho^{(1)}\right)$, we can identify

$$\mathcal{L}_{ij} = \frac{\log\lambda_i - \log\lambda_j}{\lambda_i - \lambda_j}\rho^{(1)}_{ij} = \frac{A(\lambda_j/\lambda_i)}{\lambda_i}\rho^{(1)}_{ij}, \tag{3.22}$$

where

$$A(x) = \frac{\log x}{x - 1}. \tag{3.23}$$

If $\rho^{(1)}_{ij}$ is also diagonal with eigenvalues $\lambda^{(1)}_i$, then $\mathcal{L}$ is diagonal with eigenvalues $\lambda^{(1)}_i/\lambda_i$:

$$\mathcal{L}_{ij} = \frac{\lambda^{(1)}_i A(\lambda_j/\lambda_i)}{\lambda_i}\delta_{ij} = \frac{\lambda^{(1)}_i}{\lambda_i}\delta_{ij}. \tag{3.24}$$

---

[8]The factor of 1/2 difference compared to [49] is due to the factor of 1/2 in the quadratic term in (3.14).
[9]This follows directly from the definition since $\langle\Delta K\rangle^2_\rho = S(\rho\|\sigma)^2 = \mathcal{O}(\lambda^4)$ and $\Delta K^2 = \lambda^2\mathcal{L}^2 + \mathcal{O}(\lambda^3)$ where $\Delta K = \log\rho - \log\sigma$.

where we used $A(1) = 1$. With these ingredients, we can prove theorem 1. We prove that $\mathrm{Tr}(\sigma \mathcal{L}^2) \geq \mathrm{Tr}(\rho^{(1)}\mathcal{L})$ with an equality if and only if $\left[\sigma, \rho^{(1)}\right] = 0$. Applying this inequality to $V^{(2)}(\rho\|\sigma) = 2\,\mathrm{Tr}(\sigma \mathcal{L}^2)$ then proves the lower bound. We emphasize that the proof is inherently finite dimensional and does not directly apply to infinite dimensional Hilbert spaces.

*Proof.* Assume $\left[\sigma, \rho^{(1)}\right] \neq 0$. In the eigenbasis of $\sigma$, we can write

$$\mathrm{Tr}\left(\sigma \mathcal{L}^2\right) = \sum_{\lambda_i < \lambda_j} \lambda_i \mathcal{L}_{ij}\mathcal{L}_{ji} + \sum_{\lambda_i > \lambda_j} \lambda_i \mathcal{L}_{ij}\mathcal{L}_{ji} + \sum_{\lambda_i = \lambda_j} \lambda_i \mathcal{L}_{ij}\mathcal{L}_{ji} \tag{3.25}$$

$$= \sum_{\lambda_i < \lambda_j} A(\lambda_j/\lambda_i)\rho^{(1)}_{ij}\mathcal{L}_{ji} + \sum_{\lambda_i > \lambda_j} A(\lambda_j/\lambda_i)\rho^{(1)}_{ij}\mathcal{L}_{ji} + \sum_{\lambda_i = \lambda_j} A(\lambda_j/\lambda_i)\rho^{(1)}_{ij}\mathcal{L}_{ji}, \tag{3.26}$$

where on the second line, we used (3.22). Using

$$A(\lambda_j/\lambda_i) = (\lambda_i/\lambda_j)A(\lambda_i/\lambda_j) \tag{3.27}$$

and relabeling the dummy indices $i \leftrightarrow j$, the second term can be written as

$$\sum_{\lambda_i > \lambda_j} (\lambda_i/\lambda_j)A(\lambda_i/\lambda_j)\rho^{(1)}_{ij}\mathcal{L}_{ji} = \sum_{\lambda_i < \lambda_j} (\lambda_j/\lambda_i)A(\lambda_j/\lambda_i)\rho^{(1)*}_{ij}\mathcal{L}_{ij}, \tag{3.28}$$

where

$$\rho^{(1)*}_{ij}\mathcal{L}_{ij} = \frac{\log \lambda_i - \log \lambda_j}{\lambda_i - \lambda_j}\big|\rho^{(1)}_{ij}\big|^2 = \rho^{(1)}_{ij}\mathcal{L}_{ji} \tag{3.29}$$

is symmetric in $i, j$. Thus the second term in (3.26) can be written as

$$\sum_{\lambda_i < \lambda_j} (\lambda_j/\lambda_i)A(\lambda_j/\lambda_i)\rho^{(1)}_{ij}\mathcal{L}_{ji}. \tag{3.30}$$

We get

$$\mathrm{Tr}\left(\sigma \mathcal{L}^2\right) = \sum_{\lambda_i < \lambda_j} B(\lambda_j/\lambda_i)\rho^{(1)}_{ij}\mathcal{L}_{ji} + \sum_{\lambda_i = \lambda_j} \rho^{(1)}_{ij}\mathcal{L}_{ji}, \tag{3.31}$$

where

$$B(x) = (1 + x)A(x) = \frac{x + 1}{x - 1}\log x. \tag{3.32}$$

We also used $A(1) = 1$ in the diagonal term. As illustrated in Figure 2, it can be shown that

$$B(x) > 2, \quad \text{when} \quad x > 1. \tag{3.33}$$

Because of this and $\rho^{(1)}_{ij}\mathcal{L}_{ji} > 0$, when $\lambda_i < \lambda_j$, we get

$$\mathrm{Tr}\left(\sigma \mathcal{L}^2\right) > 2\sum_{\lambda_i < \lambda_j} \rho^{(1)}_{ij}\mathcal{L}_{ji} + \sum_{\lambda_i = \lambda_j} \rho^{(1)}_{ij}\mathcal{L}_{ji} = \mathrm{Tr}(\rho^{(1)}\mathcal{L}), \tag{3.34}$$

where the final equality follows by using the symmetricity of $\rho^{(1)}_{ij}\mathcal{L}_{ji}$. We finally get

$$V^{(2)}(\rho\|\sigma) > 2S^{(2)}(\rho\|\sigma), \tag{3.35}$$

when $\left[\sigma, \rho^{(1)}\right] \neq 0$. Assuming $\left[\sigma, \rho^{(1)}\right] = 0$, the cross-terms vanish in (3.26) and $V^{(2)}(\rho\|\sigma) = 2S^{(2)}(\rho\|\sigma)$ by $A(1) = 1$.

$\square$

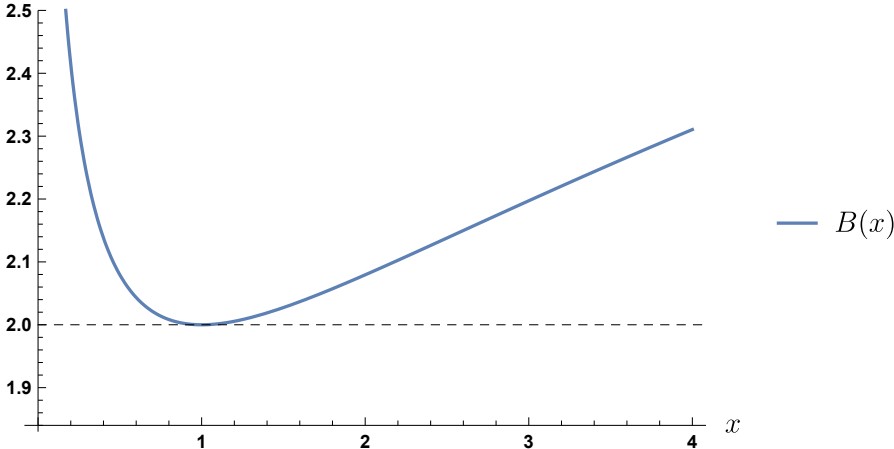

Figure 2: The function $B(x) = \frac{x+1}{x-1}\log x$. It has a global minimum $B(1) = 2$ in the region $x > 0$.

An interesting question is whether there exists special classes of density matrices for which there is also a constant upper bound for the ratio (3.19). Such an upper bound would imply an upper bound for the perturbative variance by perturbative relative entropy. To gain more intuition, it is useful to study the lower bound (3.20) in explicit examples. At least in the simple examples studied next, no upper bound appears.[10]

### 3.2.1 Single qubit

We consider a single qubit example for which the Hilbert space is two dimensional. A general initial density matrix $\sigma$ has two eigenvalues which we parametrize as $\frac{1}{2} + a$ and $\frac{1}{2} - a$ with $-\frac{1}{2} < a < \frac{1}{2}$. Working in the eigenbasis of $\sigma$, we consider the following one-parameter family of states $\rho(\lambda) = \sigma + \lambda\rho^{(1)}$:

$$\sigma = \frac{1}{2}\begin{pmatrix} 1+2a & 0 \\ 0 & 1-2a \end{pmatrix}, \quad \rho^{(1)} = \frac{1}{2}\begin{pmatrix} 0 & 1 \\ 1 & 0 \end{pmatrix}, \quad \rho(\lambda) = \frac{1}{2}\begin{pmatrix} 1+2a & \lambda \\ \lambda & 1-2a \end{pmatrix}, \quad (3.36)$$

where $a, \lambda \in \mathbb{R}$. The eigenvalues of $\rho(\lambda)$ are

$$p_\pm = \frac{1}{2}(1 \pm \sqrt{\lambda^2 + 4a^2}) \tag{3.37}$$

and the positivity of $p_-$ requires that

$$\lambda^2 \le (1-2a)(1+2a). \tag{3.38}$$

We can now demonstrate the lower bound (3.20) for the family $\rho(\lambda)$. The commutator between the initial state and the perturbation vanishes if and only if $a = 0$:

$$[\sigma, \rho^{(1)}] = a\begin{pmatrix} 0 & 1 \\ -1 & 0 \end{pmatrix}. \tag{3.39}$$

Hence we expect saturation of the lower bound when $a = 0$. Relative entropy and its variance can be explicitly computed for the states (3.36), but the expressions are quite complicated.

---

[10]An additional example will be presented in section 6.1.3, where relative entropy and its variance are derived for a spinless fermion chain.

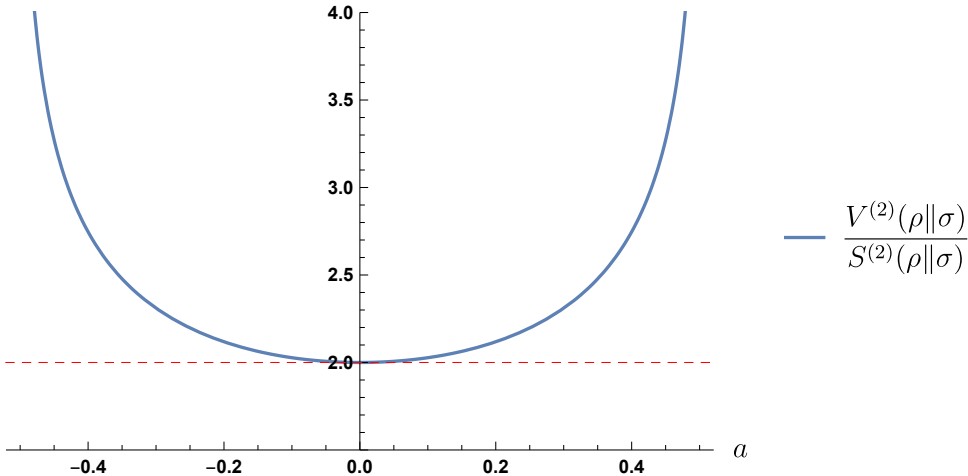

Figure 3: The variance over entropy ratio (3.43) as a function of $-\frac{1}{2} < a < \frac{1}{2}$. It saturates the lower bound (red line) at $a = 0$ where we have $[\sigma, \rho^{(1)}] = 0$.

For $a = 0$ they are

$$S(\rho\|\sigma)|_{a=0} = \frac{1}{2}\left[(1+\lambda)\log(1+\lambda) + (1-\lambda)\log(1-\lambda)\right] = \frac{\lambda^2}{2} + \mathcal{O}(\lambda^3) \tag{3.40}$$

$$V(\rho\|\sigma)|_{a=0} = \frac{1-\lambda^2}{4}\left[\log(1+\lambda) - \log(1-\lambda)\right]^2 = \lambda^2 + \mathcal{O}(\lambda^3) \tag{3.41}$$

so that the lower bound is saturated as expected in this case. For $a \neq 0$ we can expand the non-perturbative expressions of $S, V$ or use the perturbative formulas (3.15) and (3.18) directly. The results agree and are given by

$$S^{(2)}(\rho\|\sigma) = \frac{1}{4a}\log\left(\frac{1+2a}{1-2a}\right), \quad V^{(2)}(\rho\|\sigma) = \frac{1}{8a^2}\log^2\left(\frac{1+2a}{1-2a}\right) = 2S^{(2)}(\rho\|\sigma)^2. \tag{3.42}$$

We find that the ratio obeys the lower bound

$$\frac{V^{(2)}(\rho\|\sigma)}{S^{(2)}(\rho\|\sigma)} = \frac{1}{2a}\log\left(\frac{1+2a}{1-2a}\right) \geq 2 \tag{3.43}$$

with an equality if and only if $a = 0$ as required by Theorem 1. The ratio is depicted in Figure 3.

As an additional application of this example, we demonstrate the vanishing property of the variance described in Appendix B.2. One can see from (3.41) that $V(\rho\|\sigma)$ vanishes at three distinct points when $a = 0$:

$$V(\rho\|\sigma)|_{a=0} = 0, \quad \lambda = 0, \pm 1. \tag{3.44}$$

When $a \neq 0$, only the zero at $\lambda = 0$ remains corresponding to $V(\sigma\|\sigma) = 0$. The two extra zeros at $a = 0$ are explained by the vanishing theorem which states that

$$V(\rho\|\sigma) = 0, \quad \Longleftrightarrow \quad \begin{cases} \langle\phi|\sigma|\psi\rangle = 0, & \forall|\phi\rangle \in \ker\rho, \, \forall|\psi\rangle \in (\ker\rho)^{\perp} \\ \langle\psi_1|\rho|\psi_2\rangle \propto \langle\psi_1|\sigma|\psi_2\rangle, & \forall|\psi_1\rangle, |\psi_2\rangle \in (\ker\rho)^{\perp} \end{cases}, \tag{3.45}$$

where both of the conditions on the right have to be satisfied at the same time. When $\rho$ is full-rank ($\ker\rho = \{0\}$), the condition on the right hand side reduces to $\rho = \sigma$.[11] Hence for full-rank $\rho$, the variance vanishes if and only if $\rho = \sigma$, but there can be additional zeros otherwise.

---

[11] The proportionality constant is fixed by normalization to be the same.

In our qubit example, $\rho(\lambda)$ is full-rank except when the inequality (3.38) is saturated:

$$\lambda = \pm\sqrt{1-(2a)^2}. \tag{3.46}$$

At saturation, $\ker\rho$ one-dimensional and spanned by the vector $(1,-\mu)^{\mathsf{T}}$ where

$$\mu = \frac{\sqrt{1-(2a)^2}}{1-2a}. \tag{3.47}$$

Then the orthogonal complement $(\ker\rho)^{\perp}$ is spanned by $(1,\mu^{-1})^{\mathsf{T}}$. One can check that the second condition on the right hand side of (3.45) is satisfied for all $a$, but the first condition holds only for $a = 0$ corresponding to $\lambda = \pm 1$.

### 3.2.2 Maximally mixed initial state

In the above single qubit example, the lower bound is saturated when $\sigma$ is proportional to the identity matrix, or in other words, when $\sigma$ is maximally mixed. This should hold more generally for *arbitrary* perturbations $\rho^{(1)}$ in Hilbert spaces of dimension $N \geq 2$, because the identity matrix commutes with all matrices. So let $\mathbf{1}_N$ be the $N$-dimensional identity matrix and let $\sigma = (1/N)\mathbf{1}_N \equiv \sigma_{\max}$ be maximally mixed. To check saturation of the lower bound (3.20) we can use the fact that relative entropy and relative entropy variance generally reduce to von Neumann entropy $S(\rho)$ and capacity[12] $C(\rho)$ when $\sigma = \sigma_{\max}$:

$$S(\rho\|\sigma_{\max}) = -S(\rho) + \log N, \quad V(\rho\|\sigma_{\max}) = C(\rho), \tag{3.48}$$

where $\rho$ is arbitrary. Computing the expansions of von Neumann entropy and capacity explicitly using $\rho = \sigma_{\max} + \lambda\rho^{(1)} + \mathcal{O}(\lambda^2)$, we find

$$S(\rho) = \log N + \frac{\lambda^2}{2}S^{(2)}(\rho) + \mathcal{O}(\lambda^3), \quad C(\rho) = \frac{\lambda^2}{2}C^{(2)}(\rho) + \mathcal{O}(\lambda^3), \tag{3.49}$$

where[13]

$$C^{(2)}(\rho) = 2S^{(2)}(\rho) = 2N\operatorname{Tr}\left(\rho^{(1)}\right)^2. \tag{3.50}$$

Combining with (3.48), we get

$$\frac{V^{(2)}(\rho\|\sigma_{\max})}{S^{(2)}(\rho\|\sigma_{\max})} = \frac{C^{(2)}(\rho)}{S^{(2)}(\rho)} = 2 \tag{3.51}$$

as expected.

### 3.2.3 Two thermal states

Let us consider two thermal states $\rho_2$ and $\rho_1$ of the form

$$\rho_1 = \frac{e^{-\beta_1 H}}{\operatorname{Tr} e^{-\beta_1 H}}, \qquad \rho_2 = \frac{e^{-\beta_2 H}}{\operatorname{Tr} e^{-\beta_2 H}}. \tag{3.52}$$

When the Hamiltonian $H$ is quadratic in creation/annihilation operators, the states are Gaussian, so the result should reduce to the previously studied case in [20]. With a straightforward calculation, we obtain

$$V(\rho_2\|\rho_1) = (\beta_2 - \beta_1)^2 \left[\langle H^2\rangle_{\beta_2} - \langle H\rangle_{\beta_2}^2\right], \tag{3.53}$$

---

[12]By capacity we mean the quantity $C(\rho) = \operatorname{Tr}[\rho(\log\rho)^2] - S(\rho)^2$, which for a reduced density matrix is known as the capacity of entanglement (other names include for example variance of surprisal and varentropy), see Appendix B.1. For a thermal state $\rho_\beta$, it becomes the heat capacity $C(\beta)$.

[13]This is of course in agreement with the general definitions for $V^{(2)}(\rho\|\sigma_{\max})$ and $S^{(2)}(\rho\|\sigma_{\max})$.

where all the terms involving logarithms of traces have cancelled. From this equation we recognize the heat capacity $C(\beta_2)$ of a thermal state and we end up with a simple result

$$V(\rho_2\|\rho_1) = \left(1 - \frac{\beta_1}{\beta_2}\right)^2 C(\beta_2) \,. \tag{3.54}$$

In the limit $\beta_1 \to 0$, $\rho_1$ becomes a maximally mixed state, and the relative entropy variance reduces to the heat capacity,

$$V(\rho_2\|\rho_1) = C(\beta_2) \,. \tag{3.55}$$

On the other hand, in the limit $\beta_2 \to \infty$, $\rho_2$ reduces to the ground state, and the relative entropy variance vanishes (along with $C(\beta_2) \to 0$).[14]

Clearly, $[\rho_1, \rho_2] = 0$ for all temperatures $\beta_1$ and $\beta_2$ so that the lower bound (3.20) should be saturated for temperature perturbations $\beta_2 = \beta_1 + \lambda \beta^{(1)} + \mathcal{O}(\lambda^2)$. We can check this explicitly. Relative entropy is given by

$$S(\rho_2\|\rho_1) = -(\beta_2 - \beta_1)\langle H \rangle_{\beta_2} - \log\frac{\operatorname{Tr} e^{-\beta_2 H}}{\operatorname{Tr} e^{-\beta_1 H}} \tag{3.56}$$

which expanded to second order in $\lambda$ gives

$$S^{(2)}(\rho_2\|\rho_1) = \left(\frac{\beta^{(1)}}{\beta_1}\right)^2 C(\beta_1) \,, \tag{3.57}$$

where $C(\beta_1)$ is the heat capacity of the initial thermal state $\rho_1$. Because $(\beta_2 - \beta_1)^2$ is second order in $\lambda$, we can just replace $C(\beta_2)$ by its initial value $C(\beta_1)$ to obtain variance of relative entropy (3.54) at order $\mathcal{O}(\lambda^2)$. We get

$$V^{(2)}(\rho\|\sigma) = 2\left(\frac{\beta^{(1)}}{\beta_1}\right)^2 C(\beta_1) = 2S^{(2)}(\rho\|\sigma) \,, \tag{3.58}$$

which saturates the bound (3.20).

Interestingly, non-perturbative relative entropy variance between two thermal states turns out to be proportional to the capacity of entanglement (3.54). This might have implications for thermodynamics of AdS black holes in the AdS/CFT correspondence where the holographic dual of the capacity of entanglement is known [50, 51]. However, the holographic dual of relative entropy variance is not yet known, but further results in this direction will be reported in upcoming work [52].

## 3.3 Relation to parameter estimation

The framework of perturbative asymmetric hypothesis testing is related to parameter estimation and quantum Fisher information [53]. Quantum parameter estimation is the problem of determining the value of a parameter $\lambda$ appearing in a density matrix $\rho(\lambda)$ by performing $n$ independent measurements of an observable $E(x)$. For each measurement, the probability of the outcome $x$ is

$$p(x|\lambda) = \operatorname{Tr}\rho(\lambda)E(x), \qquad \sum_x E(x) = \mathbf{1} \,. \tag{3.59}$$

Denoting the outcomes of $n$ measurements by $x_i$, which are random variables, an estimator is a function $\lambda_{\text{est}} = \lambda_{\text{est}}(x_1, \ldots, x_n)$ used to estimate $\lambda$ from the data $\{x_i\}$. Suppose that the estimator is unbiased so that

$$\langle \lambda_{\text{est}} \rangle \equiv \int d^n x \, p(x_1|\lambda) \cdots p(x_n|\lambda) \lambda_{\text{est}}(x_1, \ldots, x_n) = \lambda \,, \tag{3.60}$$

---

[14]In a system with a degenerate ground state, at zero temperature the density matrix reduces to a flat state (all non-zero eigenvalues are equal), for which the capacity of entanglement is zero [43, 51].

the quantum Cramér–Rao bound then states that

$$\langle(\lambda_{\text{est}}-\lambda)^2\rangle \geq \frac{1}{nF_\lambda} \, , \tag{3.61}$$

where

$$F_\lambda = \text{Tr}(\rho\, L_\lambda^2) = \text{Tr}\left(\frac{d\rho}{d\lambda}L_\lambda\right) , \tag{3.62}$$

is the quantum Fisher information [54]. Here, the symmetric logarithmic derivative operator $L_\lambda$ is defined implicitly via

$$\frac{d\rho}{d\lambda} = \frac{1}{2}(L_\lambda\rho + \rho L_\lambda) \, . \tag{3.63}$$

We focus on states $\rho(\lambda) = \sigma + \lambda\rho^{(1)}$ with $\lambda \ll 1$ that are perturbatively close to $\rho(0) = \sigma$. Setting $\lambda = 0$ in the above equations gives

$$\langle\lambda_{\text{est}}^2\rangle \geq \frac{1}{nF} \, , \tag{3.64}$$

with

$$F \equiv \text{Tr}(\sigma L^2) = \text{Tr}(\rho^{(1)}L), \qquad \rho^{(1)} = \frac{1}{2}(L\sigma + \sigma L). \tag{3.65}$$

The bound (3.64) gives the best accuracy for estimating the small parameter $\lambda$.

Quantum Fisher information (3.65) is closely related to perturbative relative entropy[15] which has a similar expression (3.15). In the eigenbasis of $\sigma$ with eigenvalues $\lambda_i$, the symmetric logarithmic derivative has the expression

$$L_{ij} = \frac{2}{\lambda_i + \lambda_j}\rho_{ij}^{(1)} \, , \tag{3.66}$$

and can be compared with the expression (3.22) for the logarithmic derivative $\mathcal{L}$. When $[\sigma, \rho^{(1)}] = 0$, the two expressions are equal: we have $\mathcal{L}_{ij} = L_{ij} = (\lambda_i^{(1)}/\lambda_i)\delta_{ij}$ where $\lambda_i^{(1)}$ are the eigenvalues of $\rho^{(1)}$ in the eigenbasis of $\sigma$. In general, we can prove the following inequality whose proof is similar to the proof of Theorem 1.

---

[15]The definition of quantum Fisher information is not unique and different ones can be found in the literature. In [55], a divergence-based Fisher information $J$ is introduced and is defined to be exactly equal to the perturbative relative entropy $J \equiv S^{(2)}(\rho\|\sigma)$. The same definition is also used in [49].

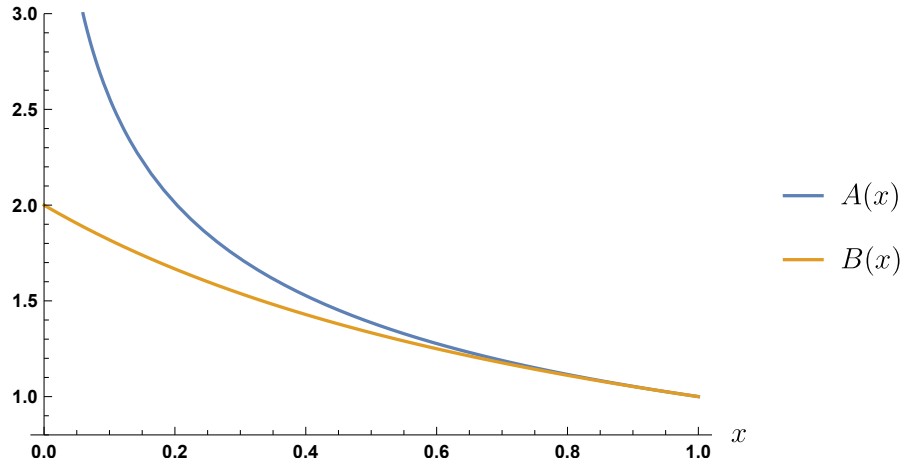

Figure 4: Plot of the functions $A(x) = \frac{\log x}{x-1}$ (blue) and $B(x) = \frac{2}{x+1}$ (yellow) for $0 \leq x \leq 1$. They satisfy the inequality $A(x) \geq B(x)$ in this region.

**Theorem 2.** *Consider the perturbative expansion $\rho = \sigma + \lambda \rho^{(1)} + \frac{\lambda^2}{2}\rho^{(2)} + \mathcal{O}(\lambda^3)$, we have*

$$F \leq S^{(2)}(\rho \| \sigma) \tag{3.67}$$

*with an equality if and only if $[\sigma, \rho^{(1)}] = 0$.*

*Proof.* Assuming $[\sigma, \rho^{(1)}] \neq 0$, we have

$$S^{(2)}(\rho \| \sigma) = \text{Tr}(\rho^{(1)} \mathcal{L}) = \sum_{\lambda_i > \lambda_j} \frac{A(\lambda_j/\lambda_i)}{\lambda_j} |\rho^{(1)}_{ij}|^2 + \sum_{\lambda_i < \lambda_j} \frac{A(\lambda_j/\lambda_i)}{\lambda_j} |\rho^{(1)}_{ij}|^2 + \sum_{\lambda_i = \lambda_j} \rho^{(1)}_{ij} \mathcal{L}_{ji}, \tag{3.68}$$

where $A(x) = \frac{\log x}{x-1}$. Using that

$$\frac{A(\lambda_j/\lambda_i)}{\lambda_j} = \frac{A(\lambda_i/\lambda_j)}{\lambda_i}, \tag{3.69}$$

and relabeling the dummy indices $i \longleftrightarrow j$ in the second term, we get

$$S^{(2)}(\rho \| \sigma) = 2 \sum_{\lambda_i > \lambda_j} \frac{A(\lambda_j/\lambda_i)}{\lambda_j} |\rho^{(1)}_{ij}|^2 + \sum_{\lambda_i = \lambda_j} \rho^{(1)}_{ij} L_{ji}, \tag{3.70}$$

where we used $\mathcal{L}_{ii} = L_{ii}$ in the last term. Applying the inequality

$$A(x) = \frac{\log x}{x-1} \geq \frac{2}{x+1} \equiv B(x), \tag{3.71}$$

which is displayed in Figure 4, we obtain

$$S^{(2)}(\rho \| \sigma) > 2 \sum_{\lambda_i > \lambda_j} \frac{B(\lambda_j/\lambda_i)}{\lambda_j} |\rho^{(1)}_{ij}|^2 + \sum_{\lambda_i = \lambda_j} \rho^{(1)}_{ij} L_{ji} = \text{Tr}(\rho^{(1)} L) = F, \tag{3.72}$$

where the inequality is strict. Assuming $[\sigma, \rho^{(1)}] = 0$, the cross-terms vanish in (3.68) and $S^{(2)}(\rho \| \sigma) = F$ by $\mathcal{L}_{ii} = L_{ii}$. $\qquad\square$

We can also combine (3.67) with the lower bound (3.20) to give

$$2F \leq 2S^{(2)}(\rho\|\sigma) \leq V^{(2)}(\rho\|\sigma) \tag{3.73}$$

with equality if and only if $[\sigma, \rho^{(1)}] = 0$. This shows that both $S^{(2)}$ and $V^{(2)}/2$ give quantum Cramér–Rao bounds, although the quantum Fisher information $F$ provides the tightest bound.

The inequality (3.67) provides a heuristic connection between perturbative hypothesis testing and parameter estimation. Suppose that the estimator is asymptotically normal, that is the probability distribution for the value of the estimator[16] is effectively described by a Gaussian distribution for large $n$. Then the Cramér–Rao bound (3.64) implies that the optimal probability distribution for the estimate is

$$f_n^*(\lambda_{\text{est}}) \sim e^{-n(\lambda_{\text{est}}^2/2)F}, \qquad n \to \infty \,. \tag{3.74}$$

This distribution (3.74) is similar to the optimal type II error probability in asymmetric hypothesis testing (2.13) between two perturbatively close states $\sigma$ and $\rho(\lambda) = \sigma + \lambda\rho^{(1)}$:

$$\beta_n^* \sim e^{-n(\lambda^2/2)S^{(2)}(\rho\|\sigma)}, \qquad n \to \infty \,, \tag{3.75}$$

where $\lambda$ is fixed here. The inequality (3.67) then implies that

$$\beta_n^* \lesssim f_n^*(\lambda), \qquad n \to \infty. \tag{3.76}$$

This can be interpreted heuristically as follows: the binary problem of distinguishing $\rho(\lambda)$ from $\sigma$ is easier than estimating the exact value of $\lambda$.

# 4 Generalities on measurements

In this section, we compare different measurement protocols in a setting where we have a large number $n$ of copies of a physical system. We begin by discussing independent measurements on the $n$ copies, and explain why they fail to be optimal. We then turn to optimal measurements for distinguishing between two states $\rho$ and $\sigma$ in the context of asymmetric hypothesis testing. Following section 2.2, we call a measurement optimal if it saturates the refined quantum Stein's lemma in the asymptotic limit $n \to +\infty$. We would like to understand this optimal measurement in order to apply it in many-body systems in the remainder of this paper. We also consider the likelihood ratio test, which is optimal among the classical measurements. Simple examples where these measurements can be described and tested are then discussed. In Appendix A, we describe and discuss similar measurements for symmetric hypothesis testing.

We recall that we take $n$ copies of the system so that we have to distinguish between the states $\rho^{\otimes n}$ and $\sigma^{\otimes n}$ in the asymptotic limit $n \to +\infty$. More precisely, we look for a Hermitian operator $A^{(n)}$ with $0 \leq A^{(n)} \leq 1$ which minimizes the type II error $\beta_n = \text{Tr}\,\sigma^{\otimes n}A^{(n)}$ while ensuring that the type I error $\alpha_n = \text{Tr}\,\rho^{\otimes n}(1 - A^{(n)})$ remains bounded.

## 4.1 Independent measurements

The likelihood ratio test and the optimal measurement, which are described below, use in a crucial way correlations between the $n$ copies. In this section, we demonstrate that independent measurements perform badly. A trivial but notable exception is the case where $\rho$ is a

---

[16]We denote the estimator (a random variable) and its value (an estimate) by the same symbol.

pure state, for which the optimal measurement is simply the projector onto this pure state on each copy. This example is discussed in section 4.4.1.

Let's consider an independent measurement, by which we mean a factorized measurement of the form

$$A^{(n)} = A_1^{(n)} \otimes A_2^{(n)} \otimes \cdots \otimes A_n^{(n)} , \tag{4.1}$$

and denote

$$a_i^{(n)} = \operatorname{Tr} \rho A_i^{(n)}, \qquad b_i^{(n)} = \operatorname{Tr} \sigma A_i^{(n)} , \tag{4.2}$$

which satisfy $0 < a_i < 1$ and $0 < b_i < 1$. The type I and type II errors are then given by

$$\alpha_n = 1 - \prod_{i=1}^{n} a_i^{(n)}, \qquad \beta_n = \prod_{i=1}^{n} b_i^{(n)} . \tag{4.3}$$

We see that the type I error $\alpha_n$ becomes dangerously uncontrolled in the asymptotic limit. To obtain a bounded type I error, we have to make the $a_i^{(n)}$ tend to 1 as $n \to \infty$. This implies that the operators $A_i^{(n)}$ should become close to the identity. This will make the $b_i^{(n)}$ also close to one and spoil the type II error $\beta_n$.

To illustrate this argument, consider the following example. Let's pick

$$A_i^{(n)} = 1 - \frac{1}{n} B , \tag{4.4}$$

where $B$ is some bounded positive Hermitian operator. This ensures that the type I error remains smaller than 1, since we have

$$\alpha_n = 1 - \left(1 - \frac{1}{n} \operatorname{Tr} \rho B \right)^n \underset{n \to \infty}{\sim} 1 - e^{-\operatorname{Tr} \rho B} . \tag{4.5}$$

However, we see that the type II error is

$$\beta_n = \left(1 - \frac{1}{n} \operatorname{Tr} \sigma B \right)^n \underset{n \to \infty}{\sim} e^{-\operatorname{Tr} \sigma B} . \tag{4.6}$$

Thus we see that $\beta_n$ goes to a finite limit as $n \to \infty$, instead of decaying exponentially to zero, as in an optimal measurement. Hence, we expect that in general independent measurements should be far from optimal.

We can reformulate the independent measurement optimization as follows. Denote

$$a_i^{(n)} = 1 - \epsilon_i^{(n)} = e^{-v_i^{(n)}} . \tag{4.7}$$

We then have to impose $\sum_i v_i^{(n)} \leq -\log(1 - \varepsilon)$ while at the same time optimizing $\sum_i \beta_1^*(\epsilon_i^{(n)}) = \sum_i \beta_1^*(1 - e^{-v_i^{(n)}})$. This leads us to consider the function $f(x) \equiv \beta_1^*(1 - e^{-x})$. We need to optimize $\sum_i f(v_i^{(n)})$ subject to the constraint $\sum_i v_i^{(n)} \leq -\log(1 - \varepsilon)$. If the function $f(x)$ is convex, the optimal choice is to choose one of the $v_i^{(n)}$ to be equal to $-\log(1 - \varepsilon)$ while taking the others to be equal to zero. In other words, multiple measurements yield in this case no improvement over a single measurement.

If, on the other hand, $f(x)$ is concave, then the optimal choice is to choose all $v_i^{(n)}$ equal to each other, and the resulting error is

$$\beta_1^*(1 - (1 - \varepsilon)^{1/n})^n , \tag{4.8}$$

whose detailed form for large $n$ depends on the small $\varepsilon$ behavior of $\beta_1^*(\varepsilon)$. Of course, if $f(x)$ is neither concave or convex, a more detailed analysis is required.

## 4.2 Optimal measurement

Let's now describe an optimal measurement which was used in [37] to prove the quantum Stein's lemma. Although we will often refer to it as *the* optimal measurement, it is important to note that it is not unique.[17] We define the modular Hamiltonians $K$ and $\widetilde{K}$ by

$$K \equiv -\log \sigma, \qquad \widetilde{K} \equiv -\log \rho . \tag{4.9}$$

We consider $n$ copies of the system with the states $\sigma^{\otimes n}$ and $\rho^{\otimes n}$ labeled by $i = 1, \ldots, n$. We denote by $\{|\mathbf{E}\rangle\}$ and $\{|\widetilde{\mathbf{E}}\rangle\}$ the set of normalized eigenstates of $\sigma^{\otimes n}$ and $\rho^{\otimes n}$. They are of the form

$$
\begin{aligned}
|\mathbf{E}\rangle &= |E_1\rangle \otimes |E_2\rangle \otimes \cdots \otimes |E_n\rangle , \\
|\widetilde{\mathbf{E}}\rangle &= |\widetilde{E}_1\rangle \otimes |\widetilde{E}_2\rangle \otimes \cdots \otimes |\widetilde{E}_n\rangle ,
\end{aligned} \tag{4.10}
$$

and are labeled by their eigenvalues of $K$ and $\widetilde{K}$ respectively. We can define the average modular operators

$$
\begin{aligned}
K^{(n)} &= -\frac{1}{n}\log\sigma^{\otimes n} = \frac{1}{n}\sum_{i=1}^{n} K_i , \\
\widetilde{K}^{(n)} &= -\frac{1}{n}\log\rho^{\otimes n} = \frac{1}{n}\sum_{i=1}^{n} \widetilde{K}_i .
\end{aligned} \tag{4.11}
$$

We will use the notation $|\mathbf{E}|$ and $|\widetilde{\mathbf{E}}|$ to denote the eigenvalues of the states $|\mathbf{E}\rangle$ and $|\widetilde{\mathbf{E}}\rangle$ for the average modular operators. In other words,

$$|\mathbf{E}| = \frac{1}{n}\sum_{i=1}^{n} E_i, \qquad |\widetilde{\mathbf{E}}| = \frac{1}{n}\sum_{i=1}^{n} \widetilde{E}_i . \tag{4.12}$$

To describe the optimal measurement, we decompose the state $|\widetilde{\mathbf{E}}\rangle$ in the $\{|\mathbf{E}\rangle\}$ basis

$$|\widetilde{\mathbf{E}}\rangle = \sum_{\mathbf{E}} \langle \mathbf{E}|\widetilde{\mathbf{E}}\rangle |\mathbf{E}\rangle . \tag{4.13}$$

We then restrict the sum only to the states $|\mathbf{E}\rangle$ satisfying the acceptance condition $|\mathbf{E}| - |\widetilde{\mathbf{E}}| \geq \mathcal{E}$ for some fixed $\mathcal{E}$ that we will call the acceptance threshold. This defines the states

$$|\xi(\widetilde{\mathbf{E}})\rangle = \sum_{\mathbf{E}\,:\,|\mathbf{E}|-|\widetilde{\mathbf{E}}|\geq\mathcal{E}} \langle \mathbf{E}|\widetilde{\mathbf{E}}\rangle |\mathbf{E}\rangle . \tag{4.14}$$

We define the acceptance subspace

$$\mathcal{H}_Q = \operatorname*{span}_{\widetilde{\mathbf{E}}} \{|\xi(\widetilde{\mathbf{E}})\rangle\} . \tag{4.15}$$

The optimal measurement is then the projection onto this subspace:

$$A^{(n)} = P_{\mathcal{H}_Q} . \tag{4.16}$$

Unfortunately, explicit constructions of the acceptance subspace and the projection are nontrivial even in simple applications, as we will see.

---

[17]This is especially true since our definition of optimality relies on an asymptotic limit $n \to +\infty$. Any measurement satisfying (2.14) is considered optimal, so it is clear that there will be many optimal measurements.

To obtain the optimal type II error $\beta_n$ for a bounded type I error $\alpha_n \leq \varepsilon$, the optimal acceptance threshold is

$$\mathcal{E} = S(\rho\|\sigma) + \sqrt{\frac{V(\rho\|\sigma)}{n}}\Phi^{-1}(\varepsilon) . \tag{4.17}$$

As explained in section 2.2, this measurement leads to a bounded type I error $\alpha_n \leq \varepsilon$ and a type II error exponent

$$-\frac{1}{n}\log\beta_n \underset{n\to+\infty}{\sim} S(\rho\|\sigma) + \sqrt{\frac{V(\rho\|\sigma)}{n}}\Phi^{-1}(\varepsilon) + \mathcal{O}\left(\frac{\log n}{n}\right) . \tag{4.18}$$

The proof of optimality of this measurement is given in [37].

## 4.3  Likelihood ratio test

The optimal measurement described above is in general rather complicated to implement. In this section, we review a simpler measurement, which is efficient and becomes optimal in the classical case, when $\rho$ and $\sigma$ commute [25]. When $\rho$ and $\sigma$ are viewed as classical probability distributions, this measurement is the likelihood ratio (Neyman–Pearson) test which is known to be optimal in classical hypothesis testing.

In this setup, we consider two probability distributions $P$ and $Q$ on the same probability space $\Omega$, and we would like to distinguish them by making a test modeled as a function $A : \Omega \to [0,1]$. Let's consider $n$ copies of the system. The task is then to distinguish between the probability distributions $P^{(n)}$ and $Q^{(n)}$ on $\Omega^n$ defined as

$$P^{(n)}(x) = \prod_{i=1}^{n} P(x_i), \qquad Q^{(n)}(x) = \prod_{i=1}^{n} Q(x_i) , \tag{4.19}$$

with a function $A^{(n)} : \Omega^n \to [0,1]$. The optimal type II error is defined as

$$\beta_n^*(\varepsilon) = \inf_{A^{(n)}}\left\{\mathrm{E}_{Q^{(n)}}\left[A^{(n)}\right] \mid \mathrm{E}_{P^{(n)}}\left[1 - A^{(n)}\right] \leq \varepsilon\right\} , \tag{4.20}$$

where $\mathrm{E}_{\mathcal{P}}$ denotes the expected value in the probability distribution $\mathcal{P}$. We are interested in the asymptotic limit $n \to +\infty$. We have the estimate

$$-\frac{1}{n}\log\beta_n^*(\varepsilon) \underset{n\to+\infty}{\sim} S(P\|Q) + \sqrt{\frac{V(P\|Q)}{n}}\Phi^{-1}(\varepsilon) + \mathcal{O}\left(\frac{\log n}{n}\right) . \tag{4.21}$$

The first order in $n$ result was originally obtained by Chernoff and Stein and the second order correction by Strassen [56] (see [57] for a review). In the above expression, the relative entropy and its variance are defined as the first and second cumulant, in the probability distribution $P$, of the log-likelihood ratio $\log\frac{P(x)}{Q(x)}$, i.e.

$$S(P\|Q) = \sum_{x\in\Omega} P(x)\log\frac{P(x)}{Q(x)}, \qquad V(P\|Q) = \sum_{x\in\Omega} P(x)\left(\log\frac{P(x)}{Q(x)} - S(P\|Q)\right)^2 . \tag{4.22}$$

The measurement that achieves optimality (in this classical setting) is the likelihood ratio test. It is a deterministic test, choosing the function $A^{(n)}$ to be an indicator function

$$A^{(n)} = \mathbf{1}\left\{x \in \Omega^n \ \middle| \ \frac{1}{n}\log\frac{P^{(n)}(x)}{Q^{(n)}(x)} \geq \mathcal{E}\right\} , \tag{4.23}$$

which takes the value 1 on an acceptance subspace, the subset of $x \in \Omega^n$ satisfying the acceptance condition $\frac{1}{n} \log \frac{P^{(n)}(x)}{Q^{(n)}(x)} \geq \mathcal{E}$, and 0 otherwise. The optimal choice of threshold $\mathcal{E}$ is

$$\mathcal{E} = S(P\|Q) + \sqrt{\frac{V(P\|Q)}{n}} \Phi^{-1}(\varepsilon) . \tag{4.24}$$

To apply this measurement to quantum systems, we need to express it in quantum mechanical language using the setup described in the previous section. We take the probability space $\Omega = \{|\mathbf{E}\rangle\}$ to be a basis of eigenstates of $K^{(n)} = -\frac{1}{n} \log \sigma^{\otimes n}$. The probability distributions are the ensemble probabilities given by

$$P^{(n)}(\mathbf{E}) = \langle \mathbf{E}|\rho^{\otimes n}|\mathbf{E}\rangle, \qquad Q^{(n)}(\mathbf{E}) = \langle \mathbf{E}|\sigma^{\otimes n}|\mathbf{E}\rangle , \tag{4.25}$$

and we have $\frac{1}{n} \log Q^{(n)}(\mathbf{E}) = |\mathbf{E}|$ from the definition (4.12). The acceptance condition is

$$|\mathbf{E}| + \frac{1}{n} \log \langle \mathbf{E}|\rho^{\otimes n}|\mathbf{E}\rangle \geq \mathcal{E} , \tag{4.26}$$

which can also be written more transparently as

$$\frac{1}{n} \sum_{i=1}^{n} \langle E_i|K - \widetilde{K}|E_i\rangle \geq \mathcal{E} . \tag{4.27}$$

We note that this measurement only involves the diagonal part of $\rho$ (defined with respect to the basis defined by $\sigma$), which we denote

$$\rho_D \equiv \sum_E \langle E|\rho|E\rangle |E\rangle\langle E| . \tag{4.28}$$

We can then define the "classical" acceptance subspace

$$\mathcal{H}_C = \text{span} \left\{ |\mathbf{E}\rangle \,\middle|\, |\mathbf{E}| + \frac{1}{n} \log \langle \mathbf{E}|\rho^{\otimes n}|\mathbf{E}\rangle \geq \mathcal{E} \right\} . \tag{4.29}$$

To implement the likelihood ratio test, we then replace the indicator function of the acceptance subspace by an operator, the projector onto $\mathcal{H}_C$:

$$A^{(n)} = P_{\mathcal{H}_C} . \tag{4.30}$$

When $\rho$ and $\sigma$ commute, it can be seen that $\mathcal{H}_C = \mathcal{H}_Q$ so this is actually the optimal measurement described in the previous subsection. From the relation with classical quantities $S(P\|Q) = S(\rho_D\|\sigma)$ and $V(P\|Q) = V(\rho_D\|\sigma)$, we see that the optimal choice of threshold is

$$\mathcal{E} = S(\rho_D\|\sigma) + \sqrt{\frac{V(\rho_D\|\sigma)}{n}} \Phi^{-1}(\varepsilon) , \tag{4.31}$$

and leads to a bounded type I error $\alpha_n \leq \varepsilon$ and a type II error exponent

$$-\frac{1}{n} \log \beta_n \underset{n \to +\infty}{\sim} S(\rho_D\|\sigma) + \sqrt{\frac{V(\rho_D\|\sigma)}{n}} \Phi^{-1}(\varepsilon) + \mathcal{O}\left(\frac{\log n}{n}\right) . \tag{4.32}$$

In general, this measurement is less efficient than the optimal measurement because the monotonicity of relative entropy implies that

$$S(\rho_D\|\sigma) \leq S(\rho\|\sigma) \tag{4.33}$$

since the map $\rho \mapsto \rho_D$ is (completely) positive and trace preserving [58]. Nonetheless, this measurement achieves an exponentially decreasing type II error for bounded type I error. The likelihood ratio test with $n_{\mathrm{LRT}}$ copies of the system achieves the same accuracy to leading order as the optimal measurement with $n_{\mathrm{opt}}$ copies with

$$n_{\mathrm{LRT}} = \frac{S(\rho\|\sigma)}{S(\rho_D\|\sigma)} n_{\mathrm{opt}} \,. \tag{4.34}$$

In the simple example of a qubit, the likelihood ratio test can be implemented using a quantum circuit, displayed in Figure 5, and a comparison between the likelihood ratio test and the optimal measurement is shown in Figure 6.

### 4.4 Examples

In this section, we describe the optimal measurement in some simple cases.

#### 4.4.1 Pure versus mixed

We consider the simplest possible example. We take $\rho$ to be a pure state and $\sigma$ to be a general mixed state

$$\rho = |\psi\rangle\langle\psi|, \qquad \sigma = e^{-K}. \tag{4.35}$$

In this case, an optimal measurement is just the projector $A = |\psi\rangle\langle\psi|$. On $n$ copies of the system, we take the factorized measurement $A^{(n)} = A^{\otimes n}$. The type I error $\alpha_n = 0$ and the type II error is given by

$$-\frac{1}{n}\log\beta_n = \mathrm{Tr}\,\rho K = S(\rho\|\sigma)\,, \tag{4.36}$$

which indeed saturates the quantum Stein's lemma. The second order asymptotics in $n$ do not play a role because

$$V(\rho\|\sigma) = 0\,, \tag{4.37}$$

according to the proposition explained in section B.2.

#### 4.4.2 Global thermal states

We consider two thermal states with different temperatures

$$\rho = \frac{1}{\mathrm{Tr}\,e^{-\beta_2 H}} e^{-\beta_2 H}, \qquad \sigma = \frac{1}{\mathrm{Tr}\,e^{-\beta_1 H}} e^{-\beta_1 H} \tag{4.38}$$

and we would like to distinguish between them. The modular Hamiltonians are

$$\widetilde{K} = -\log\rho = \beta_2(H + F_2), \qquad K = -\log\sigma = \beta_1(H + F_1), \tag{4.39}$$

where the free energy is defined as $F_i = -\beta_i^{-1}\log\mathrm{Tr}\,e^{-\beta_i H}$ for $i = 1, 2$. The relative modular Hamiltonian is

$$\Delta K = K - \widetilde{K} = (\beta_1 - \beta_2)H + \beta_1 F_1 - \beta_2 F_2\,. \tag{4.40}$$

The relative entropy and variance are

$$
\begin{aligned}
S(\rho\|\sigma) &= \langle\Delta K\rangle_\rho = (\beta_1 - \beta_2)E_2 + \beta_1 F_1 - \beta_2 F_2\,, &\tag{4.41}\\
V(\rho\|\sigma) &= \langle\Delta K^2\rangle_\rho - \langle\Delta K\rangle_\rho^2 = (\beta_1 - \beta_2)^2(\langle H^2\rangle_\rho - \langle H\rangle_\rho^2) = \left(1 - \frac{\beta_1}{\beta_2}\right)^2 C(\beta_2)\,,
\end{aligned}
$$

where $E_2 = \langle H \rangle_\rho$. We are in a situation where $\rho$ and $\sigma$ commute so the likelihood ratio test is actually the optimal measurement. It can be described as follows. We consider $n$ copies of the system and we define the average

$$\Delta K^{(n)} = \frac{1}{n} \sum_{i=1}^{n} \Delta K_i \,. \tag{4.42}$$

Let $\{|\mathbf{E}\rangle\}$ be a basis of eigenstates of $\sigma^{\otimes n}$. These are formed from eigenstates of $H$. Notice that we are using the actual energies to label the states as opposed to using the eigenvalues of the modular Hamiltonian. In particular, we denote by $|\mathbf{E}|$ the average energy of the corresponding state

$$|\mathbf{E}| = \frac{1}{n} \sum_{i=1}^{n} E_i \,. \tag{4.43}$$

The measurement is simply the projection onto the states in this basis with the acceptance condition

$$\langle \mathbf{E} | \Delta K^{(n)} | \mathbf{E} \rangle \geq S(\rho \| \sigma) + \sqrt{\frac{V(\rho \| \sigma)}{n}} \Phi^{-1}(\varepsilon) \,. \tag{4.44}$$

This translates into the condition

$$(\beta_1 - \beta_2)|\mathbf{E}| \geq (\beta_1 - \beta_2) E_2 + \frac{\beta_1 - \beta_2}{\beta_2} \sqrt{\frac{C(\beta_2)}{n}} \Phi^{-1}(\varepsilon) \,. \tag{4.45}$$

We have to distinguish two cases depending on the sign of $\beta_1 - \beta_2$. The acceptance condition is

$$\begin{cases} |\mathbf{E}| \geq E_* & \beta_1 > \beta_2 \\ |\mathbf{E}| \leq E_* & \beta_1 < \beta_2 \end{cases}, \tag{4.46}$$

where the threshold energy is

$$E_* = E_2 + \frac{1}{\beta_2} \sqrt{\frac{C(\beta_2)}{n}} \Phi^{-1}(\varepsilon) \,. \tag{4.47}$$

The measurement is then a projection on the states satisfying the condition

$$A^{(n)} = \sum_{|\mathbf{E}| \gtrless E_*} |\mathbf{E}\rangle \langle \mathbf{E}| \,. \tag{4.48}$$

It is interesting to note that the optimal measurement actually doesn't depend on the value of $\beta_1$, but only on whether it is bigger or smaller than $\beta_2$.

# 5 Measurements of a qubit

In this section, we consider a simple system to illustrate the measurements that we have been discussing. The system is just a single qubit in two possible states $\rho$ or $\sigma$. We are interested in the optimal measurement on $n$ copies of the system in the asymptotic limit where $n$ is large.

## 5.1 Likelihood ratio test

The best classical measurement is the likelihood ratio test and was discussed in section 4.3. In this section, we will write it explicitly for the case of a qubit. We will also give a quantum circuit that realizes it.

### 5.1.1  Setup

Let $\{|0\rangle, |1\rangle\}$ denote the basis which diagonalizes $\sigma$,

$$\sigma = p|1\rangle\langle 1| + (1-p)|0\rangle\langle 0| , \tag{5.1}$$

with $0 \leq p \leq 1$. The likelihood ratio test only involves the diagonal part $\rho_D$ of $\rho$, which we can write as

$$\rho_D = q|1\rangle\langle 1| + (1-q)|0\rangle\langle 0|. \tag{5.2}$$

A basis of the Hilbert space for the $n$ copies is given by the states

$$|\mathbf{E}\rangle = |a_1 a_2 \dots a_n\rangle , \qquad a_i \in \{0,1\} , \tag{5.3}$$

labeled by the bit strings $a_1 a_2 \dots a_n$. The acceptance condition for the likelihood ratio test takes the form

$$|\mathbf{E}| + \frac{1}{n} \log \langle \mathbf{E}|\rho^{\otimes n}|\mathbf{E}\rangle \geq \mathcal{E} . \tag{5.4}$$

Denoting by $n(\mathbf{E})$ the number of 1s in $\mathbf{E}$ (the Hamming weight of the bit string), this is

$$n(\mathbf{E}) \geq n_* , \qquad n_* \equiv \left\lceil \frac{n}{\log\left(\frac{(1-p)q}{(1-q)p}\right)} \left(\mathcal{E} + \log\left(\frac{1-p}{1-q}\right)\right)\right\rceil , \tag{5.5}$$

where we use the ceiling function $\lceil \cdot \rceil$ so that $n_*$ is an integer. The optimal value for $\mathcal{E}$ is given in (4.31) in terms of the relative entropy and its variance

$$
\begin{aligned}
S(\rho_D \| \sigma) &= q(\log q - \log p) + (1-q)(\log(1-q) - \log(1-p)) , \\
V(\rho_D \| \sigma) &= q(1-q)\left(\log\left(\frac{q(1-p)}{p(1-q))}\right)\right)^2 ,
\end{aligned}
\tag{5.6}
$$

and leads to the acceptance threshold

$$n_* = \left\lceil nq + \text{sign}(q-p)\sqrt{\frac{q(1-q)}{n}} \, \Phi^{-1}(\varepsilon)\right\rceil . \tag{5.7}$$

The acceptance subspace is

$$\mathcal{H}_C = \underset{\mathbf{E}}{\text{span}} \, \{|\mathbf{E}\rangle \mid n(\mathbf{E}) \geq n_*\} , \tag{5.8}$$

and the measurement is the projection onto $\mathcal{H}_C$. We can also identify $\mathcal{H}_C$ with a subset of $\{0,1\}^n$, the complement of the Hamming sphere of radius $n_* - 1$ centered at the zero string.

### 5.1.2  Quantum circuit for the likelihood ratio test

We now describe a quantum circuit that implements the likelihood ratio test. In the language of quantum computing, our problem can be posed as follows. We are given a blackbox gate $V$ acting on a pair of qubits producing a state we wish to identify. More explicitly, acting with $V$ on $|00\rangle$ and tracing over the second qubit gives a density matrix $\rho_V$ for the first qubit, and we assume that there can be only two possibilities:

$$\rho_V = \rho \ \text{ or } \ \sigma , \tag{5.9}$$

where $\rho$ and $\sigma$ are known a priori but we do not know the outcome. Our goal is to determine which alternative is true by making a measurement on $n$ of these pairs of qubits, and operating only on the first qubit of each pair.

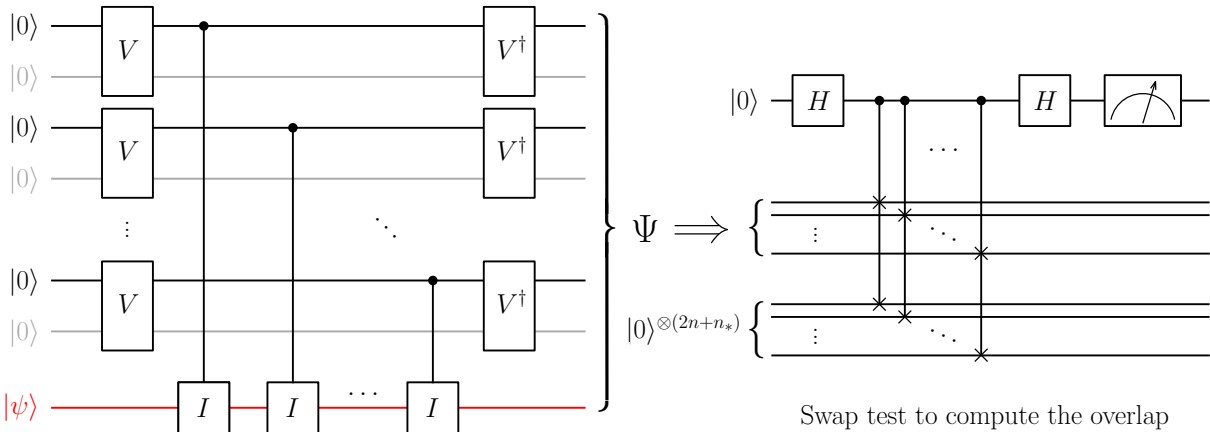

Circuit that prepares the state $\Psi$

Swap test to compute the overlap

Figure 5: Quantum circuit for the likelihood ratio test. The input consists of $n$ pairs of qubits in the state $|00\rangle$ and a register in the state $|\psi\rangle$. The role of the register is to count the number of 1s in the first qubits of each pair. This is done by using a controlled-$I$ gate, where $I$ is an increment operation. Such a gate is activated if and only if the control qubit (with a black dot) is in the state $|1\rangle$. This prepares a state $\Psi$ in the first part of the figure. The result of the likelihood ratio test is then obtained by measuring the overlap of the first $2n + n_*$ qubits of $\Psi$ with the state $|0\rangle^{\otimes(2n+n_*)}$. This can be done using a swap test, where we have an ancilla qubit with a series of controlled-SWAP gates, which swap two qubits (with two crosses) if and only if the ancilla qubit (with a black dot) is in the state $|1\rangle$, as represented in the second part of the figure.

The likelihood ratio test is the best classical measurement and becomes the optimal measurement when $\rho$ and $\sigma$ commute. From the previous analysis, the measurement is a projection $P_{\mathcal{H}_C}$ onto the acceptance subspace (5.8). Hence, we would like to compute

$$\operatorname{Tr} \rho_V^{\otimes n} P_{\mathcal{H}_C} . \tag{5.10}$$

If this quantity is close to one, we declare that $\rho_V = \rho$ while if it closer to zero, we declare that $\rho_V = \sigma$. Because the state $V|00\rangle$ is a purification of $\rho_V$, we can rewrite (5.10) as the overlap

$$\operatorname{Tr} \rho_V^{\otimes n} P_{\mathcal{H}_C} = \langle 0|^{\otimes 2n} (V^{\dagger})^{\otimes n} P_{\mathcal{H}_C} V^{\otimes n} |0\rangle^{\otimes 2n} , \tag{5.11}$$

where $P_{\mathcal{H}_C}$ only acts on the first qubit on each pair.

This quantity can be computed using the quantum circuit depicted in Figure 5. We start with $n$ pairs of qubits in the state $|0\rangle$ together with a register of $n$ auxiliary qubits in the state $|\psi\rangle$. We first act with $V$ on each pair. We then use a controlled-$I$ gate where $I$ is a "increment" gate which counts the number of 1s in the register while preserving the superposition.

The register is designed to incorporate the threshold condition associated with the projection $P_{\mathcal{H}_C}$ by measuring the overlap of some of its qubits with some fixed state. For example, we can take a register of $n + 1$ qubits and count the number of 1s as follows. We initialize the register in the state $|\psi\rangle = |1\rangle \otimes |0\rangle^{\otimes n}$ and define $I$ to be the cyclic permutation $i \mapsto i + 1$ on the $n + 1$ qubits. If the number of 1s is $k$, all the qubits in the register are in the state $|0\rangle$ except for a $|1\rangle$ in the $(k + 1)$-th position. Then, we can see that by measuring the overlap of the first $n_*$ qubits of the register with $|0\rangle^{\otimes n_*}$, we exactly implement the projection $P_{\mathcal{H}_C}$.[18] Indeed, all the states with $n(\mathbf{E}) \le n_* - 1$ are projected out. Measuring at the same time the overlap of the $n$

---

[18] We thank Michael Walter for this idea.

pairs of qubits with $|00\rangle^{\otimes n}$ precisely gives (5.11). The remaining qubits of the register should remain unmeasured.

This overlap operation should be implemented using a swap test between the $2n+n_*$ qubits consisting of our $n$ qubit pairs and the $n_*$ first qubit of the register, with $2n + n_*$ auxiliary qubits in the state $|0\rangle$. This allows us to measure the overlap (5.11) to arbitrary precision using iterations of the circuit. We note that the register can be optimized by using only $\log n$ qubits and storing the number of 1s in binary instead of unary.

## 5.2  Optimal measurement

We now investigate the optimal measurement for a qubit. When $\rho$ and $\sigma$ commute, the optimal measurement reduces to the likelihood ratio test, which was described in the previous section. Here, we would like to study the optimal measurement more generally, in a setup when $\rho$ and $\sigma$ do not commute. We consider a very simple non-commuting example, taking

$$\sigma = e^{-K}, \qquad \rho = e^{-\widetilde{K}} , \tag{5.12}$$

with

$$
\begin{aligned}
K &= E_0|0\rangle\langle 0| + E_1|1\rangle\langle 1| , \\
\widetilde{K} &= E_0|\widetilde{0}\rangle\langle\widetilde{0}| + E_1|\widetilde{1}\rangle\langle\widetilde{1}| ,
\end{aligned}
\tag{5.13}
$$

where $E_1 \geq E_0$. Moreover, we assume that the change of basis is just a rotation matrix

$$
\begin{aligned}
|\widetilde{0}\rangle &= \cos\theta\,|0\rangle - \sin\theta\,|1\rangle , \\
|\widetilde{1}\rangle &= \sin\theta\,|0\rangle + \cos\theta\,|1\rangle .
\end{aligned}
\tag{5.14}
$$

In a basis where $|0\rangle = \begin{pmatrix} 1 \\ 0 \end{pmatrix}$ and $|1\rangle = \begin{pmatrix} 0 \\ 1 \end{pmatrix}$, we have

$$
\sigma = \begin{pmatrix} e^{-E_0} & 0 \\ 0 & e^{-E_1} \end{pmatrix}, \qquad
\rho = \begin{pmatrix} e^{-E_0}\cos^2\theta + e^{-E_1}\sin^2\theta & (e^{-E_1} - e^{-E_0})\cos\theta\sin\theta \\ (e^{-E_1} - e^{-E_0})\cos\theta\sin\theta & e^{-E_0}\sin^2\theta + e^{-E_1}\cos^2\theta \end{pmatrix} ,
\tag{5.15}
$$

and we have $e^{-E_0} + e^{-E_1} = 1$ so it is useful to define $p$ such that

$$e^{-E_1} = p, \qquad e^{-E_0} = 1 - p , \tag{5.16}$$

and we have $p \leq \frac{1}{2}$. The relative entropy is

$$S(\rho\|\sigma) = (E_1 - E_0)\left(e^{-E_0} - e^{-E_1}\right)\sin^2\theta . \tag{5.17}$$

The basis states of $\sigma$ and $\rho$ are defined as bit strings

$$
\begin{aligned}
|\mathbf{E}\rangle &= |a_1 a_2 \ldots a_n\rangle , & a_i &\in \{0,1\} , \\
|\widetilde{\mathbf{E}}\rangle &= |\tilde{a}_1 \tilde{a}_2 \ldots \tilde{a}_n\rangle , & \tilde{a}_i &\in \{\widetilde{0}, \widetilde{1}\} .
\end{aligned}
\tag{5.18}
$$

We define $n(\mathbf{E})$ to be the number of 1s and $n(\widetilde{\mathbf{E}})$ to be the number of $\widetilde{1}$s. The acceptance condition with threshold $\mathcal{E}$ takes the simple form

$$n(\mathbf{E}) \geq n_*(\widetilde{\mathbf{E}}) , \qquad n_*(\widetilde{\mathbf{E}}) \equiv n(\widetilde{\mathbf{E}}) + \frac{n\mathcal{E}}{(E_1 - E_0)} . \tag{5.19}$$

This allows us to define the states that span the acceptance subspace. For every $\widetilde{\mathbf{E}}$, we define

$$|\xi(\widetilde{\mathbf{E}})\rangle = \sum_{\mathbf{E}:n(\mathbf{E})\geq n_*(\widetilde{\mathbf{E}})} \langle\mathbf{E}|\widetilde{\mathbf{E}}\rangle|\mathbf{E}\rangle . \tag{5.20}$$

The optimal measurement is then the projector to the acceptance subspace $\mathcal{H}_Q = \underset{\widetilde{\mathbf{E}}}{\mathrm{span}} \{|\xi(\widetilde{\mathbf{E}})\rangle\}$.

Formally, we first define the operator

$$Q = \sum_{\widetilde{\mathbf{E}}} |\xi(\widetilde{\mathbf{E}})\rangle\langle\widetilde{\mathbf{E}}| \tag{5.21}$$

so that the acceptance subspace $\mathcal{H}_Q$ is the image of $Q$. The optimal measurement is the projector onto it, given by

$$P_{\mathcal{H}_Q} = Q\frac{1}{Q^\dagger Q}Q^\dagger\,, \tag{5.22}$$

where $\boldsymbol{G} \equiv Q^\dagger Q$ is the Gram matrix of the vectors (5.20): the $2^n \times 2^n$ matrix of the overlaps $\langle\xi(\widetilde{\mathbf{E}}_1)|\xi(\widetilde{\mathbf{E}}_2)\rangle$. The above expression is well-defined because the restriction of $\boldsymbol{G}$ to the image of $Q^\dagger$ is invertible, and $P_{\mathcal{H}_Q}$ can be extended by zero on the vectors that are annihilated by $Q^\dagger$. Note that the above expression makes it clear that $P_{\mathcal{H}_Q}^2 = P_{\mathcal{H}_Q}$. We see that explicit construction of the projector involves finding the inverse of the Gram matrix $\boldsymbol{G}$, which is a challenging computational problem.

**Complexity of measurements.** It is intuitively clear that the optimal measurement is more complicated than the likelihood ratio test, since the former involves a more complicated construction of the acceptance space and the projector. It would be interesting to formalize this intuition by defining various notions of complexity of a measurement. The definitions of complexity could be based on different resources, and could also depend on the algorithm carrying out the measurement or computing the projector. A simple algorithm independent characteristic resource is the size of the acceptance subspace, or more precisely, its dimension. If one of the states to be compared is pure, the optimal measurement involves the projection to the state. In this simplest case, the acceptance space is smallest with just one state, while its complement is maximal. Hence, for comparing the complexity different measurements, it is helpful to define the minimum dimension of the acceptance space and its complement,

$$\dim \mathcal{H}_{\mathrm{acc}}^< \equiv \min\{\dim \mathcal{H}_{\mathrm{acc}}, \dim \mathcal{H} - \dim \mathcal{H}_{\mathrm{acc}}\}\,. \tag{5.23}$$

This defines a complexity measure which depends on the predetermined maximum size $\varepsilon$ of the type I error, the number $n$ of identical copies, and the two states $\rho, \sigma$ through the acceptance threshold $n_*$. Once these are given, we can compare the minimum acceptance dimension $\dim \mathcal{H}_{acc}^<$ of the optimal measurement and the likelihood ratio test. The latter depends on the volume of the Hamming sphere and its complement, so we have an analytical formula

$$\dim \mathcal{H}_{\mathrm{acc},C}^< = \min\left\{\sum_{k=0}^{n_*-1}\binom{n}{k}, \sum_{k=n_*}^{n}\binom{n}{k}\right\}\,. \tag{5.24}$$

For the optimal measurement, finding an analytical formula or at least an estimate for the minimum acceptance dimension $\dim \mathcal{H}_{\mathrm{acc},Q}^<$ is a mathematical challenge. We study it numerically for $n$ up to 14, by performing the Gram–Schmidt orthogonalization of the vectors $|\xi(\widetilde{\mathbf{E}})\rangle$ that span the acceptance space and then counting the number of orthonormal basis vectors. The (very limited) investigation suggests that $\dim \mathcal{H}_{\mathrm{acc},Q}^<$ grows exponentially with $n$ with a faster rate than $\dim \mathcal{H}_{\mathrm{acc},C}^<$.[19] This indicates that already at the level of the acceptance spaces the

---

[19]Such numerical observations need to be taken cautiously because the Gram-Schmidt algorithm is known to be unstable: small rounding errors can result in an imprecise estimate for the dimension of the spanned subspace [59, 60]. Understand this better would require a more systematic analysis, with a comparison of different orthogonalization algorithms.

optimal measurement is "more complex" than the likelihood ratio test. There are additional levels of complexity involved in computing the Gram matrix and finding its inverse, it would be interesting to develop rigorous complexity measures taking into account everything involved in constructing the projection.

**Numerical results.** The numerical implementation of the optimal measurement and the likelihood ratio test are done in a Mathematica notebook that we have made publicly available [61]. We analyze the numerical implementation of the measurements only up to $n = 14$, but this already proves sufficient to see some interesting features. For the threshold value $\mathcal{E}$, we use the optimal value (4.17). Including the second order term (in $n$) is necessary because $n$ is not very large (the second order term brings the $\varepsilon$-dependence). Choosing parameter values such that the finite $n$ effects are not too strong, we see that the optimal measurement is better by an order of magnitude. This is depicted in Figure 6. This demonstrates that quantum hypothesis testing is much more efficient than classical hypothesis testing. The tradeoff is that quantum hypothesis testing is more complex. The growth of the minimum acceptance dimension with $n$ is exponential for both measurements, but the growth rate appears to be faster for the optimal quantum measurement. It would be interesting to carry out a more extensive numerical investigation and see how generic this feature is.

**Some mathematical observations.** We finish this section by providing some partial results to the more challenging problem of constructing the optimal measurement in the general case. The partial results illustrate interesting connections to combinatorics and coding theory, which should inspire further study. For the rest of this discussion, we will restrict to the case $\theta = \frac{\pi}{4}$ where many simplifications occur. In this case, the rotation matrix (5.14) is just the Hadamard matrix and we have $|\widetilde{0}\rangle = |-\rangle$ and $|\widetilde{1}\rangle = |+\rangle$. In this case, we have a rather explicit description of the states $|\xi(\widetilde{\mathbf{E}})\rangle$:

$$|\xi(\widetilde{\mathbf{E}})\rangle = \frac{1}{2^{n/2}} \sum_{\substack{\mathbf{E} \\ n(\mathbf{E}) \geq n_*(\widetilde{\mathbf{E}})}} (-1)^{n_{0\widetilde{1}}(\mathbf{E},\widetilde{\mathbf{E}})} |\mathbf{E}\rangle \,, \tag{5.25}$$

where $n_{01}(\mathbf{E}, \widetilde{\mathbf{E}})$ is the number of pairs $(a_i, \tilde{a}_i)$ which are equal to $(0, \widetilde{1})$ using (5.18). We now need to do the Gram-Schmidt procedure for these vectors to obtain a basis of $\mathcal{H}_Q$. This requires to compute the Gram matrix $G$ of overlaps $\langle \xi(\widetilde{\mathbf{E}}_1)|\xi(\widetilde{\mathbf{E}}_2)\rangle$. The overlaps can be expressed as partial sums of products of binomial coefficients. Using a generalization of Vandermonde's identity, we can re-express the overlap as follows. Define the polynomial

$$P_n(x) = (1+x)^{n(\widetilde{\mathbf{E}}_1+\widetilde{\mathbf{E}}_2)}(1-x)^{n-n(\widetilde{\mathbf{E}}_1+\widetilde{\mathbf{E}}_2)} = \sum_{k=0}^{n} P_{n,k} x^k \,, \tag{5.26}$$

where $n(\widetilde{\mathbf{E}}_1 + \widetilde{\mathbf{E}}_2)$ is the number of 1s in the the boolean sum (*i.e.* the sum in the ring $\mathbb{Z}_2$) of $\widetilde{\mathbf{E}}_1$ and $\widetilde{\mathbf{E}}_2$. The overlap is then obtained as a partial sum of the coefficients $P_{n,k}$

$$\langle \xi(\widetilde{\mathbf{E}}_1)|\xi(\widetilde{\mathbf{E}}_2)\rangle \;\; = \;\; \frac{1}{2^n} \sum_{k=n-n_*(\widetilde{\mathbf{E}}_1,\widetilde{\mathbf{E}}_2)}^{n} P_{n,k} \,, \tag{5.27}$$

where $n_*(\widetilde{\mathbf{E}}_1, \widetilde{\mathbf{E}}_2) = \max(n_*(\widetilde{\mathbf{E}}_1), n_*(\widetilde{\mathbf{E}}_2))$. We refer to Appendix C for details on the derivation of this formula. There, it is also shown that $P_{n,k}$ are related to binary Krawtchouk polynomials $\mathcal{K}_k(x; n)$, and the overlaps of the Gram matrix take the explicit form

$$\langle \xi(\widetilde{\mathbf{E}}_1)|\xi(\widetilde{\mathbf{E}}_2)\rangle \;\; = \;\; \frac{1}{2^n} \sum_{k=n-n_*(\widetilde{\mathbf{E}}_1,\widetilde{\mathbf{E}}_2)}^{n} (-1)^k \mathcal{K}_k(n(\widetilde{\mathbf{E}}_1+\widetilde{\mathbf{E}}_2); n) \,. \tag{5.28}$$

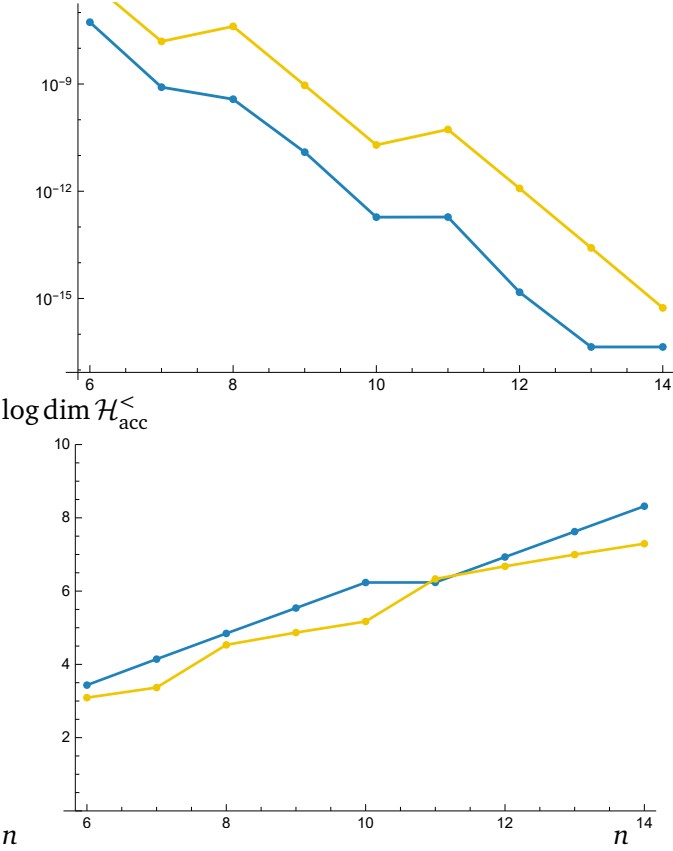

Figure 6: Optimal quantum measurement (blue) vs. optimal classical measurement (yellow). We see (left plot) that the optimal measurement gives a type II error $\beta$ that is one order of magnitude smaller for $n \sim 14$. We also see (right plot) that the minimum acceptance dimension is much larger for the optimal quantum measurement than for the optimal classical measurement. The curves with the logarithmic $y$-axis indicate exponential growth in $n$ with a faster rate for the optimal measurement. These plots are done with the parameters $\theta = \frac{\pi}{3}, p = 0.015, \varepsilon = 0.2$ using a Mathematica notebook that we have made publicly available [61].

It is also interesting that this problem seems related to coding theory and combinatorics. In Appendix C, we show that the Gram matrix is an element of the Terwilliger algebra [62] of the Hamming cube $H = \{0,1\}^n$ (see [63, 64]). This is done by identifying the labels $\xi(\widetilde{\mathbf{E}})$ as subsets of $H$, given by the supports of the bit strings $\widetilde{\mathbf{E}}$. In this way we obtain the explicit expansion

$$G = \sum_{i,j,t=0}^{n} x_{ij}^t M_{ij}^t , \qquad x_{ij}^t = \frac{1}{2^n} \sum_{k=n-\max(i,j)}^{n} (-1)^k \mathcal{K}_k(i+j-2t; n) , \qquad (5.29)$$

in the basis $\{M_{ij}^t\}$ of the Terwilliger algebra. Identifying the expansion coefficients $x_{ij}^t$ then allows at least a block diagonalization of $G$, exploiting the results of [63], which may turn out to be a useful step towards finding $G^{-1}$, and for the construction of the projector $P_{\mathcal{H}_Q}$.

# 6 Measurements in fermion chains

In this section, we study subsystem measurements in spinless fermion chains. Our goal is to construct measurements that are optimal in distinguishing between two different states, while acting only on a small subsystem. We will take these two states to be two thermal states with different temperatures. We will mostly focus on simpler hopping models, but some of our results also apply to fermion chains with Hamiltonians being arbitrary bilinears of creation and annihilation operators. This setup is the discrete analog of the chiral fermion CFT that will be studied in section 7.2.2. For small subsystem sizes, we will be able to give a more explicit description of the optimal measurement.

## 6.1 Spinless fermion chains

We consider spinless fermions on a chain of length $L \to +\infty$ with periodic boundary conditions.[20] The total Hamiltonian of the chain is

$$H = \sum_{1 \le i,j \le L} \left[ \psi_i^\dagger \hat{A}_{ij} \psi_j + \frac{1}{2} \left( \psi_i^\dagger \hat{B}_{ij} \psi_j^\dagger - \psi_i \hat{B}_{ij} \psi_j \right) \right], \tag{6.1}$$

and the fermion operators obey the anticommutation relations

$$\{\psi_i, \psi_j^\dagger\} = \delta_{ij}, \quad \{\psi_i, \psi_j\} = \{\psi_i^\dagger, \psi_j^\dagger\} = 0. \tag{6.2}$$

Here $\hat{A}$ is real symmetric and $\hat{B}$ is real antisymmetric to ensure Hermiticity. In addition, they are taken to be positive semi-definite so that the total energy is non-negative. The hats are used to denote $L \times L$ matrices supported on the whole chain, to be distinguished with matrices restricted to a subsystem that we study below.

As an example, the anisotropic XY model can be mapped to a Hamiltonian of the form (6.1) via a Jordan–Wigner transformation [65]. We will consider the simpler isotropic XY model in section 6.3.3 below.

### 6.1.1 Diagonalization of fermion Hamiltonians

The Hamiltonian (6.1) can be diagonalized by the Bogoliubov transformation

$$\eta_k = \sum_{i=1}^L \left( \frac{\hat{v}_{ki} + \hat{u}_{ki}}{2} \psi_i + \frac{\hat{v}_{ki} - \hat{u}_{ki}}{2} \psi_i^\dagger \right), \tag{6.3}$$

$$\eta_k^\dagger = \sum_{i=1}^L \left( \frac{\hat{v}_{ki} - \hat{u}_{ki}}{2} \psi_i + \frac{\hat{v}_{ki} + \hat{u}_{ki}}{2} \psi_i^\dagger \right), \tag{6.4}$$

where the vectors $\hat{v}_k, \hat{u}_k$ are solutions of the equations

$$\left( \hat{A} + \hat{B} \right) \hat{v}_k = \Lambda_k \hat{u}_k, \tag{6.5}$$

$$\left( \hat{A} - \hat{B} \right) \hat{u}_k = \Lambda_k \hat{v}_k. \tag{6.6}$$

Then, the Hamiltonian takes the form [65]

$$H = \sum_{1 \le k \le L} |\Lambda_k| \eta_k^\dagger \eta_k + \text{constant}, \tag{6.7}$$

---

[20]In what follows, there is a possibility of an order of limits issue with the thermodynamic $L \to \infty$ and the perturbative $\lambda \to 0$ limits. To circumvent the issue, we simply take $L$ to be larger than any scale in the problem and take the perturbative limit $\lambda \to 0$ while keeping $L$ fixed. We thank the referee for pointing out this subtlety.

where the constant sets the zero point energy.[21] The operators $\eta_k, \eta_k^\dagger$ generate a Fock space of positive energy excitations.

For fermion chains with $\hat{B} = 0$, the diagonalization procedure can be made more explicit. One first solves the eigenvalue problem $\hat{A}\hat{v}_k = \Lambda_k \hat{v}_k$ which allows to write

$$\hat{A} = \hat{v}^\intercal \hat{D} \hat{v}, \tag{6.8}$$

where $\hat{D}$ is a diagonal matrix with entries $\Lambda_k$. Then, performing the Bogoliubov transformation

$$a_k = \sum_k \hat{v}_{ki} \psi_i, \qquad a_k^\dagger = \sum_k \hat{v}_{ki} \psi_i^\dagger, \tag{6.9}$$

the Hamiltonian becomes

$$H = \sum_k \Lambda_k a_k^\dagger a_k, \tag{6.10}$$

where $\Lambda_k$ can be negative. The form (6.7) with absolute values is obtained by performing an additional particle-hole transformation on $a_k, a_k^\dagger$ (which is automatically included in (6.4)). For our purposes, the form (6.10) is sufficient and the Bogoliubov transformation (6.9) is a special case of (6.4) with $\hat{u} = \hat{v}$.

### 6.1.2 Reduced density matrix of a subsystem

We consider a subsystem $V = \{1, \ldots, \ell\}$ containing $\ell$ fermions, and place the chain (6.1) in a global thermal state[22]

$$\hat{\sigma} = \frac{e^{-\beta H}}{\operatorname{Tr} e^{-\beta H}}. \tag{6.11}$$

The reduced density matrix (RDM) on $V$ is obtained by tracing over its complement $V^c$ and takes the form

$$\sigma \equiv \operatorname{Tr}_{V^c} \hat{\sigma} = \frac{1}{Z} e^{-K}, \qquad Z \equiv \operatorname{Tr} e^{-K}, \tag{6.12}$$

where the modular Hamiltonian[23] $K$ takes the same form as total Hamiltonian of the chain:

$$K = \sum_{1 \leq i,j \leq \ell} \left[ \psi_i^\dagger A_{ij} \psi_j + \frac{1}{2} \left( \psi_i^\dagger B_{ij} \psi_j^\dagger - \psi_i B_{ij} \psi_j \right) \right]. \tag{6.13}$$

The matrices $A, B$ are different from the matrices $\hat{A}, \hat{B}$. Indeed, the modular Hamiltonian $K$, which depends on the global state, is not equal to the Hamiltonian $H|_V$ of the subsystem.

The matrices $A, B$ in the modular Hamiltonian can be obtained from the following equations [28, 66]

$$\operatorname{Tr}(\sigma \psi_i^\dagger \psi_j) = \operatorname{Tr}(\hat{\sigma} \psi_i^\dagger \psi_j), \qquad \operatorname{Tr}(\sigma \psi_i^\dagger \psi_j^\dagger) = \operatorname{Tr}(\hat{\sigma} \psi_i^\dagger \psi_j^\dagger), \qquad i, j \in V, \tag{6.14}$$

which follow from the fact that expectation values of operators supported in the subsystem can be computed using either the global state or the reduced state. The two-point functions are sufficient, because higher-order correlators reduce to two-point functions by Gaudin's theorem (an extension of Wick's theorem). Since both $\sigma$ and $\hat{\sigma}$ are exponentials of one-body operators, these traces can be computed explicitly (see Appendix D) to write the equations in terms of the parameters appearing in $K$ and $H$.

---

[21]The constant is explicitly $\frac{1}{2} \sum_k \left( \hat{A}_{kk} - |\Lambda_k| \right)$.

[22]We expect that a similar analysis could go through also for states that are exponentials of one-body operators, but we restrict our attention to thermal states.

[23]This is a slight abuse of language since the modular Hamiltonian is usually defined unnormalized, i.e. $\sigma = e^{-K}$, as in previous sections. Regardless, in this section, we define the modular Hamiltonian implicitly via (6.12).

For simplicity, we will restrict to free fermion chains with $\hat{B} = 0$, so that the Hamiltonian is

$$H = \sum_{1 \le i,j \le L} \psi_i^\dagger \hat{A}_{ij} \psi_j \,. \tag{6.15}$$

Due to the absence of the pair creation/annihilation terms, the anomalous two-point function $\mathrm{Tr}(\sigma \psi_i^\dagger \psi_j^\dagger) = 0$ vanishes. This is reflected in the modular Hamiltonian which has $B = 0$ [28]:

$$K = \sum_{i,j \in V} \psi_i^\dagger A_{ij} \psi_j \,. \tag{6.16}$$

The partition function $Z$ can now be easily obtained in terms of $A$ as

$$Z = \det(1 + e^{-A}) \,, \tag{6.17}$$

where the determinant is taken over the matrix indices.

Let $C$ denote the thermal two-point function restricted to the subsystem

$$C_{ij} \equiv \mathrm{Tr}(\hat{\sigma}\, \psi_i^\dagger \psi_j), \qquad i,j \in V \,, \tag{6.18}$$

determined by the Hamiltonian $H$. From the first equation in (6.14) it follows that [29]

$$A = \log\left(\frac{1-C}{C}\right), \tag{6.19}$$

from which we also obtain an expression for $Z$ in terms of $C$:

$$Z = \frac{1}{\det(1-C)} \,. \tag{6.20}$$

Hence for free fermions, the reduced density matrix of a subsystem in a thermal state is simply given by the thermal two-point function $C$.

### 6.1.3 Relative entropy and its variance for free fermions

We introduce a second global thermal state $\hat{\rho}$ with temperature $\tilde{\beta}$. This induces a different reduced density matrix $\rho$ on the subsystem:

$$\rho = \frac{1}{\tilde{Z}} e^{-\tilde{K}}, \qquad \tilde{K} = \sum_{i,j \in V} \psi_i^\dagger \tilde{A}_{ij} \psi_j \,. \tag{6.21}$$

Let us now compute the relative entropy and the relative entropy variance for the two reduced density matrices. Relative entropy is given by

$$S(\rho \| \sigma) = \langle K - \tilde{K} \rangle_\rho + \log \frac{Z}{\tilde{Z}} \,, \tag{6.22}$$

where we have

$$\langle K - \tilde{K} \rangle_\rho = \sum_{i,j \in V} (A_{ij} - \tilde{A}_{ij})\langle \psi_i^\dagger \psi_j \rangle_\rho = \mathrm{Tr}\left[(A - \tilde{A})\,\tilde{C}\right], \tag{6.23}$$

and we used

$$\tilde{C}_{ij} = \langle \psi_i^\dagger \psi_j \rangle_{\hat{\rho}} = \langle \psi_i^\dagger \psi_j \rangle_\rho, \qquad i,j \in V. \tag{6.24}$$

The partition functions are given by (6.20):

$$\log \frac{\tilde{Z}}{Z} = \log \det\left(\frac{1-C}{1-\tilde{C}}\right) = -\mathrm{Tr} \log\left(\frac{1-\tilde{C}}{1-C}\right). \tag{6.25}$$

As a result, we obtain for the relative entropy

$$S(\rho\|\sigma) = \text{Tr}\left[(A-\widetilde{A})\widetilde{C} + \log\left(\frac{1-\widetilde{C}}{1-C}\right)\right], \tag{6.26}$$

and the relative entropy variance is given by

$$V(\rho\|\sigma) = \langle\Delta K^2\rangle_\rho - \langle\Delta K\rangle_\rho^2, \tag{6.27}$$

which doesn't depend on the partition functions. The first term can be written as

$$\langle\Delta K^2\rangle_\rho = \sum_{i,j,k,l\in V} \Delta A_{ij}\Delta A_{kl}\langle\psi_i^\dagger\psi_j\psi_k^\dagger\psi_l\rangle_\rho, \tag{6.28}$$

where $\Delta A_{ij} = A_{ij} - \widetilde{A}_{ij}$. Because $\rho$ is an exponential of one-body operators, we can use Gaudin's theorem to compute the four-point function [67] (see also Appendix D). The result is

$$\langle\psi_i^\dagger\psi_j\psi_k^\dagger\psi_l\rangle_\rho = \langle\psi_i^\dagger\psi_j\rangle_\rho\langle\psi_k^\dagger\psi_l\rangle_\rho + \langle\psi_i^\dagger\psi_l\rangle_\rho\langle\psi_j\psi_k^\dagger\rangle_\rho \tag{6.29}$$

$$= \widetilde{C}_{ij}\widetilde{C}_{kl} - \widetilde{C}_{il}(\delta_{jk} - \widetilde{C}_{kj}), \tag{6.30}$$

and we get

$$\langle\Delta K^2\rangle_\rho = \sum_{i,j\in V}\Delta A_{ij}\widetilde{C}_{ij}\sum_{k,l\in V}\Delta A_{kl}\widetilde{C}_{kl} + \sum_{i,j,k,l\in V}\Delta A_{ij}\Delta A_{kl}\widetilde{C}_{il}(\delta_{jk} - \widetilde{C}_{kj}) \tag{6.31}$$

$$= \text{Tr}[\Delta A\widetilde{C}]^2 + \text{Tr}[\Delta A^2\widetilde{C}(1-\widetilde{C})]. \tag{6.32}$$

The first term equals $\langle\Delta K\rangle_\rho^2$ which cancels in (6.27) and leaves us with

$$V(\rho\|\sigma) = \text{Tr}[(A-\widetilde{A})^2\widetilde{C}(1-\widetilde{C})]. \tag{6.33}$$

As far as the authors are aware, the expressions (6.26) and (6.33) for relative entropy and its variance have not appeared in the literature before. However, sandwiched Rényi relative entropy between RDMs of a free fermion chain was computed in [35] (see also [36]) and one can check that the relative entropy (6.26) matches with the first derivative of their expression. Unfortunately, we did not manage to compute the second derivative to see whether the result matches with the variance. As an independent consistency check of (6.33), we will see below that it obeys the lower bound (3.20).

The expressions for $S(\rho\|\sigma)$ and $V(\rho\|\sigma)$ can be written explicitly in terms of eigenvalues and eigenvectors of $A, \widetilde{A}$. We have

$$A_{ij} = \sum_{k\in V}E_k v_{ki}v_{kj}, \qquad \widetilde{A}_{ij} = \sum_{k\in V}\widetilde{E}_k\widetilde{v}_{ki}\widetilde{v}_{kj}, \tag{6.34}$$

so that

$$S(\rho\|\sigma) = \sum_k\left[\sum_l\frac{E_k}{1+e^{\widetilde{E}_l}}(v_k\cdot\widetilde{v}_l)^2 - \frac{\widetilde{E}_k}{1+e^{\widetilde{E}_k}} + \log\left(\frac{1+e^{-E_k}}{1+e^{-\widetilde{E}_k}}\right)\right], \tag{6.35}$$

where $v_k\cdot\widetilde{v}_l = \sum_i v_{ki}\widetilde{v}_{li}$ is the overlap between the eigenvectors. There is also a similar expression for the variance.

A further simplification occurs if $A$ and $\widetilde{A}$ commute so that their eigenvectors are the same:

$$v_i\cdot\widetilde{v}_j = \delta_{ij}. \tag{6.36}$$

In this case, one obtains simple expressions

$$S(\rho\|\sigma) = \sum_k \left[ \frac{E_k - \widetilde{E}_k}{1 + e^{\widetilde{E}_k}} + \log\left( \frac{1 + e^{-E_k}}{1 + e^{-\widetilde{E}_k}} \right) \right], \qquad V(\rho\|\sigma) = \frac{1}{4} \sum_k \frac{(\widetilde{E}_k - E_k)^2}{\cosh^2\left( \frac{\widetilde{E}_k}{2} \right)} . \tag{6.37}$$

The vanishing of the commutator of $A, \widetilde{A}$ is equivalent to commutativity of the RDMs $[\rho, \sigma] = 0$. This can be seen by performing Bogoliubov transformations

$$\psi_i = \sum_{k \in V} v_{ki} c_k, \qquad \psi_i^\dagger = \sum_{k \in V} v_{ki} c_k^\dagger ,$$
$$\psi_i = \sum_{k \in V} \widetilde{v}_{ki} \widetilde{c}_k, \qquad \psi_i^\dagger = \sum_{k \in V} \widetilde{v}_{ki} \widetilde{c}_k^\dagger , \tag{6.38}$$

on $K$ and $\widetilde{K}$ respectively. In a similar way the full Hamiltonian was diagonalized using (6.9), the modular Hamiltonians become

$$K = \sum_{k \in V} E_k c_k^\dagger c_k, \qquad \widetilde{K} = \sum_{k \in V} \widetilde{E}_k \widetilde{c}_k^\dagger \widetilde{c}_k . \tag{6.39}$$

If (6.36) holds one finds from (6.38) that $c_k = \widetilde{c}_k$ and $c_k^\dagger = \widetilde{c}_k^\dagger$ so that $[K, \widetilde{K}] = 0$. In addition, one can check that for a perturbative entanglement spectrum of the form $\widetilde{E}_k = E_k + \lambda E_k^{(1)}$, the expressions (6.37) saturate the lower bound (3.20), as expected for commuting RDMs.

## 6.2 Optimal measurement

In this section, we describe the implementation of the optimal measurement for spinless fermion chains. This involves computing overlaps between eigenstates of two modular Hamiltonians, which can be done using the generalized dick's theorem [38, 39]. For free fermions, this gives a prescription on how the overlaps $v_i \cdot \widetilde{v}_j$ between eigenvectors translate into overlaps between eigenstates $\langle E_I | \widetilde{E}_J \rangle$. For completeness, we will consider general modular Hamiltonians of the form (6.13) with non-trivial $A$ and $B$. We will restrict to modular Hamiltonians of free fermions with $B = 0$ in the end.

### 6.2.1 Eigenstates of modular Hamiltonians and their overlaps

To unify the computations, we introduce some convenient notation. Let

$$\psi = (\psi_1, \dots, \psi_\ell)^\mathsf{T}, \qquad \psi^\dagger = (\psi_1^\dagger, \dots, \psi_\ell^\dagger)^\mathsf{T} , \tag{6.40}$$

be $\ell$-dimensional vectors. We define similarly the $\ell$-dimensional vectors $c, c^\dagger$ and $\widetilde{c}, \widetilde{c}^\dagger$, and combine them further into $2\ell$-dimensional vectors as

$$\Psi = \begin{pmatrix} \psi \\ \psi^\dagger \end{pmatrix}, \qquad \alpha = \begin{pmatrix} c \\ c^\dagger \end{pmatrix}, \qquad \widetilde{\alpha} = \begin{pmatrix} \widetilde{c} \\ \widetilde{c}^\dagger \end{pmatrix} . \tag{6.41}$$

Following the analysis for the Hamiltonian of the chain, modular Hamiltonians $K, \widetilde{K}$ of the form (6.13) are diagonalized by transformations

$$\alpha = W\Psi, \qquad \widetilde{\alpha} = \widetilde{W}\Psi , \tag{6.42}$$

where

$$W = \frac{1}{2} \begin{pmatrix} v + u & v - u \\ v - u & v + u \end{pmatrix}, \qquad \widetilde{W} = \frac{1}{2} \begin{pmatrix} \widetilde{v} + \widetilde{u} & \widetilde{v} - \widetilde{u} \\ \widetilde{v} - \widetilde{u} & \widetilde{v} + \widetilde{u} \end{pmatrix}. \tag{6.43}$$

The transformation matrices are obtained by solving equation (6.6) for $A$ and $B$ (and similarly for $\widetilde{v}, \widetilde{u}$):

$$(A+B)v_k = E_k u_k \, , \tag{6.44}$$

$$(A-B)u_k = E_k v_k \, . \tag{6.45}$$

The matrices $v, u, \widetilde{v}, \widetilde{u}$ are real and orthogonal so that $W, \widetilde{W}$ real and orthogonal as well.[24] They are thus Bogoliubov transformations, because the real Bogoliubov group is the orthogonal group (see Appendix D.1).

As a result, the modular Hamiltonians become

$$K = \sum_{k \in V} |E_k| c_k^\dagger c_k + E_{\text{vac}}, \qquad \widetilde{K} = \sum_{k \in V} |\widetilde{E}_k| \widetilde{c}_k^\dagger \widetilde{c}_k + \widetilde{E}_{\text{vac}} \, . \tag{6.46}$$

The exact values of $E_{\text{vac}}, \widetilde{E}_{\text{vac}}$ are not important for the upcoming analysis. From these expressions it follows that eigenstates are generated by acting on two quasi-particle vacua $|E_{\text{vac}}\rangle, |\widetilde{E}_{\text{vac}}\rangle$ with creation operators. The vacua are defined via

$$c_k |E_{\text{vac}}\rangle = 0, \qquad \widetilde{c}_k |\widetilde{E}_{\text{vac}}\rangle = 0, \qquad \text{for all } k \in V \, , \tag{6.47}$$

and the eigenstates are

$$|E_{i_1 \ldots i_n}\rangle = c_{i_1}^\dagger c_{i_2}^\dagger \cdots c_{i_n}^\dagger |E_{\text{vac}}\rangle = |a_1 a_2 \cdots a_\ell\rangle \, , \tag{6.48}$$

$$|\widetilde{E}_{i_1 \ldots i_n}\rangle = \widetilde{c}_{i_1}^\dagger \widetilde{c}_{i_2}^\dagger \cdots \widetilde{c}_{i_n}^\dagger |\widetilde{E}_{\text{vac}}\rangle = |\widetilde{a}_1 \widetilde{a}_2 \cdots \widetilde{a}_\ell\rangle \, ,$$

where we used $\ell$-bit binary strings to keep track of the occupation numbers of the modes $k$. The corresponding eigenvalues are

$$E_{i_1 \ldots i_n} = E_{\text{vac}} + |E_{i_1}| + \ldots + |E_{i_n}|, \quad \widetilde{E}_{i_1 \ldots i_n} = \widetilde{E}_{\text{vac}} + |\widetilde{E}_{i_1}| + \ldots + |\widetilde{E}_{i_n}|. \tag{6.49}$$

These eigenvalues are invariant under permutations of $\{i_1, \ldots, i_n\}$ so we assume that the indices in (6.48) are in an increasing sequence $i_1 < i_2 < \ldots < i_n$. This choice removes some additional sign factors in formulas below.

We want to compute overlaps between these eigenstates

$$\langle E_{i_1 \ldots i_n} | \widetilde{E}_{j_1 \ldots j_m}\rangle = \langle E_{\text{vac}} | c_{i_n} \cdots c_{i_1} \widetilde{c}_{j_1}^\dagger \cdots \widetilde{c}_{j_m}^\dagger |E_{\text{vac}}\rangle. \tag{6.50}$$

Standard Wick's theorem does not directly apply to correlators of this type because $\widetilde{c}_i^\dagger$ is not the Hermitian conjugate of $c_i$. The trick is to realize that the operators $\alpha$ and $\widetilde{\alpha}$ are related via a Bogoliubov transformation $T$ (orthogonal matrix):

$$\widetilde{\alpha} = T\alpha \, , \tag{6.51}$$

which is explicitly

$$T = \widetilde{W} W^{\mathsf{T}} = \frac{1}{2} \begin{pmatrix} \widetilde{v}v^{\mathsf{T}} + \widetilde{u}u^{\mathsf{T}} & \widetilde{v}v^{\mathsf{T}} - \widetilde{u}u^{\mathsf{T}} \\ \widetilde{v}v^{\mathsf{T}} - \widetilde{u}u^{\mathsf{T}} & \widetilde{v}v^{\mathsf{T}} + \widetilde{u}u^{\mathsf{T}} \end{pmatrix}. \tag{6.52}$$

We introduce the operator $\mathcal{T}$ that implements the Bogoliubov transformation $T$ in the Hilbert space [38, 39]:

$$\widetilde{\alpha} = \mathcal{T} \alpha \mathcal{T}^{-1} = T\alpha \tag{6.53}$$

---

[24]Reality of for example $v$ follows from the fact that it obeys $(A + B)(A - B)v_k = E_k^2 v_k$ where $(A+B)(A-B) = (A+B)(A+B)^{\mathsf{T}}$ is real and symmetric.

and we have that $\mathcal{T}$ is unitary since $T$ is real. The expression for $\mathcal{T}$ in terms of $\alpha$ is not relevant in what follows. However, if $T$ can be written as an exponential $T = e^{-\Omega S}$, where $\Omega$ is the matrix (D.3) and $S$ is antisymmetric, then $\mathcal{T}$ is an exponential of one-body operators [38, 39].

It follows that $|\widetilde{E}_{\text{vac}}\rangle = \mathcal{T} |E_{\text{vac}}\rangle$ so that all the eigenstates of the modular Hamiltonians are related according to

$$|\widetilde{E}_{i_1 \ldots i_n}\rangle = \mathcal{T} |E_{i_1 \ldots i_n}\rangle . \tag{6.54}$$

The overlaps (6.50) are therefore

$$\langle E_{i_1 \ldots i_n} | \widetilde{E}_{j_1 \ldots j_m}\rangle = \langle E_{\text{vac}} | c_{i_n} \cdots c_{i_1} \mathcal{T} c_{j_1}^\dagger \cdots c_{j_m}^\dagger |E_{\text{vac}}\rangle , \tag{6.55}$$

and unitarity of $\mathcal{T}$ ensures that these overlaps determine a unitary basis rotation in the Hilbert space.

All the operators in (6.55) are expressed in terms of the annihilation and creation operators $c, c^\dagger$ which allows the use of Wick's theorem. In Appendix D, we show that the overlaps involving two operators are

$$\frac{\langle E_i | \widetilde{E}_j \rangle}{\langle E_{\text{vac}} | \widetilde{E}_{\text{vac}} \rangle} = (T_{11}^{-1})_{ij}, \quad \frac{\langle E_{ij} | \widetilde{E}_{\text{vac}} \rangle}{\langle E_{\text{vac}} | \widetilde{E}_{\text{vac}} \rangle} = (T_{11}^{-1} T_{12})_{ij}, \quad \frac{\langle E_{\text{vac}} | \widetilde{E}_{ij} \rangle}{\langle E_{\text{vac}} | \widetilde{E}_{\text{vac}} \rangle} = (T_{12} T_{11}^{-1})_{ij} , \tag{6.56}$$

where $T_{11} = T_{22}, T_{12} = T_{21}$ are the two $\ell \times \ell$ blocks of (6.52) and the overlap between the vacua is

$$\langle E_{\text{vac}} | \widetilde{E}_{\text{vac}} \rangle = (\det T_{11})^{1/2} . \tag{6.57}$$

The overlaps (6.55) involving more operators can be computed using generalized Wick's theorem [39] and it is non-zero only when $n + m = 2t$ is even. In that case:

$$\frac{\langle E_{i_1 \ldots i_n} | \widetilde{E}_{i_{n+1} \ldots i_{2t}} \rangle}{\langle E_{\text{vac}} | \widetilde{E}_{\text{vac}} \rangle} = \sum_{\text{pairings}} (-1)^P \prod_{\text{pairs}} (\text{contraction of a pair}) , \tag{6.58}$$

where the sum is over pairings $\{i_1, \ldots, i_{2t}\} \rightarrow \{(i_{j_1}, i_{k_1}), \ldots, (i_{j_t}, i_{k_t})\}$ and $P$ is the signature of the permutation $i_n \ldots i_1 i_{n+1} \ldots i_{2t} \rightarrow i_{j_1} i_{k_1} \ldots i_{j_t} i_{k_t}$ involved in the pairing. The contractions appearing on the right hand side are the three two-point overlaps (6.56) and we refer to Appendix D for more details. In other words, all the overlaps (6.55) can be expressed in terms of the two-point overlaps (6.56) using the generalized Wick's theorem.

The computation of the contractions (6.56) requires the knowledge of $v, u$ and $\widetilde{v}, \widetilde{u}$ that determine the block matrices $T_{ij}$ according to (6.52). These can be computed from (6.45) knowing $A, B$ and $\widetilde{A}, \widetilde{B}$ which are obtained from two-point functions in the global state according to (6.14). Although these equations are in general difficult to solve, they become simpler for free fermions, because $B$ vanishes and $A$ is directly given in terms of $C$ according to (6.19). We will demonstrate this below for the XY model.

The power of this approach is that it gives a way to compute the overlaps without the need of the explicit form of the ground states $|E_{\text{vac}}\rangle, |\widetilde{E}_{\text{vac}}\rangle$. It can therefore be applied to modular Hamiltonians of the general form (6.13). However, there is one situation where the above computation of the overlaps fails: when $\det T_{11} = 0$ so that $T_{11}$ is not invertible. This happens when the two quasi-particle vacua are orthogonal.

### 6.2.2 Overlaps of eigenstates for free fermions

The above algorithm to compute overlaps simplifies for free fermions since $B = \widetilde{B} = 0$ which implies that we can use the Bogoliubov transformations (6.43) with $u = v$ and $\widetilde{u} = \widetilde{v}$. Hence all the overlaps are determined by the eigenvectors $v, \widetilde{v}$ of the two-point functions $C, \widetilde{C}$.

With $B = \widetilde{B} = 0$, the modular Hamiltonians are

$$K = \sum_{i,j \in V} \psi_i^\dagger A_{ij} \psi_j, \qquad \widetilde{K} = \sum_{i,j \in V} \psi_i^\dagger \widetilde{A}_{ij} \psi_j \,. \tag{6.59}$$

As shown before, they take the diagonal form (6.39) after the transformation (6.38):

$$W = \begin{pmatrix} v & 0 \\ 0 & v \end{pmatrix}, \qquad \widetilde{W} = \begin{pmatrix} \widetilde{v} & 0 \\ 0 & \widetilde{v} \end{pmatrix} \,. \tag{6.60}$$

From these we get

$$T = \begin{pmatrix} \widetilde{v}v^\intercal & 0 \\ 0 & \widetilde{v}v^\intercal \end{pmatrix}, \tag{6.61}$$

which is block diagonal. The overlap between the quasi-particle vacua is then

$$\langle E_{\text{vac}} | \widetilde{E}_{\text{vac}} \rangle = (\det \widetilde{v}v^\intercal)^{1/2} = 1 \,, \tag{6.62}$$

where we used the fact that the determinant of $\widetilde{v}v^\intercal \in SO(2\ell)$ is unity. In this case, the quasi-particle vacua coincide with the true vacuum $|E_{\text{vac}}\rangle = |\widetilde{E}_{\text{vac}}\rangle = |0\rangle$ (annihilated by $\psi_i$).

Noting that $(T_{11})^{-1} = v\widetilde{v}^\intercal$, the only non-zero contractions are

$$\langle E_i | \widetilde{E}_j \rangle = (v\widetilde{v}^\intercal)_{ij} = v_i \cdot \widetilde{v}_j \,. \tag{6.63}$$

Because of this, the higher order overlaps (6.55) are non-zero if and only if $n = m$. The generalized Wick's theorem (6.58) for $t = n$ gives

$$\langle E_{i_1 \dots i_n} | \widetilde{E}_{j_1 \dots j_n} \rangle = \sum_{p \in S_n} \text{sgn}(p) \langle E_{i_1} | \widetilde{E}_{j_{p(1)}} \rangle \cdots \langle E_{i_n} | \widetilde{E}_{j_{p(n)}} \rangle \,, \tag{6.64}$$

with the sum over permutations $p$ of $n$ elements. Writing $I = \{i_1, \dots, i_n\}, J = \{j_1, \dots, j_m\}$, the overlaps can be written compactly as a matrix minor [25]

$$\langle E_I | \widetilde{E}_J \rangle = \begin{cases} \det\limits_{I,J} v\widetilde{v}^\intercal & n = m \\ 0 & n \neq m \end{cases} \,. \tag{6.65}$$

The result (6.65) could have been obtained directly from the correlator (6.50) without reference to the generalized Wick's theorem. For example, inverting (6.38) yields

$$\langle E_i | \widetilde{E}_j \rangle = \langle 0 | c_i \widetilde{c}_j^\dagger | 0 \rangle = \sum_{k,k' \in V} v_{ik} \widetilde{v}_{jk'} \langle 0 | \psi_k \psi_{k'}^\dagger | 0 \rangle = \sum_{k \in V} v_{ik} \widetilde{v}_{jk'} \delta_{kk'} = v_i \cdot \widetilde{v}_j \,, \tag{6.66}$$

with a similar strategy for the higher order correlators. It is for modular Hamiltonians with $B \neq 0$ when the generalized Wick's theorem becomes very useful.

## 6.3 Examples

We now give explicit examples for the general procedure described above.

---

[25]Minor $I, J$ is the determinant of the $n \times n$ submatrix formed of elements $i, j$ with $i \in I, j \in J$.

### 6.3.1 A single fermion subsystem

The simplest possible subsystem contains only a single fermion. For a generic quadratic modular Hamiltonian (6.13) with $\ell = 1$, the matrix $B$ does not contribute as it is antisymmetric. Hence modular Hamiltonians of a single fermion at site $k = 1$ take the form

$$K = E\,\psi_1^\dagger \psi_1, \qquad \widetilde{K} = \widetilde{E}\,\psi_1^\dagger \psi_1 \,. \tag{6.67}$$

The two-dimensional Hilbert space of the fermion is spanned by the vacuum state $|0\rangle$ and the state

$$|1\rangle \equiv \psi_1^\dagger |0\rangle \,, \tag{6.68}$$

with a fermion occupying site $k = 1$. In the above formalism, they are eigenstates of the modular Hamiltonians since we have $T = \mathbf{1}_{2\times2}$.

The fermion Hilbert space spanned by $|0\rangle, |1\rangle$ is equivalent to the single qubit Hilbert space studied in section 5. The two RDMs of the fermion take the form

$$\rho = (1-q)|0\rangle\langle0| + q|1\rangle\langle1|, \qquad \sigma = (1-p)|0\rangle\langle0| + p|1\rangle\langle1| \,, \tag{6.69}$$

with

$$q = \frac{1}{1+e^{\widetilde{E}}}, \qquad p = \frac{1}{1+e^{E}} \,. \tag{6.70}$$

We see that the RDMs always commute. As a result, the optimal measurement is given by the likelihood ratio test described in section 4.3. The acceptance subspace for the RDMs (6.69) was determined in section 5. Relative entropy and its variance are given by (6.37) and the acceptance condition becomes

$$n(\mathbf{E}) \geq n_* \equiv \left\lceil \frac{n}{1+e^{\widetilde{E}}} + \frac{1}{\sqrt{n}}\,\frac{\mathrm{sgn}\!\left(e^{E}-e^{\widetilde{E}}\right)}{2\cosh\!\left(\frac{\widetilde{E}}{2}\right)}\,\Phi^{-1}(\varepsilon) \right\rceil \,, \tag{6.71}$$

where $n(\mathbf{E})$ is the number of fermions in the $n$ copies of the subsystem. The optimal measurement is then a projection onto states that contain $n_*$ or more fermions.

### 6.3.2 Two fermion subsystem

The situation is more interesting for subsystems containing more fermions. We consider here a subsystem of two fermions in a free fermion chain, taking the two fermions to be on sites $i = 1, 2$. The matrices $A, \widetilde{A}$ have two eigenvalues $E_{1,2}, \widetilde{E}_{1,2}$ and eigenvectors which we parametrize as

$$v = \begin{pmatrix} \cos\varphi & -\sin\varphi \\ \sin\varphi & \cos\varphi \end{pmatrix}, \qquad \widetilde{v} = \begin{pmatrix} \cos\widetilde{\varphi} & -\sin\widetilde{\varphi} \\ \sin\widetilde{\varphi} & \cos\widetilde{\varphi} \end{pmatrix}. \tag{6.72}$$

Using the binary string notation for the eigenstates, we have

$$|E_{\mathrm{vac}}\rangle = |00\rangle \,, \quad |E_1\rangle = |10\rangle \,, \quad |E_2\rangle = |01\rangle \,, \quad |E_{12}\rangle = |11\rangle \,, \tag{6.73}$$

$$|\widetilde{E}_{\mathrm{vac}}\rangle = |\widetilde{0}\widetilde{0}\rangle \,, \quad |\widetilde{E}_1\rangle = |\widetilde{1}\widetilde{0}\rangle \,, \quad |\widetilde{E}_2\rangle = |\widetilde{0}\widetilde{1}\rangle \,, \quad |\widetilde{E}_{12}\rangle = |\widetilde{1}\widetilde{1}\rangle \,. \tag{6.74}$$

There is a total of sixteen overlaps. From (6.65), the non-zero overlaps are

$$\langle E_{\mathrm{vac}}|\widetilde{E}_{\mathrm{vac}}\rangle = 1, \qquad \langle E_{12}|\widetilde{E}_{12}\rangle = 1 \,, \tag{6.75}$$

and

$$\langle E_i|\widetilde{E}_j\rangle = \begin{pmatrix} \cos(\varphi-\widetilde{\varphi}) & -\sin(\varphi-\widetilde{\varphi}) \\ \sin(\varphi-\widetilde{\varphi}) & \cos(\varphi-\widetilde{\varphi}) \end{pmatrix}. \tag{6.76}$$

Thus the unitary rotation

$$|\widetilde{E}_I\rangle = \sum_J U_{IJ}|E_J\rangle \,, \tag{6.77}$$

is given by

$$U = \begin{pmatrix} 1 & 0 & 0 & 0 \\ 0 & \cos(\varphi - \tilde{\varphi}) & -\sin(\varphi - \tilde{\varphi}) & 0 \\ 0 & \sin(\varphi - \tilde{\varphi}) & \cos(\varphi - \tilde{\varphi}) & 0 \\ 0 & 0 & 0 & 1 \end{pmatrix} \,, \tag{6.78}$$

and it acts non-trivially only on the subspace spanned by $|E_1\rangle, |E_2\rangle$.

The basis rotation (6.78) is effectively the same as the one studied in section 5 where the optimal measurement on a single qubit is constructed. The eigenstates $|E_1\rangle$ and $|E_2\rangle$, with a single fermion on either site 1 or 2, correspond to the rotation between two states of a qubit. In addition, we also have an unrotated qubit. As discussed in section 5.2, the explicit description of the optimal measurement for the one-qubit case is challenging due to the difficult inversion of the Gram matrix. We will thus describe the suboptimal but simpler likelihood ratio test.

Assuming for simplicity that the two eigenvalues of $K$ are equal $E_1 = E_2 \equiv E_{\text{vac}} + \Delta, E_{12} = E_{\text{vac}} + 2\Delta$ with $\Delta > 0$, and likewise for the tilded values, we have

$$-\log\rho = E_0|00\rangle\langle00| + \Delta(|10\rangle\langle10| + |01\rangle\langle01|) + (E_0 + 2\Delta)|11\rangle\langle11| \,, \tag{6.79}$$
$$-\log\widetilde{\rho} = \widetilde{E}_0|\widetilde{0}\widetilde{0}\rangle\langle\widetilde{0}\widetilde{0}| + \widetilde{\Delta}(|\widetilde{1}\widetilde{0}\rangle\langle\widetilde{1}\widetilde{0}| + |\widetilde{0}\widetilde{1}\rangle\langle\widetilde{0}\widetilde{1}|) + (\widetilde{E}_0 + 2\widetilde{\Delta})|\widetilde{1}\widetilde{1}\rangle\langle\widetilde{1}\widetilde{1}| \,,$$

where $E_0 \equiv E_{\text{vac}} + \log Z$ and a similar definition of $\widetilde{E}_0$. The eigenstates of $\rho^{\otimes n}, \sigma^{\otimes n}$ are

$$|\mathbf{E}\rangle \equiv |a_1 a_2 \cdots a_{2n-1} a_{2n}\rangle \,, \tag{6.80}$$
$$|\widetilde{\mathbf{E}}\rangle \equiv |\widetilde{a}_1 \widetilde{a}_2 \cdots \widetilde{a}_{2n-1} \widetilde{a}_{2n}\rangle \,,$$

labelled by $2n$-bit strings. The average modular energies are

$$|\mathbf{E}| = E_0 + \frac{n(\mathbf{E})}{n}\Delta \,, \tag{6.81}$$
$$|\widetilde{\mathbf{E}}| = \widetilde{E}_0 + \frac{n(\widetilde{\mathbf{E}})}{n}\widetilde{\Delta} \,,$$

where $n(\mathbf{E}), n(\widetilde{\mathbf{E}})$ count the number of 1s in the binary strings. The acceptance condition $|\mathbf{E}| - |\widetilde{\mathbf{E}}| \geq \mathcal{E}$ becomes

$$n(\mathbf{E}) \geq n_* \equiv \left\lceil n(\widetilde{\mathbf{E}})\frac{\widetilde{\Delta}}{\Delta} + \frac{(\widetilde{E}_0 - E_0)n}{\Delta} + \frac{n\mathcal{E}}{\Delta} \right\rceil \,. \tag{6.82}$$

The likelihood ratio test is then the projector

$$P_{\mathcal{H}_C} = \sum_{n(\mathbf{E}) \geq n_*} |\mathbf{E}\rangle\langle\mathbf{E}| \,. \tag{6.83}$$

Note that $\widetilde{E}_0 - E_0$ cancels in (6.82) with the same term coming from relative entropy once the threshold $\mathcal{E} = S(\rho_D\|\sigma) + \dots$ is substituted. It's also possible to obtain an explicit expression for $S(\rho_D\|\sigma)$ using the overlaps (6.76).

The acceptance space is given by (the complement of) the Hamming sphere of radius $n_*$ centered at zero in the Hamming cube $\{0,1\}^{2n}$. While the likelihood ratio test is in general a suboptimal measurement, it becomes optimal when the reduced density matrices commute. The next example gives a situation where this happens.

### 6.3.3 Example: XY model at finite temperature

The isotropic XY spin chain has the Hamiltonian [65]

$$H = \sum_{i=1}^{L}(\sigma_i^x \sigma_{i+1}^x + \sigma_i^y \sigma_{i+1}^y) \,, \tag{6.84}$$

where $\sigma_i^{x,y}$ is the Pauli matrix at site $i$ and the boundary conditions are periodic. In the thermodynamic limit, this Hamiltonian can be mapped to a periodic free fermion chain [65][26]

$$H = \frac{1}{2}\sum_{i=1}^{L-1}(\psi_i^\dagger \psi_{i+1} + \psi_{i+1}^\dagger \psi_i) = \sum_{i,j} \psi_i^\dagger \hat{A}_{ij} \psi_j \,, \tag{6.85}$$

where

$$\hat{A}_{ij} = \frac{1}{2}\left(\delta_{i,j+1} + \delta_{i+1,j}\right) \,. \tag{6.86}$$

Hence the Hamiltonian is of the form (6.1) with $\hat{B}_{ij} = 0$ and the eigenvectors $\hat{v}_k$ and eigenvalues $\Lambda_k$ can be found in [65]. Due to translation invariance, the thermal two-point function is a function of $i-j$ only, and in the thermodynamic limit $L \to \infty$, it takes the form

$$C_{ij} = \frac{1}{\pi}\int_0^\pi dq\, \frac{\cos[q(i-j)]}{e^{\beta \cos q}+1} \,. \tag{6.87}$$

Consider now two fermions at sites $i=1$ and $i=1+r$ where $r$ is a positive integer. Then the two-point function has the form

$$C = \begin{pmatrix} a & b \\ b & a \end{pmatrix} \,, \tag{6.88}$$

where $a$ and $b$ are obtained from (6.87). The eigenvectors of $C$ are

$$v = \frac{1}{\sqrt{2}}\begin{pmatrix} 1 & -1 \\ 1 & 1 \end{pmatrix} \,, \tag{6.89}$$

which corresponds to $\varphi = \pi/4$ in equation (6.72). We see that $v$ is independent of the temperature $\beta$ and of the distance $r$. This is true in any translation invariant fermion chain for which the thermal two-point function is of the form (6.88).

Now when considering thermal states of two different temperatures, leading to two modular Hamiltonians $\widetilde{K}$ and $K$, the unitary rotation (6.78) between their eigenstates is trivial: $U_{IJ} = \delta_{IJ}$. Hence the RDMs of the two fermions commute and the optimal measurement is the likelihood ratio test. If the fermion chain is not translation invariant this is no longer true, because then the modular Hamiltonians $K, \widetilde{K}$ do not generally commute. It is interesting that translation invariance implies commutativity of two-fermion density matrices in global thermal states.

## 7 Measurements in conformal field theory

We now turn to the implementation of quantum hypothesis testing in quantum field theory. We will discuss in detail how the measurements described in section 4 are realized as operators

---

[26]Strictly speaking, the Jordan–Wigner transformation also produces an additional interaction term between $\psi_1$ and $\psi_L$ in the periodic fermion chain (6.85). However, the interaction produces contributions to $\Lambda_k$ and $\hat{v}_k$ that are subleading in the thermodynamic limit $L \to \infty$ [65]. Hence we neglect these extra contributions and focus on the periodic fermion chain (6.85) with translation symmetry.

acting on states. For simplicity, we restrict to two-dimensional conformal field theory because the infinite-dimensional group of conformal transformations in two dimensions allows for a certain flexibility. For an introduction to the subject, we refer to [68].

The physical system we consider will live on a line or on a circle. We will be particularly interested in distinguishing two different states from an interval subregion. Our main technical result is the construction of the optimal measurements for special types of states, studied by Cardy and Tonni [40]. As an illustration, we study the free chiral fermion CFT, which could be viewed as a continuous limit of the discrete fermion chain studied in section 6.

While we obtain some basic technical results in implementing measurements in conformal field theories, we are merely scratching the surface of a vast number of possibilities in the choices of theories and states. As our free fermion case will show, there are interesting analytical challenges when trying to simplify the implementation of efficient measurements.

## 7.1 Subregion measurements

We now describe the situation where we want to distinguish between two states in a $\text{CFT}_2$ while only having access to a subregion. After tracing out over the rest of the system, the two states are given by two density matrices $\sigma$ and $\rho$ supported in that subregion.

The measurements described in section 4 are given in terms of the modular Hamiltonians. In general, the modular Hamiltonian of a reduced density matrix would be a complicated non-local operator and be difficult to study. For a special class of states in a $\text{CFT}_2$, the modular Hamiltonian is local: it can be written as a suitable integral of the stress tensor. We will restrict to these types of states in the following two sections, drawing on the results of [40]. We will first describe the optimal measurement in the generic situation, and then explore in some more detail the task of distinguishing between two thermal states at different temperatures in the next section. We will explain how to implement the likelihood ratio test to distinguish between the vacuum and a primary excitation.

### 7.1.1 Setup

Let's now describe the setup. The $\text{CFT}_2$ is defined on a line or on a circle and the subregion we consider is an interval $I = [-\frac{\ell}{2}, \frac{\ell}{2}]$. We consider the Euclidean spacetime described by a coordinate $z$. We cut out little disks of size $\epsilon$ around the endpoints of $I$ to regulate the entanglement entropy. The boundary conditions are given by two boundary states $|a\rangle$ and $|b\rangle$ and they contribute a finite amount to the entanglement entropy via Affleck–Ludwig boundary entropies [40].

We consider two reduced density matrices $\sigma$ and $\rho$ defined on the interval $I$. The corresponding modular Hamiltonians $K = -\log\sigma$ and $\widetilde{K} = -\log\rho$ are assumed to be local. As a result, each of them can be viewed as generating a flow along a vector field, as represented on the left of Figure 7. To define the optimal measurement, we are interested in the eigenstates of both $K$ and $\widetilde{K}$, and their overlaps. To obtain a useful description of these states, we will use the flexibility of two-dimensional CFTs to conformally transform the setup to a simpler geometry for each state, as represented on the right of Figure 7. In this simpler geometry, the modular Hamiltonian becomes a dilatation operator, whose eigenstates are easily described.

We first use the conformal map

$$z \longmapsto w = f(z), \tag{7.1}$$

which takes the spacetime to an annulus of width $W$.[27] More precisely, the interval is mapped to $w \in [-\frac{W}{2}, \frac{W}{2}]$, and the imaginary part of $w$ is periodic with period $2\pi$. The modular Hamil-

---

[27]Not to be confused with the notation $W$ for the Bogoliubov transformation in section 6.2.

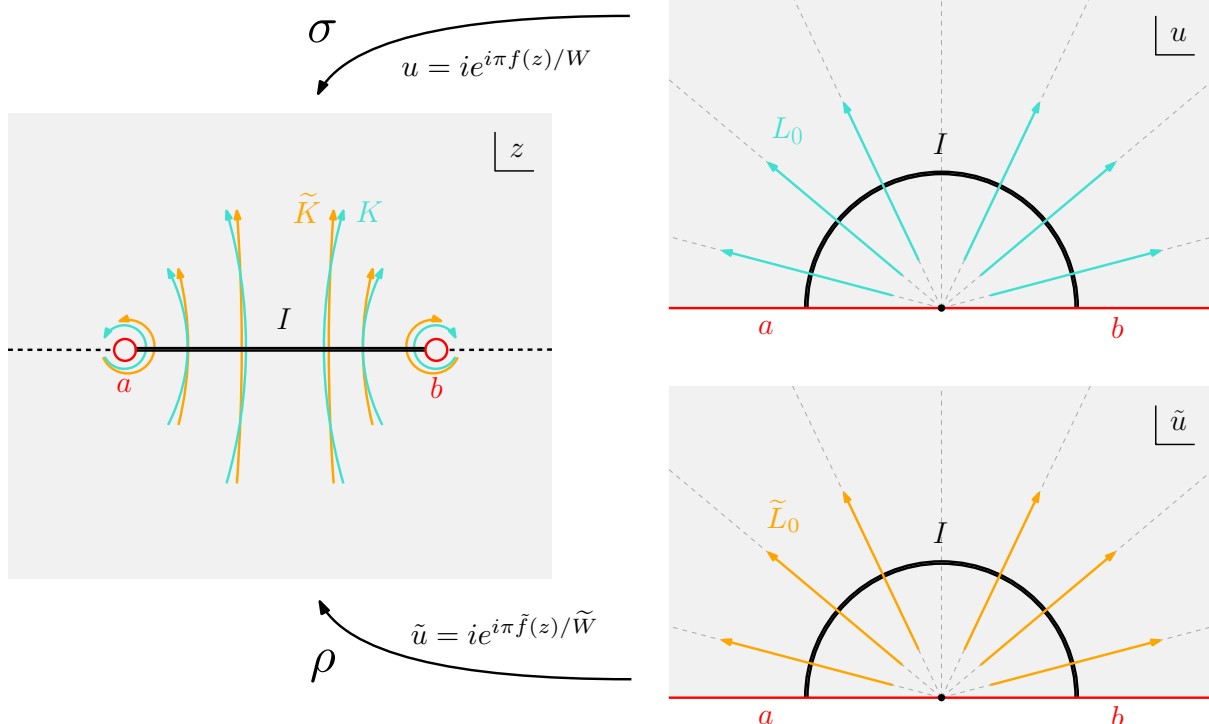

Figure 7: The modular Hamiltonians $K = -\log \sigma$ and $\widetilde{K} = -\log \rho$ are conformally mapped to dilatation operators in the upper half plane. The entanglement spectrum is then obtained using radial quantization. The inverse maps give expressions for $K$ and $\widetilde{K}$ in the original spacetime, giving a way to compute the overlaps of their eigenstates, as required to implement the optimal measurement. The modular flows are depicted in blue for $K$ and in orange for $\widetilde{K}$.[28]

tonian in these new variables becomes simple: it just generates translations in the imaginary $w$ direction.

To describe the eigenstates of the modular Hamiltonian, it is useful to consider the universal cover by allowing the imaginary part of $w$ to be unconstrained. The geometry becomes an infinite strip. We can then map it to the upper half plane with

$$w \longmapsto u = ie^{i\pi w/W} . \tag{7.2}$$

The interval becomes a half unit circle $C_+$, ranging from $u = 1$ to $u = -1$. As explained in [41], the choice of boundary conditions is such that one can extend this to the other half plane and perform radial quantization on the full plane. The modular Hamiltonian $K$ is simply related to the generator $L_0$ of dilatations in this geometry:

$$K = \frac{2\pi^2}{W}\left(L_0 - \frac{c}{24}\right) + \frac{cW}{12} , \tag{7.3}$$

where $c$ is the central charge and the additive constant ensures that $\operatorname{Tr} e^{-K} = 1$ [40]. We refer to [69] for a more detailed discussion of this setup. The upshot of all these manipulations is that we can now relate the spectrum of the modular Hamiltonian to the spectrum of $L_0$ in the presence of two boundary conditions $|a\rangle$ and $|b\rangle$. For example, we can choose $|a\rangle = |b\rangle = |\bar{0}\rangle$ where the Cardy state $|\bar{0}\rangle$ projects onto the vacuum sector of the theory [41], so that the only states in the entanglement spectrum are the vacuum and its descendants.

In the $u$-plane, we obtain from radial quantization the Virasoro generators

$$L_n = \frac{1}{2\pi i} \oint_C du\, u^{n+1} T(u) = \frac{1}{2\pi i} \int_{C_+} du\, u^{n+1} T(u) - \frac{1}{2\pi i} \int_{C_+} d\bar{u}\, \bar{u}^{n+1} \overline{T}(\bar{u})\,, \qquad (7.4)$$

where $C$ is the unit circle. This is then translated to an integral over the original interval $I$:

$$L_n = -\frac{W}{2\pi^2} \int_I dz\, i^n e^{in\pi f(z)/W} \frac{T(z)}{f'(z)} + \text{h.c.} \qquad (7.5)$$

The entanglement spectrum of a state $\sigma$ can then be generated by acting with these operators on the vacuum.

We can use the same procedure for another state $\rho$ using a different map $w = \tilde{f}(z)$ giving an annulus of width $\widetilde{W}$. The spectrum of $\rho$ is then generated by another Virasoro algebra

$$\widetilde{L}_n = -\frac{\widetilde{W}}{2\pi^2} \int_I dz\, i^n e^{in\pi \tilde{f}(z)/\widetilde{W}} \frac{T(z)}{\tilde{f}'(z)} + \text{h.c.} \qquad (7.6)$$

Similarly, the modular Hamiltonian $\widetilde{K} = -\log \rho$ is them given by

$$\widetilde{K} = \frac{2\pi^2}{\widetilde{W}}\left(\widetilde{L}_0 - \frac{c}{24}\right) + \frac{c\widetilde{W}}{12}\,. \qquad (7.7)$$

Since both Virasoro algebras are written on the interval, we can compare them. Their commutators can be computed using the general commutation relation of two stress tensors in a $\text{CFT}_2$:

$$-i[T(z), T(z')] = (T(z) + T(z'))\partial_z \delta(z - z') - \frac{3c}{12\pi}\partial_z^3 \delta(z - z')\,. \qquad (7.8)$$

We can restrict to the vacuum sector by choosing the boundary condition $|a\rangle = |b\rangle = |\bar{0}\rangle$. Then, the eigenstates of $K$ are given by the eigenstates of $L_0$ which takes the form

$$|\Delta\rangle = L_{-1}^{\ell_1} L_{-2}^{\ell_2} \dots |0\rangle\,. \qquad (7.9)$$

Similarly the eigenstates of $\widetilde{K}$ at the eigenstates of $\widetilde{L}_0$ and take the form

$$|\widetilde{\Delta}\rangle = \widetilde{L}_{-1}^{\tilde{\ell}_1} \widetilde{L}_{-2}^{\tilde{\ell}_2} \dots |0\rangle\,. \qquad (7.10)$$

The general commutation relation (7.8) can be used to compute the commutators $[L_n, \widetilde{L}_m]$, even though this is difficult in practice. This then gives a way to compute the overlaps $\langle\Delta|\widetilde{\Delta}\rangle$, as required to describe the optimal measurement.

### 7.1.2 Optimal measurement

The optimal measurement can then be implemented in this language, following section 4.2. Let's now consider $n$ copies of the system. The eigenstates of $\sigma^{\otimes n}$ and $\rho^{\otimes n}$ are respectively denoted

$$|\mathbf{\Delta}\rangle = |\Delta_1\rangle \otimes |\Delta_2\rangle \otimes \dots \otimes |\Delta_n\rangle\,, \qquad |\widetilde{\mathbf{\Delta}}\rangle = |\widetilde{\Delta}_1\rangle \otimes |\widetilde{\Delta}_2\rangle \otimes \dots \otimes |\widetilde{\Delta}_n\rangle\,. \qquad (7.11)$$

---

[28]The picture makes it look like that the Euclidean modular flows both live in the same Euclidean spacetime, which is not generally true. It is their Lorentzian versions, which define the operators $K$ and $\widetilde{K}$, which both live in the original spacetime.

Using the formula (7.3), we see that the average modular energies for $K$ and $\widetilde{K}$ are respectively

$$K^{(n)} = \frac{2\pi^2}{W}\left(|\mathbf{\Delta}| - \frac{c}{24}\right) + \frac{cW}{12}, \qquad \widetilde{K}^{(n)} = \frac{2\pi^2}{\widetilde{W}}\left(|\widetilde{\mathbf{\Delta}}| - \frac{c}{24}\right) + \frac{c\widetilde{W}}{12}, \qquad (7.12)$$

where the average conformal dimension is denoted

$$|\mathbf{\Delta}| = \frac{1}{n}\sum_{i=1}^{n}\Delta_i, \qquad |\widetilde{\mathbf{\Delta}}| = \frac{1}{n}\sum_{i=1}^{n}\widetilde{\Delta}_i . \qquad (7.13)$$

The optimal measurement is then described by first decomposing $|\widetilde{\mathbf{\Delta}}\rangle$ in the $\{|\mathbf{\Delta}\rangle\}$ basis

$$|\widetilde{\mathbf{\Delta}}\rangle = \sum_{\mathbf{\Delta}}\langle\mathbf{\Delta}|\widetilde{\mathbf{\Delta}}\rangle|\mathbf{\Delta}\rangle , \qquad (7.14)$$

where we have $\langle\mathbf{\Delta}|\widetilde{\mathbf{\Delta}}\rangle = \prod_{i=1}^{n}\langle\Delta_i|\widetilde{\Delta}_i\rangle$. We then restrict the sum over $\mathbf{\Delta}$ to those satisfying the acceptance condition $K^{(n)} - \widetilde{K}^{(n)} \geq \mathcal{E}$ which is here:

$$\frac{2\pi^2}{W}\left(|\mathbf{\Delta}| - \frac{c}{24}\right) - \frac{2\pi^2}{\widetilde{W}}\left(|\widetilde{\mathbf{\Delta}}| - \frac{c}{24}\right) + \frac{c}{12}(W - \widetilde{W}) \geq \mathcal{E} . \qquad (7.15)$$

This allows us to define the states

$$|\xi(\widetilde{\mathbf{\Delta}})\rangle \equiv \sum_{\mathbf{\Delta}\,:\,|\mathbf{E}| - |\widetilde{\mathbf{E}}| \geq \mathcal{E}}\langle\mathbf{\Delta}|\widetilde{\mathbf{\Delta}}\rangle|\mathbf{\Delta}\rangle . \qquad (7.16)$$

The optimal measurement is the projector onto the subspace

$$\mathcal{H}_Q = \operatorname*{span}_{\widetilde{\mathbf{\Delta}}}\{|\xi(\widetilde{\mathbf{\Delta}})\rangle\} , \qquad (7.17)$$

with the choice of acceptance threshold being

$$\mathcal{E} = S(\rho\|\sigma) + \sqrt{\frac{V(\rho\|\sigma)}{n}}\Phi^{-1}(\varepsilon) . \qquad (7.18)$$

### 7.1.3 Likelihood ratio test

We will see that the optimal measurement is difficult to describe explicitly. A simpler measurement, which is suboptimal but still performs well, is the likelihood ratio test discussed in section 4.3. The measurement projects on part of the spectrum of $\sigma^{\otimes n}$. More precisely, it is a projection on the acceptance subspace

$$\mathcal{H}_C = \operatorname{span}\left\{|\mathbf{\Delta}\rangle \ \middle| \ \frac{1}{n}\sum_{i=1}^{n}\langle\Delta_i|K - \widetilde{K}|\Delta_i\rangle \geq \mathcal{E}\right\} , \qquad (7.19)$$

and the best value of $\mathcal{E}$ is given in (4.31). We can rewrite the acceptance condition as

$$\frac{2\pi^2}{W}\left(|\mathbf{\Delta}| - \frac{c}{24}\right) - \frac{2\pi^2}{\widetilde{W}}\left(\widetilde{L}_0(\mathbf{\Delta}) - \frac{c}{24}\right) + \frac{c}{12}(W - \widetilde{W}) \geq \mathcal{E} , \qquad (7.20)$$

where we define the averages

$$|\mathbf{\Delta}| \equiv \frac{1}{n}\sum_{i=1}^{n}\Delta_i, \qquad \widetilde{L}_0(\mathbf{\Delta}) \equiv \frac{1}{n}\sum_{i=1}^{n}\langle\Delta_i|\widetilde{L}_0|\Delta_i\rangle . \qquad (7.21)$$

To obtain a more explicit description, we should compute $\langle\Delta|\widetilde{L}_0|\Delta\rangle$, which can be written

$$\langle\Delta|\widetilde{L}_0|\Delta\rangle = \sum_{\widetilde{\Delta}}\widetilde{\Delta}|\langle\Delta|\widetilde{\Delta}\rangle|^2 . \qquad (7.22)$$

As a result, a fairly explicit description of this measurement can be given with only the knowledge of the overlaps $\langle\Delta|\widetilde{\Delta}\rangle$.

## 7.2 Thermal states

As a concrete example of the procedure described above, we can consider the problem of distinguishing two thermal states of different temperatures, having only access to a subregion. We take the subregion to be an interval $I = [-\frac{\ell}{2}, \frac{\ell}{2}]$ in the infinite line. Following [70], the reduced density matrix obtained from a thermal state is associated to the conformal mapping

$$f_\beta(z) = \log\left( \frac{e^{2\pi z/\beta} - e^{-\pi\ell/\beta}}{e^{\pi\ell/\beta} - e^{2\pi z/\beta}} \right), \tag{7.23}$$

which allows to obtain the corresponding modular Hamiltonian, as described in section 7.1.1.

We consider two reduced density matrices $\sigma$ and $\rho$ in the interval $I$ obtain from global thermal states of inverse temperature $\beta_1$ and $\beta_2$. The corresponding modular Hamiltonians are explicitly

$$K \equiv -\log\sigma = 2\beta_1 \int_{-\ell/2}^{\ell/2} dx\, \frac{\sinh\left[\frac{\pi}{\beta_1}\left(\frac{\ell}{2} - x\right)\right]\sinh\left[\frac{\pi}{\beta_1}\left(\frac{\ell}{2} + x\right)\right]}{\sinh\left(\frac{\pi\ell}{\beta_1}\right)} T_{00}(x) + c(\beta_1) \tag{7.24}$$

$$\widetilde{K} \equiv -\log\rho = 2\beta_2 \int_{-\ell/2}^{\ell/2} dx\, \frac{\sinh\left[\frac{\pi}{\beta_2}\left(\frac{\ell}{2} - x\right)\right]\sinh\left[\frac{\pi}{\beta_2}\left(\frac{\ell}{2} + x\right)\right]}{\sinh\left(\frac{\pi\ell}{\beta_2}\right)} T_{00}(x) + c(\beta_2), \tag{7.25}$$

where $T_{00}$ is the energy density of the CFT and $c(\beta_{1,2})$ are normalization constants.

### 7.2.1 Entropy and variance

In a thermal state at temperature $\beta$, the one-point function of the energy density is $\langle T_{00} \rangle = \frac{\pi c}{6\beta^2}$. We can determine the constant $c(\beta)$ in (7.24), because we know that the entanglement entropy is

$$S(\beta) = \frac{c}{3}\log\left( \frac{\beta}{\pi\epsilon}\sinh(\pi\ell/\beta) \right) + g_a + g_b , \tag{7.26}$$

where $\epsilon$ is the UV cut-off and $g_a, g_b$ are the Affleck–Ludwig boundary entropies originating from boundary conditions at the entangling points [40]. This allows us to compute the relative entropy

$$S(\rho\|\sigma) = \frac{c}{6}\left(1 - \frac{\beta_1^2}{\beta_2^2}\right)\left(1 - \frac{\pi\ell}{\beta_1}\coth\left(\frac{\pi\ell}{\beta_1}\right)\right) + \frac{c}{3}\log\left( \frac{\beta_1\sinh(\pi\ell/\beta_1)}{\beta_2\sinh(\pi\ell/\beta_2)} \right). \tag{7.27}$$

The variance can be computed directly from the formulas (7.24) and the two-point function

$$\langle T_{00}(x) T_{00}(y) \rangle = \frac{c}{4\pi^2}\frac{1}{(x-y)^4} . \tag{7.28}$$

At leading order in the small interval limit $\ell/\beta_1 \to 0$, we have

$$S(\rho\|\sigma) = \frac{c\pi^4}{540}\left(1 - \frac{\beta_1}{\beta_2}\right)^2\left(\frac{\ell}{\beta_1}\right)^4 + \mathcal{O}\left(\frac{\ell}{\beta_1}\right)^6 \tag{7.29}$$

$$V(\rho\|\sigma) = \frac{c\pi^4}{162}\left(1 - \frac{\beta_1}{\beta_2}\right)^2\left(\frac{\ell}{\beta_1}\right)^4 + \mathcal{O}\left(\frac{\ell}{\beta_1}\right)^6 .$$

We note that we have the ratio

$$\lim_{\ell/\beta_1 \to 0} \frac{V(\rho\|\sigma)}{S(\rho\|\sigma)} = \frac{10}{3} , \tag{7.30}$$

satisfying the lower bound (3.20).[29] It turns out that this ratio is an interesting quantity to study for more general states, and further results on this ratio will be presented elsewhere.

---

[29]The lower bound was proven only for finite dimensional Hilbert spaces so it is interesting to see that it also holds in a field theory example.

### 7.2.2 Free fermion

The description of the optimal measurement in section 7.1 is valid for a general $CFT_2$. We can try to be a bit more explicit by considering the example of the free fermion in two dimensions. This theory can be seen as a continuum analog of the fermion chain considered in the previous section. The free boson is very similar and presented in Appendix E.

We consider a free fermion $\psi$ on a circle with antiperiodic boundary conditions (Neveu-Schwarz sector). It has a mode decomposition

$$\psi(u) = \sum_{n \in \mathbb{Z}+\frac{1}{2}} \psi_n u^{-n-1/2} \, . \tag{7.31}$$

As above, we can compute the Fourier mode

$$\psi_n = \frac{i^n}{2\sqrt{i\pi W}} \int_I dz \sqrt{f'(z)} \, e^{i\pi n f(z)/W} \psi(z) + \text{h.c.} \, , \tag{7.32}$$

where we are using the notation

$$f(z) = f_{\beta_1}(z), \qquad \widetilde{f}(z) = f_{\beta_2}(z) \, . \tag{7.33}$$

The anticommutation relation of the field is

$$\{\psi(z), \psi(z')\} = \{\overline{\psi}(z), \overline{\psi}(z')\} = 2\pi i \, \delta(z-z'), \qquad \{\psi(z), \overline{\psi}(z')\} = 0 \, . \tag{7.34}$$

This implies that for the Fourier modes, we have

$$\{\psi_n, \psi_m\} = \frac{i^{n+m}}{2W} \int_I dz \, f'(z) e^{i\pi(m+n)f(z)/W} + \text{h.c.} \, , \tag{7.35}$$

from which one can show that $\{\psi_n, \psi_m\} = \delta_{m+n}$. For the state $\rho$, we have similarly

$$\widetilde{\psi}_n = \frac{i^n}{2\sqrt{i\pi\widetilde{W}}} \int_I dz \sqrt{\widetilde{f}'(z)} \, e^{i\pi n \widetilde{f}(z)/\widetilde{W}} \psi(z) + \text{h.c.} \tag{7.36}$$

We would like to compute overlaps between the eigenstates of $\rho$ and that of $\sigma$. This information is contained in the commutator

$$\{\psi_n, \widetilde{\psi}_{-m}\} = A_{nm} \, , \tag{7.37}$$

where

$$A_{nm} = \frac{i^{n+m}}{2\sqrt{W\widetilde{W}}} \int_I dz \sqrt{\widetilde{f}'(z)f'(z)} \, e^{inf(z)/W + im\widetilde{f}(z)/\widetilde{W}} + \text{h.c.} \tag{7.38}$$

Although explicit, this integral is hard to compute analytically.

The Hilbert space is a Fock space generated by acting on the vacuum with creation operators. A basis adapted to $\sigma$ is given by

$$|\Delta_{\mathbf{s}}\rangle = \psi_{-s_1} \psi_{-s_2} \ldots |0\rangle \, , \tag{7.39}$$

where $\mathbf{s} = (s_k)_k$ with $s_k \in \mathbb{Z} + \frac{1}{2}$ and $s_k > 0$, which we take to be in an increasing sequence. The conformal dimension (eigenvalue of $L_0$) of such a state is

$$L_0|\Delta_{\mathbf{s}}\rangle = \Delta_{\mathbf{s}}|\Delta_{\mathbf{s}}\rangle, \qquad \Delta_{\mathbf{s}} = \sum_k s_k \, . \tag{7.40}$$

Similarly, we can consider a basis adapted to $\rho$ given by the states

$$|\widetilde{\Delta}_{\tilde{\mathbf{s}}}\rangle = \widetilde{\psi}_{-\tilde{s}_1}\widetilde{\psi}_{-\tilde{s}_2}\ldots|0\rangle\,, \tag{7.41}$$

where $\tilde{\mathbf{s}} = (\tilde{s}_k)_k$ being an increasing sequence.

To describe the optimal measurement, we would like to compute the overlap $\langle\Delta_{\mathbf{s}}|\widetilde{\Delta}_{\tilde{\mathbf{s}}}\rangle$. We see that the overlap is non-zero if and only if $|\mathbf{s}| = |\tilde{\mathbf{s}}|$ where $|\cdot|$ denotes the cardinality of the set $s$. Moreover, we see that the overlap is simply given by the corresponding minor of the matrix $A$

$$\langle\Delta_{\mathbf{s}}|\widetilde{\Delta}_{\tilde{\mathbf{s}}}\rangle = \det_{\mathbf{s}\tilde{\mathbf{s}}} A \equiv M_{\mathbf{s}\tilde{\mathbf{s}}}\,, \tag{7.42}$$

which defines a matrix $M$. The eigenvalue $E$ of $K$ is related to that of $L_0$ via the relation (7.3).

We now consider $n$ copies of the system to implement the optimal measurement. Following section 7.1.2, we have the acceptance condition (7.15). This allows us to define the states $|\xi(\widetilde{\Delta})\rangle$ using the overlaps computed above. The optimal measurement is then the projector onto the subspace (7.17) spanned by these states. It is difficult to obtain a more explicit description of this optimal measurement. The first obstacle is the computation of the integral (7.38) which is needed to obtain the states $|\xi(\widetilde{\Delta})\rangle$ more explicitly. Furthermore, even if we managed to have a simple expression for these states, describing the subspace (7.17) will be even harder, involving their orthonormalization using for example the Gram–Schmidt process. This procedure was discussed in section 5.2 in the much simpler case of a qubit, where it already leads to a challenging combinatorial problem.

It is then of interest to find suboptimal but simpler measurements which still perform well. A good candidate is the likelihood ratio test discussed in section 4.3 in a general context. Following section 7.1.3, implementing this measurement in CFT only requires the computation of the one-point function $\langle\Delta_{\mathbf{s}}|\widetilde{L}_0|\Delta_{\mathbf{s}}\rangle$. For the free fermion, it can be written as

$$\langle\Delta_{\mathbf{s}}|\widetilde{L}_0|\Delta_{\mathbf{s}}\rangle = \sum_{\tilde{\mathbf{s}}}\widetilde{\Delta}_{\tilde{\mathbf{s}}}|\langle\Delta_{\mathbf{s}}|\widetilde{\Delta}_{\tilde{\mathbf{s}}}\rangle|^2 = \sum_{\tilde{\mathbf{s}}}(\Sigma_k\tilde{s}_k)\,|M_{\mathbf{s}\tilde{\mathbf{s}}}|^2\,. \tag{7.43}$$

This only requires the computation of $A_{nm}$ and its minors, which is much more tractable, as compared to what is required to describe explicitly the optimal measurement.

## 7.3 Primary excitation

We now consider a setup consisting of a primary excitation that we wish to distinguish from the vacuum. We are interested in the case where we have only access to a subregion of the system. We will take the example of an interval in the circle. Let $\sigma$ and $\rho$ be the states on this interval corresponding respectively to the vacuum and to the excitation.[30] Considering $n$ copies of this setup, we would like to distinguish between the two states

$$\sigma^{\otimes n} \quad \text{and} \quad \rho^{\otimes n}\,. \tag{7.44}$$

The optimal measurement is more difficult to describe because in this case, we do not have an analytic expression for the modular Hamiltonian of the excitation. Nonetheless, we will be able to implement the likelihood ratio test, as discussed in section 7.1.3.

Consider a two-dimensional CFT on a circle with circumference $L$ at zero temperature. The Euclidean space is then an infinite cylinder of circumference $L$ with a complex coordinate

---

[30]The excitation is now the null hypothesis $\rho$ in the conventions of section 2. This choice is slightly unnatural, because normally the excitation is the signal one wants to detect with the vacuum state being the null hypothesis. However, in the present CFT context, $\rho$ being the excitation is more convenient to analyze. See also footnote 7.

$w = \phi + i\tau$ where $\phi \sim \phi + L$ is the spatial coordinate and $\tau \in \mathbb{R}$ is the Euclidean time coordinate. We will study the interval $I = [0, \ell]$ with $0 < \ell < L$ on the $\tau = 0$ circle. We map the cylinder to the complex plane using the map

$$w \longmapsto z = e^{2\pi i w / L} , \tag{7.45}$$

so that the Cauchy slice $\tau = 0$ is mapped to the $|z| = 1$ circle. The interval $I$ is mapped to the circular arc between $z = 1$ and $z = e^{2\pi i \lambda}$ with $\lambda = \ell/L$. Using a primary operator $\Phi$, we create an excited state $|\Phi\rangle = \Phi(0)|0\rangle$ in radial quantization by performing the path integral over the unit disk with $\Phi(0)$ inserted at the origin. The corresponding bra state is then defined as $\langle\Phi| = \langle 0|\Phi(0)^\dagger$ where $\Phi(z,\bar{z})^\dagger = (1/\bar{z})^{2\bar{h}_\Phi}(1/z)^{2h_\Phi}\Phi^\dagger(1/\bar{z}, 1/z)$ so that $\Phi^\dagger$ is inserted at $z = \infty$.

We further perform the conformal transformation

$$z \longmapsto \zeta = e^{i\pi\lambda} \frac{z-1}{z - e^{2i\pi\lambda}} , \tag{7.46}$$

which maps the Cauchy slice $|z| = 1$ to the real axis with the interval $I$ mapped to the negative real axis.[31] We define two reduced density matrices on $I$ by tracing over its complement:

$$\sigma = \mathrm{Tr}_{I^c}|0\rangle\langle 0|, \qquad \rho = \frac{1}{Z}\mathrm{Tr}_{I^c}|\Phi\rangle\langle\Phi| . \tag{7.47}$$

The vacuum modular Hamiltonian is defined as $K \equiv -\log\sigma$. In our conventions, $K/(2\pi)$ generates counter-clockwise rotations in the $\zeta$-plane.

The excited state $\rho$ is computed by a path integral over the $\zeta$-plane with a cut along the negative real axis and with operator insertions $\Phi(e^{-\pi i\lambda})$ and $\Phi^\dagger(e^{\pi i\lambda})$. We rotate the boundary conditions above and below the cut to the positive real axis using $\sigma^{1/2}$ which gives the Rindler representation of the density matrix:

$$\rho = \frac{\sigma^{1/2}\Phi^\dagger(e^{\pi i\lambda})\Phi(e^{-\pi i\lambda})\sigma^{1/2}}{\langle\Phi^\dagger(e^{\pi i\lambda})\Phi(e^{-\pi i\lambda})\rangle}. \tag{7.48}$$

Here the vacuum 2-point function $\langle\cdot\rangle = \mathrm{Tr}(\sigma\,\cdot)$ in the denominator ensures that $\mathrm{Tr}\,\rho = 1$.[32] See [71] for an analogous representation of $\rho$ in higher dimensions.

As in [71], we expand $\rho$ in the short interval limit $\lambda \to 0$ using the OPE[33]

$$\Phi^\dagger(e^{\pi i\lambda})\Phi(e^{-\pi i\lambda}) = \langle\Phi^\dagger(e^{\pi i\lambda})\Phi(e^{-\pi i\lambda})\rangle\left(1 + (2\pi\lambda)^\Delta C^{\mathcal{O}}_{\Phi\Phi^\dagger}\mathcal{O}(1) + \dots\right) , \tag{7.49}$$

where $\Delta$ is the scaling dimension of the lightest primary $\mathcal{O}$ of the theory that couples to $\Phi$ (in the sense that the OPE coefficient $C^{\mathcal{O}}_{\Phi\Phi^\dagger}$ is non-zero), which we assume to be spinless and real for simplicity. Since two-point functions of real primaries are normalized to the Kronecker delta, we can lower the index in the OPE coefficient $C^{\mathcal{O}}_{\Phi\Phi^\dagger} = C_{\mathcal{O}\Phi\Phi^\dagger}$.

Based on the OPE, we take the expansion parameter to be $(\pi\lambda)^\Delta$ so that

$$\rho = \sigma + (\pi\lambda)^\Delta\rho^{(1)} + \dots \tag{7.50}$$

with

$$\rho^{(1)} = 2^\Delta C_{\mathcal{O}\Phi\Phi^\dagger}\sigma^{1/2}\mathcal{O}(1)\sigma^{1/2}. \tag{7.51}$$

We can now start constructing the acceptance subspace. Given an eigenbasis $|\mathbf{E}\rangle$ of $\sigma^{\otimes n}$ in $\mathcal{H}_A^{\otimes n}$, the optimal classical measurement is determined by an acceptance condition of the form

$$\mathbf{E} + \frac{1}{n}\log\langle\mathbf{E}|\rho^{\otimes n}|\mathbf{E}\rangle \geq \mathcal{E} . \tag{7.52}$$

---

[31]See [2] for more details on this setup.

[32]The expression (7.48) is Hermitian since the adjoint maps the operator insertions $\Phi^\dagger$ and $\Phi$ into each other.

[33]Note that $\left(e^{\pi i\lambda} - e^{-\pi i\lambda}\right)^{h_{\mathcal{O}}}\left(e^{-\pi i\lambda} - e^{\pi i\lambda}\right)^{h_{\mathcal{O}}} = (2\pi\lambda)^\Delta$ for small $\lambda$.

We first consider the case $n = 1$ of a single copy, for which we have $|\mathbf{E}\rangle = |E\rangle$. From

$$\langle E|\rho|E\rangle = e^{-E} + (\pi\lambda)^{\Delta}\langle E|\rho^{(1)}|E\rangle + \dots , \tag{7.53}$$

we obtain

$$E + \log\langle E|\rho|E\rangle = (\pi\lambda)^{\Delta}e^{E}\langle E|\rho^{(1)}|E\rangle + \dots \tag{7.54}$$

Next, using the above Rindler quantization, we see that

$$E + \log\langle E|\rho|E\rangle = 2^{\Delta}(\pi\lambda)^{\Delta}C_{\mathcal{O}\Phi\Phi^{\dagger}}\langle E|\mathcal{O}(1)|E\rangle , \tag{7.55}$$

where the states $|E\rangle$ now live on the positive real axis in the complex $\zeta$-plane. Rotating the expectation value $\langle E|\mathcal{O}(1)|E\rangle$ to the negative real axis and mapping back to the $w$-cylinder, we get

$$E + \log\langle E|\rho|E\rangle = \left(\frac{L}{2\pi}\right)^{\Delta}(\pi\lambda)^{2\Delta}C_{\mathcal{O}\Phi\Phi^{\dagger}}\mathcal{O}(E) , \tag{7.56}$$

where $\mathcal{O}(E) \equiv \langle E|\mathcal{O}(\ell/2)|E\rangle$ is the one-point function in the eigenstate $|E\rangle$ of the operator $\mathcal{O}$ inserted at the midpoint of the interval $I$. Hence to determine the acceptance subspace, one has to compute these one-point functions first. This can be seen as a precomputation that can be done once and for all for each $\mathcal{O}$ that one wishes to use.

Let us now return to the case of $n$ copies using the same notation as in section 4.2. We denote

$$|\mathbf{E}\rangle = |E_1\rangle \otimes |E_2\rangle \otimes \dots \otimes |E_n\rangle , \tag{7.57}$$

and eigenstate of $\sigma^{\otimes n}$ and we use $|\mathbf{E}| = \frac{1}{n}\sum_{i=1}^{n} E_i$. The acceptance condition is

$$\mathbf{E} + \frac{1}{n}\log\langle\mathbf{E}|\rho^{\otimes n}|\mathbf{E}\rangle \geq S(\rho_D\|\sigma) + \sqrt{\frac{V(\rho_D\|\sigma)}{n}}\Phi^{-1}(\varepsilon) , \tag{7.58}$$

and we have

$$|\mathbf{E}| + \frac{1}{n}\log\langle\mathbf{E}|\rho^{\otimes n}|\mathbf{E}\rangle = \left(\frac{L}{2\pi}\right)^{\Delta}(\pi\lambda)^{2\Delta}C_{\mathcal{O}\Phi\Phi^{\dagger}}\mathcal{O}(\mathbf{E}) , \tag{7.59}$$

where we denote the average of the precomputed values

$$\mathcal{O}(\mathbf{E}) = \frac{1}{n}\sum_{i=1}^{n}\mathcal{O}(E_i) . \tag{7.60}$$

In the short interval limit, relative entropy has the expansion[34]

$$S(\rho\|\sigma) = \frac{(\pi\lambda)^{2\Delta}}{2}S^{(2)}(\rho\|\sigma) + \dots \tag{7.61}$$

Although it might be subtle to properly define $\rho_D$ in a continuum CFT, we expect that $S(\rho_D\|\sigma)$ has a similar expansion since positivity and monotonicity implies that $0 \leq S(\rho_D\|\sigma) \leq S(\rho\|\sigma)$. Hence, in the short interval limit, the acceptance condition becomes

$$C_{\mathcal{O}\Phi\Phi^{\dagger}}\mathcal{O}(\mathbf{E}) \geq \frac{1}{2}\left(\frac{2\pi}{L}\right)^{\Delta}S^{(2)}(\rho_D\|\sigma) . \tag{7.62}$$

This is a condition on the one-point functions of the lightest primary $\mathcal{O}$ which couples to $\Phi$, inserted at the interval midpoint. The measurement that implements the likelihood ratio test is then the projection on the eigenstates of $\sigma^{\otimes n}$ satisfying this condition:

$$A^{(n)} = \mathcal{P}_{\mathcal{H}_C}, \qquad \mathcal{H}_C = \text{span}\{|\mathbf{E}\rangle \mid (7.62)\} . \tag{7.63}$$

---

[34]The explicit expression for $S^{(2)}(\rho\|\sigma)$ can be found in [72].

# 8 Discussion

In this paper we have reviewed some aspects of quantum hypothesis testing and studied a few applications in quantum many-body systems and two-dimensional conformal field theories. We have mostly focused on asymmetric testing, with a few comments about the symmetric counterpart. We believe that we have only scratched the surface of this subject and would like to conclude by mentioning some possible avenues for future investigation.

We have seen that the error estimates of different types of hypothesis testing involve different interesting quantum information theoretic quantities. One is therefore led to wonder which notions of distance on the space of states can arise in error estimates of different types of quantum hypothesis testing, and whether there is a more direct connection between properties of the distance measure and features of the type of test.

We have also observed that the (non-unique) optimal measurement which saturates the error bound in the large $n$ limit tends to be rather difficult to implement in practice. For the case of asymmetric testing, the measurement we studied requires knowledge of the spectra of eigenstates of the modular Hamiltonians associated to subsystems, which is in general difficult if not impossible to obtain. An important question is therefore whether there are simpler testing protocols that one can develop which still do reasonably well in the large $n$ limit. In this paper we have considered the likelihood ratio test as a possible alternative, but it would be interesting to explore this question in more detail. From a practical point of view, one ultimately would like to find the simplest possible protocol whose asymptotic error does not deviate too much from the optimal one.

An important assumption of quantum hypothesis testing is the ability to perform simultaneous (collective) measurements on $n$ copies of the system, for arbitrarily large $n$. Clearly, this assumption is not realistic, and the finite $n$ or finite blocklength case has been considered in [16, 17]. One could imagine applying finite $n$ measurements in cases where one has an evenly spaced collection of subsystems in a translation invariant state, where the distance between the subsystems is large enough for the subsystems to be approximately uncorrelated. But the situation that is most realistic is arguably to make a repeated series of single-shot measurements, *i.e.* one prepares the systems in a particular state, makes a measurement, and then repeats this procedure $n$ times. It is not necessarily true that the best strategy in this case is to repeat the optimal $n = 1$ measurement $n$ times, it is conceivable that a series of different measurement protocol yields a better outcome. Such adaptive measurement strategies in symmetric testing are known to attain the optimal error probability of collective strategies [73] and we leave the asymmetric case to future work. There are various closely related questions which deserve further study, such as distinguishing more than two states through POVM's [74], and contrasting these results with continuous parameter measurements and ideas from quantum metrology.

One important motivation for this work came from quantum gravity and holography. For example, in [75] a relationship was found between distinguishability measures and bulk reconstruction in entanglement wedges. One could imagine that the quantum hypothesis testing protocol whose errors are bounded by these measures plays an operational role in the actual reconstruction process and it would be interesting to explore this in more detail. Many other questions in quantum gravity center around the issue of whether or not different states can be distinguished by low energy observers, and if so, whether the necessary measurements are very complex or not. Translated into the language of quantum hypothesis testing, one would like to bound the error associated to restricted measurements (*e.g.* the measurements can only be made by low energy observers). In particular, can one bound the errors in hypothesis testing as a function of the maximal complexity of the measurements? This question involves the need to first develop rigorous definitions of complexity of a measurement. We briefly touched

upon this in section 5.2 by considering the minimum dimension of the acceptance space as one resource associated with a measurement. More sophisticated definitions would take into account additional steps involved in the construction of the POVM, and the time and space associated with the algorithms or circuits executing the measurement. We hope to return to some of these questions in future work.

## Acknowledgments

We thank M. Walter for very useful discussions and a critical reading of the manuscript. JK and EKV are supported in part by the Academy of Finland grant no 1297472. JK is also supported in part by a grant from the Osk. Huttunen Foundation. JdB is supported by the European Research Council under the European Unions Seventh Framework Programme (FP7/2007-2013), ERC Grant agreement ADG 834878. The work of VG is supported by the Delta ITP consortium, a program of the NWO that is funded by the Dutch Ministry of Education, Culture and Science (OCW). JdB and EKV thank the It from Qubit school/workshop "Quantum Information and String Theory 2019" and YITP Kyoto for hospitality and partial support during this work. VG thanks University of Helsinki for hospitality during the completion of this work and JDB, VG and JK also thank Strings 2019 for hospitality during the early parts of this work.

# A  Measurements for symmetric hypothesis testing

This paper focuses on asymmetric hypothesis testing, where we minimize the type II error $\beta$ under the condition that the type I error $\alpha$ is bounded. Section 4.2 describes the optimal measurement for asymmetric testing. In this appendix, we will discuss the optimal measurement for symmetric testing, where we try to distinguish between $\rho^{\otimes n}$ and $\sigma^{\otimes n}$ by minimizing the combined error

$$P_n = \kappa \alpha_n + (1-\kappa)\beta_n, \qquad 0 < \kappa < 1 \,, \tag{A.1}$$

where $\beta_n = \mathrm{Tr}(\sigma^{\otimes n}A)$ and $\alpha_n = 1 - \mathrm{Tr}(\rho^{\otimes n}A)$. In section 2.1, we considered the case $\kappa = \frac{1}{2}$ but the same result holds for any $\kappa$ with $0 < \kappa < 1$. Asymptotically, the optimal error is given in terms of the Chernoff distance

$$\lim_{n\to+\infty}\left(-\frac{1}{n}\log P_n\right) = -\log Q(\rho,\sigma), \qquad Q(\rho,\sigma) = \min_{0\leq s\leq 1} \mathrm{Tr}\,\rho^s\sigma^{1-s}\,. \tag{A.2}$$

The optimal measurement was obtained in [42] and is the projection on the positive part of

$$L = \kappa\rho^{\otimes n} - (1-\kappa)\sigma^{\otimes n}\,. \tag{A.3}$$

This involves diagonalizing the operator $L$ and projecting onto the subspace corresponding to positive eigenvalues. In general, it is difficult to describe explicitly this measurement. We consider simplified cases below.

## A.1  Classical testing

We use the same notation as in section 4. We take $\{|E\rangle\}$ to be the eigenstates of $\sigma$ and for $n$ copies of the system, the eigenstates of $\sigma^{\otimes n}$ can be written

$$|\mathbf{E}\rangle = |E_1\rangle \otimes |E_2\rangle \otimes \cdots \otimes |E_n\rangle \,. \tag{A.4}$$

As in section 4.3, we can define the best classical measurement by the acceptance condition

$$|\mathbf{E}| + \frac{1}{n}\log\langle\mathbf{E}|\rho^{\otimes n}|\mathbf{E}\rangle \geq \frac{1}{n}\log\left(\frac{\kappa}{1-\kappa}\right)\,, \tag{A.5}$$

where we recall that $|\mathbf{E}| \equiv \frac{1}{n} \sum_{i=1}^{n} E_i$. The measurement is the projector onto the subspace spanned by the states $|\mathbf{E}\rangle$ satisfying this condition. This is also a likelihood-ratio test but with a different threshold value.

When $\rho$ and $\sigma$ commute, the acceptance condition (A.5) is precisely the positivity of the operator $L$ so this is actually the optimal measurement. When $\rho$ and $\sigma$ don't commute, we can define the diagonal part of $\rho$

$$\rho_D \equiv \sum_E \langle E|\rho|E\rangle |E\rangle\langle E| , \tag{A.6}$$

and the above measurement optimally distinguishes between $\rho_D$ and $\sigma$ but doesn't make use of the off-diagonal components of $\rho$. This gives an error

$$\lim_{n \to +\infty} \left( -\frac{1}{n} \log P_n \right) = -\log Q(\rho_D, \sigma) , \tag{A.7}$$

and the data-processing inequality for the Chernoff distance implies that

$$-\log Q(\rho_D, \sigma) \le -\log Q(\rho, \sigma) , \tag{A.8}$$

so this measurement is suboptimal as expected. In conclusion, as in asymmetric hypothesis testing, the likelihood-ratio test (with a different threshold value) provides a simple measurement for symmetric testing which is the optimal classical measurement.

## A.2 Perturbative testing

We now consider the perturbative setting where we have

$$\rho = \sigma + \lambda \rho^{(1)} + O(\lambda^2) . \tag{A.9}$$

We define

$$L_i = \sigma \otimes \cdots \otimes \sigma \otimes \rho^{(1)} \otimes \sigma \otimes \cdots \otimes \sigma , \tag{A.10}$$

where $\rho$ is in the $i$-th position and there are $n$ tensor factors. Perturbatively, we have

$$L = (2\kappa - 1)\sigma^{\otimes n} + \lambda\kappa \sum_{i=1}^{n} L_i + \mathcal{O}(\lambda^2) . \tag{A.11}$$

We see that perturbative testing is non-trivial only for $\kappa = \frac{1}{2}$. For $\kappa > \frac{1}{2}$, $L$ is positive so that the measurement is the identity while for $\kappa < \frac{1}{2}$, $L$ is negative so the measurement is zero. Focusing on the case $\kappa = \frac{1}{2}$, the measurement is a projection on the positive part of

$$L = \frac{\lambda}{2} \sum_{i=1}^{n} L_i . \tag{A.12}$$

In the case where $\rho^{(1)}$ and $\sigma$ commute, this reduces to the classical measurement described in the previous section.

## B General properties of the relative entropy variance

The relative entropy variance is a less familiar concept than the relative entropy, and we survey here some of its properties. Introducing the modular Hamiltonians of $\rho$ and $\sigma$,

$$K = -\log\sigma, \qquad \widetilde{K} = -\log\rho , \tag{B.1}$$

we consider the so-called relative modular Hamiltonian

$$\Delta K = K - \widetilde{K} \, . \tag{B.2}$$

Then, the relative entropy and the relative entropy variance are its first and second cumulants, *i.e.* the expectation value and the variance, in the state $\rho$:

$$S(\rho\|\sigma) \;=\; \langle \Delta K \rangle_\rho \, , \tag{B.3}$$

$$V(\rho\|\sigma) \;=\; \langle \Delta K^2 \rangle_\rho - \langle \Delta K \rangle_\rho^2 \, . \tag{B.4}$$

### B.1 Relations to other quantities

We give here the relations between the relative entropy variance $V(\rho\|\sigma)$ and other information quantities.

**Rényi relative entropies.** In the literature there are different generalizations of the relative entropy. Petz's defines [44] *Rényi relative entropies* as

$$D_\alpha(\rho\|\sigma) \equiv \frac{1}{\alpha - 1} \log \operatorname{Tr} \rho^\alpha \sigma^{1-\alpha} \, , \tag{B.5}$$

with $D_1(\rho\|\sigma) = S(\rho\|\sigma)$. On the other hand, the *sandwiched Rényi entropy* or the *quantum Rényi divergence* is defined in [76,77] as

$$\widetilde{D}_\alpha(\rho\|\sigma) \equiv \frac{1}{\alpha - 1} \log \operatorname{Tr} \left[ \left( \sigma^{\frac{1-\alpha}{2\alpha}} \rho \, \sigma^{\frac{1-\alpha}{2\alpha}} \right)^\alpha \right] \, . \tag{B.6}$$

The relative entropy variance can be obtained from both versions of Rényi relative entropy [21,78],

$$V(\rho\|\sigma) = \partial_\alpha^2 [(\alpha-1) D_\alpha(\rho\|\sigma)]_{\alpha=1} = \partial_\alpha^2 [(\alpha-1) \widetilde{D}_\alpha(\rho\|\sigma)]_{\alpha=1} \, . \tag{B.7}$$

It is shown in [21] that the sandwiched Rényi entropy is the minimal quantity that satisfies the axioms expected from a relative Rényi entropy. In particular, we always have

$$D_\alpha(\rho\|\sigma) \geq \widetilde{D}_\alpha(\rho\|\sigma) \, . \tag{B.8}$$

**Refined Rényi relative entropies.** In [11], a refined version of the Rényi relative entropies was defined as

$$\widetilde{S}_\alpha(\rho\|\sigma) = \alpha^2 \partial_\alpha \left( \frac{\alpha - 1}{\alpha} \widetilde{D}_\alpha(\rho\|\sigma) \right) \, , \tag{B.9}$$

where $\widetilde{D}_\alpha(\rho\|\sigma)$ is the sandwiched Rényi entropy. In AdS/CFT, this quantity was shown to have a holographic dual when $\sigma$ is the vacuum state reduced to a spherical subregion. It is analogous to the refined Rényi entropies defined in [79]. The relative entropy variance is obtained as

$$V(\rho\|\sigma) = \partial_\alpha \widetilde{S}_\alpha(\rho\|\sigma) \big|_{\alpha=1} \, . \tag{B.10}$$

**Higher cumulants.** It's also possible to give an interpretation to the higher $\alpha$ derivatives of the Petz relative Rényi entropy $D_\alpha(\rho\|\sigma)$ at $\alpha = 1$. This is better done in the algebraic formulation given in section B.4. They correspond to cumulants of the operator $-\log \Delta_{\Psi|\Phi}$,[35] which are not equivalent to cumulants of $\Delta K$. Their first and second cumulants are the same and give the relative entropy and its variance, but the higher cumulants differ. Following [21], the higher $\alpha$ derivatives of $D_\alpha(\rho\|\sigma)$ can also be interpreted as classical cumulants of the loglikelihood of the Nussbaum–Szkola probability distributions associated to $\rho$ and $\sigma$. Note that the higher $\alpha$ derivatives of $\widetilde{D}_\alpha(\rho\|\sigma)$ differ from that of $D_\alpha(\rho\|\sigma)$ because they are different functions of $\alpha$.

---

[35]Here, the terminology can be confusing because both operators are called relative modular Hamiltonian in different contexts, although they are not equivalent.

**Capacity of entanglement.**   For density matrices in a finite dimensional Hilbert space with $\dim \mathcal{H} = N$, it is simple to derive a relationship between the Rényi entropy and its relative generalization. Let $\sigma_{\text{max}}$ be the density matrix with uniform spectrum, *i.e.* proportional to the unit matrix,

$$\sigma_{\text{max}} = \frac{1}{N} \mathbf{1}_N \ . \tag{B.11}$$

Then the Rényi relative entropy between an arbitrary state $\rho$ and $\sigma_{\text{max}}$ reduces to

$$\widetilde{D}_\alpha(\rho \| \sigma_{\text{max}}) = \log N - S_\alpha(\rho) \ , \tag{B.12}$$

where

$$S_\alpha(\rho) = \frac{1}{1-\alpha} \log \text{Tr}(\rho^\alpha) \tag{B.13}$$

is the Rényi entropy. The relative entropy, respectively, reduces to the von Neumann entropy by

$$S(\rho \| \sigma_{\text{max}}) = \log N - S(\rho) \tag{B.14}$$

and, the relative entropy variance reduces to the variance of the entropy, also known as the capacity of entanglement (see [51] and references therein),

$$V(\rho \| \sigma_{\text{max}}) = C(\rho) \equiv \text{Tr} \, \rho (\log \rho)^2 - (\text{Tr} \, \rho \log \rho)^2 \ . \tag{B.15}$$

The capacity of entanglement vanishes for a pure state $\rho_\psi = |\psi\rangle\langle\psi|$ and for the maximally mixed state $\sigma_{\text{max}}$. It follows that the relative entropy variance vanishes between a pure state and a maximally mixed state

$$V(\rho_\psi \| \sigma_{\text{max}}) = C(\rho_\psi) = 0 \ . \tag{B.16}$$

We next give necessary and sufficient for the vanishing of the relative entropy variance.

## B.2   Vanishing of the variance

The relative entropy variance $V(\rho \| \sigma)$ is nonnegative. In this section, we consider the conditions for it to vanish, for finite-dimensional Hilbert space. When $\rho$ is full-rank, the variance vanishes if and only if $\rho = \sigma$. More generally, the variance vanishes if and only if $\rho$ and $\sigma$ are proportional on the complement of $\ker \rho$, where $\ker \rho$ is the subspace on which $\rho$ vanishes. This is explained in [37] and follows from the saturation case of the Cauchy–Schwarz inequality.

This implies that the relative entropy variance $V(\rho \| \sigma)$ vanishes when $\rho = |\psi\rangle\langle\psi|$ is a pure state and $\sigma$ has no matrix element between $|\psi\rangle$ and any other state. For example, the relative entropy variance vanishes between the vacuum (the ground state) and any thermal state.

## B.3   Violation of data processing inequality

The hypothesis testing relative entropy and the relative entropy are generalized divergences $D(\rho \| \sigma)$, satisfying the data processing inequality

$$D(\rho \| \sigma) \geq D(\mathcal{N}(\rho) \| \mathcal{N}(\sigma)) \ , \tag{B.17}$$

where $\mathcal{N}$ is a quantum channel. The refinement of quantum Stein's lemma (2.14) gives an asymptotic expansion for the hypothesis testing relative entropy (2.17), involving the relative entropy and the relative entropy variance, so it is interesting to note that the latter alone does not satisfy the data processing inequality. Given a quantum channel $\mathcal{N}$, there is no general

inequality between $V(\rho\|\sigma)$ and $V(\mathcal{N}(\rho)\|\mathcal{N}(\sigma))$. This can be seen in a simple two-qubit system with pure density matrices

$$
\begin{aligned}
\rho &= |\psi\rangle\langle\psi|, & |\psi\rangle &= |00\rangle \\
\sigma &= |\chi\rangle\langle\chi|, & |\chi\rangle &= \frac{1}{\sqrt{3}}(|01\rangle + |10\rangle + |11\rangle) .
\end{aligned}
\tag{B.18}
$$

As a quantum channel, consider the partial trace over the second qubit. It produces the reduced density matrices

$$
\rho_A = |0\rangle\langle0|, \qquad \sigma_A = \frac{1}{3}(|0\rangle\langle0| + |0\rangle\langle1| + |1\rangle\langle0| + 2|1\rangle\langle1|) .
\tag{B.19}
$$

We obtain for the relative entropy[36]

$$
S(\rho\|\sigma) = +\infty, \qquad S(\rho_A\|\sigma_A) = \log(3) + \frac{2}{\sqrt{5}}\operatorname{arccoth}(\sqrt{5})
\tag{B.20}
$$

in agreement with monotonicity that says that $S(\rho_A\|\sigma_A) \le S(\rho\|\sigma)$. For the relative entropy variance, we obtain

$$
V(\rho\|\sigma) = 0, \qquad V(\rho_A\|\sigma_A) = \frac{4}{5}\log\left(\frac{2}{3+\sqrt{5}}\right)^2 .
\tag{B.21}
$$

This shows that the variance is not monotonous since we have

$$
V(\rho_A\|\sigma_A) > V(\rho\|\sigma) .
\tag{B.22}
$$

### B.4 Algebraic formulation

We can also define the relative entropy variance for infinite-dimensional Hilbert space, in the context of algebraic quantum field theory (we refer to [80] for a review). This allows a rigorous definition of this quantity in the case of conformal field theory. Araki defined the relative entropy between two states $\Psi$ and $\Phi$

$$
S_{\Psi|\Phi} = -\langle\Psi|\log\Delta_{\Psi|\Phi}|\Psi\rangle ,
\tag{B.23}
$$

in terms of the relative modular operator $\Delta_{\Psi|\Phi}$ defined with respect to a subsystem for which $\Psi$ is cyclic and separating. In the finite-dimensional case, $\rho$ and $\sigma$ are the reduced states of $\Psi$ and $\Phi$ in that subsystem. We recover the usual definition of relative entropy, as can be seen from the formula

$$
\langle\Psi|\Delta_{\Psi|\Phi}^{1-\alpha}|\Psi\rangle = \operatorname{Tr}\rho^\alpha\sigma^{1-\alpha} .
\tag{B.24}
$$

This also allows us to write the Petz relative Rényi entropy as

$$
(\alpha-1)D_\alpha(\Psi\|\Phi) = \log\langle\Psi|e^{-(\alpha-1)\log\Delta_{\Psi|\Phi}}|\Psi\rangle ,
\tag{B.25}
$$

which realizes it as a well-defined UV finite quantity in quantum field theory. In particular, taking two derivatives gives us an algebraic definition of the relative entropy variance

$$
V_{\Psi|\Phi} = \langle\Psi|(\log\Delta_{\Psi|\Phi})^2|\Psi\rangle - (\langle\Psi|\log\Delta_{\Psi|\Phi}|\Psi\rangle)^2 ,
\tag{B.26}
$$

which shows that the relative entropy variance is well-defined in quantum field theory. This formulation also gives an interpretation for the higher $\alpha$ derivatives of the Petz relative Rényi

---

[36]The computation of the logarithms is done by adding a small matrix $\varepsilon\mathbf{1}$ and taking the limit $\varepsilon \to 0$ at the end.

entropy at $\alpha = 1$. The Petz relative Rényi entropy is the cumulant generating function of the operator

$$K_{\Psi|\Phi} = -\log\Delta_{\Psi|\Phi} \ . \tag{B.27}$$

Note that this operator is not equivalent to the operator $\Delta K$ defined in (B.2). In particular, the Petz relative Rényi entropy does not generate the cumulants of $\Delta K$. It is however true that the first and second cumulants of $K_{\Psi|\Phi}$ and $\Delta K$ agree ; they give the relative entropy and its variance. An algebraic version of the sandwiched relative Rényi entropy has been investigated in [81].

## C  Optimal measurement of a qubit

We discuss here the optimal measurement in the case of a qubit and give the derivations of the formulas of section 5.2. We focus on the case $\theta = \frac{\pi}{4}$ which appears to be the simplest case when $\rho$ and $\sigma$ don't commute and we want to describe the optimal measurement. It is useful to write

$$\sigma = \left(\frac{1}{2}+q\right)|1\rangle\langle 1| + \left(\frac{1}{2}-q\right)|0\rangle\langle 0| \ , \tag{C.1}$$

so that $\frac{1}{2}+q = e^{-E_1} = 1 - e^{-E_0}$. As a result, the optimal threshold value for $\varepsilon = \frac{1}{2}$ gives

$$n_*(\widetilde{\mathbf{E}}) = n(\widetilde{\mathbf{E}}) + qn \ . \tag{C.2}$$

We recall that $|\mathbf{E}\rangle$ and $|\widetilde{\mathbf{E}}\rangle$ are binary strings

$$\begin{aligned} |\mathbf{E}\rangle &= |a_1 a_2 \ldots a_n\rangle \ , & a_i \in \{0,1\} \ , \\ |\widetilde{\mathbf{E}}\rangle &= |\tilde{a}_1 \tilde{a}_2 \ldots \tilde{a}_n\rangle \ , & \tilde{a}_i \in \{-,+\} \ , \end{aligned} \tag{C.3}$$

where we used the fact that $|\widetilde{0}\rangle = |-\rangle$ and $|\widetilde{1}\rangle = |+\rangle$ for $\theta = \frac{\pi}{4}$. It is useful to introduce the notation $n_{s\tilde{s}}(\mathbf{E},\widetilde{\mathbf{E}})$, with $s \in \{0,1\}$ and $\tilde{s} \in \{-,+\}$, counting the number of pairs $(a_i, \tilde{a}_i)$ which are equal to $(s,\tilde{s})$. We then have

$$|\xi(\widetilde{\mathbf{E}})\rangle = \frac{1}{2^{n/2}} \sum_{\substack{\mathbf{E} \\ n(\mathbf{E}) \geq n_*(\widetilde{\mathbf{E}})}} (-1)^{n_{0+}(\mathbf{E},\widetilde{\mathbf{E}})} |\mathbf{E}\rangle \ . \tag{C.4}$$

Let's now compute the overlap of two states $|\xi(\widetilde{\mathbf{E}}_1)\rangle$ and $|\xi(\widetilde{\mathbf{E}}_2)\rangle$. We can write

$$\langle\xi(\widetilde{\mathbf{E}}_1)|\xi(\widetilde{\mathbf{E}}_2)\rangle = \frac{1}{2^n} \sum_{\substack{\mathbf{E} \\ n(\mathbf{E}) \geq n_*(\widetilde{\mathbf{E}}_1,\widetilde{\mathbf{E}}_2)}} (-1)^{n_{0+}(\mathbf{E},\widetilde{\mathbf{E}}_1)+n_{0+}(\mathbf{E},\widetilde{\mathbf{E}}_2)} \ , \tag{C.5}$$

where we introduced the notation

$$n_*(\widetilde{\mathbf{E}}_1,\widetilde{\mathbf{E}}_2) = \max(n_*(\widetilde{\mathbf{E}}_1), n_*(\widetilde{\mathbf{E}}_2)). \tag{C.6}$$

We also denote $n_{\tilde{s}_1\tilde{s}_2}$ for the number of overlapping pairs $(\tilde{s}_1,\tilde{s}_2)$ in $(\widetilde{\mathbf{E}}_1,\widetilde{\mathbf{E}}_2)$ and $n_{s\tilde{s}_1\tilde{s}_2}$ for the number of overlapping pairs $(s,\tilde{s}_1,\tilde{s}_2)$ in $(\mathbf{E},\widetilde{\mathbf{E}}_1,\widetilde{\mathbf{E}}_2)$. We have the relations

$$n_{0\tilde{s}_1\tilde{s}_2} + n_{1\tilde{s}_1\tilde{s}_2} = n_{\tilde{s}_1\tilde{s}_2} \ , \tag{C.7}$$

and we have

$$n(\mathbf{E}) = n - (n_{0--} + n_{0-+} + n_{0+-} + n_{0++}) \ . \tag{C.8}$$

Hence, the acceptance condition is

$$n_{0--} + n_{0-+} + n_{0+-} + n_{0++} \geq n - n_* .\tag{C.9}$$

We can rewrite the sum over $\mathbf{E}$ as a sum over the four integers $n_{0\pm\pm}$ with the combinatorial factor

$$\binom{n_{++}}{n_{0++}}\binom{n_{+-}}{n_{0+-}}\binom{n_{-+}}{n_{0-+}}\binom{n_{--}}{n_{0--}} ,\tag{C.10}$$

counting the number of basis state $|\mathbf{E}\rangle$ for a given choice of $n_{0\pm\pm}$ . We then have

$$\langle \xi(\widetilde{\mathbf{E}}_1)|\xi(\widetilde{\mathbf{E}}_2)\rangle = \frac{1}{2^n} \sum_{\substack{\mathbf{E} \\ n(\mathbf{E})\geq n_*(\widetilde{\mathbf{E}}_1,\widetilde{\mathbf{E}}_2)}} (-1)^{n_{0+}(\mathbf{E},\widetilde{\mathbf{E}}_1)+n_{0+}(\mathbf{E},\widetilde{\mathbf{E}}_2)}\tag{C.11}$$

$$= \frac{1}{2^n} \sum_{\substack{n_{0--},n_{0-+},n_{0+-},n_{0++} \\ n_{0--}+n_{0-+}+n_{0+-}+n_{0++}\geq n-n_*}} \binom{n_{++}}{n_{0++}}\binom{n_{+-}}{n_{0+-}}\binom{n_{-+}}{n_{0-+}}\binom{n_{--}}{n_{0--}}(-1)^{n_{0+-}+n_{0-+}} .$$

It is convenient to define

$$P_{n,k} = \frac{1}{2^n} \sum_{\substack{n_{0--},n_{0-+},n_{0+-},n_{0++} \\ n_{0--}+n_{0-+}+n_{0+-}+n_{0++}=k}} \binom{n_{++}}{n_{0++}}\binom{n_{+-}}{n_{0+-}}\binom{n_{-+}}{n_{0-+}}\binom{n_{--}}{n_{0--}}(-1)^{n_{0+-}+n_{0-+}} ,\tag{C.12}$$

so that we have

$$\langle \xi(\widetilde{\mathbf{E}}_1)|\xi(\widetilde{\mathbf{E}}_2)\rangle = \frac{1}{2^n} \sum_{k=n-n_*(\widetilde{\mathbf{E}}_1,\widetilde{\mathbf{E}}_2)}^{n} P_{n,k} .\tag{C.13}$$

It can be noted that $P_{n,k}$ are coefficients of the polynomial

$$P_n(x) = (1+x)^{n_{++}}(1+x)^{n_{--}}(1-x)^{n_{+-}}(1-x)^{n_{-+}} = \sum_{k=0}^{n} P_{n,k}x^k .\tag{C.14}$$

This follows from expanding each factor using the binomial theorem. Note that we can write

$$P_n(x) = (1+x)^{n(\widetilde{\mathbf{E}}_1+\widetilde{\mathbf{E}}_2)}(1-x)^{n-n(\widetilde{\mathbf{E}}_1+\widetilde{\mathbf{E}}_2)} = \sum_{k=0}^{n} P_{n,k}x^k ,\tag{C.15}$$

where $\widetilde{\mathbf{E}}_1+\widetilde{\mathbf{E}}_2$ denotes the boolean sum. This follows from the fact that $n(\widetilde{\mathbf{E}}_1+\widetilde{\mathbf{E}}_2) = n_{++}+n_{--}$. This second expression gives an alternative representation of the coefficients $P_{n,k}$ as

$$P_{n,k} = \sum_{m=0}^{k}(-1)^m \binom{n(\widetilde{\mathbf{E}}_1+\widetilde{\mathbf{E}}_2)}{m}\binom{n-n(\widetilde{\mathbf{E}}_1+\widetilde{\mathbf{E}}_2)}{k-m} .\tag{C.16}$$

Let us introduce binary Krawtchouk polynomials $\mathcal{K}_k(X;n)$ which can be defined via the generating relation

$$(1+x)^{n-X}(1-x)^X = \sum_{k\geq 0} \mathcal{K}_k(X;n)x^k .\tag{C.17}$$

These are discrete orthogonal polynomials related to the binomial distribution which have many applications [82,83]. From the definition for $P_{n,k}$ in (C.15), we see that

$$P_{n,k} = (-1)^k \mathcal{K}_k(n(\widetilde{\mathbf{E}}_1+\widetilde{\mathbf{E}}_2);n) .\tag{C.18}$$

As a result, we can express the overlap as

$$\langle \xi(\widetilde{\mathbf{E}}_1)|\xi(\widetilde{\mathbf{E}}_2)\rangle = \frac{1}{2^n} \sum_{k=n-n_*(\widetilde{\mathbf{E}}_1,\widetilde{\mathbf{E}}_2)}^{n} (-1)^k \mathcal{K}_k(n(\widetilde{\mathbf{E}}_1+\widetilde{\mathbf{E}}_2);n) .\tag{C.19}$$

This relation might be useful since many combinatorial identities involving Krawtchouk polynomials are known [84,85].

**Relation to the Terwilliger algebra.** The Hamming cube $H_n = \{0,1\}^n$ is the set of binary strings of length $n$ with Hamming distance as the metric. The Terwilliger algebra of the Hamming cube [62, 64] is an algebraic structure which is useful in combinatorics and coding theory (see [63] and references therein). We proceed as in [63], and identify the binary strings $a_1 a_2 \cdots a_n$ with their support, the subset $X$ of labels $i$ for which the bit $a_i$ in the string takes value 1. There are $2^n$ possible such subsets, in other words every $X$ is an element of the power set $P(H_n)$ of the Hamming cube. We then define a $P(H_n) \times P(H_n)$ matrix $M_{ij}^t$ whose coefficients are

$$(M_{ij}^t)_{X_1, X_2} = \begin{cases} 1 & \text{if } |X_1| = i, |X_2| = j, |X_1 \cap X_2| = t \\ 0 & \text{otherwise} \end{cases}, \qquad X_1, X_2 \in P(H_n), \qquad (\text{C.20})$$

where we are using $|X|$ to denote the number of elements in $X$ (the number of 1s, the Hamming weight of the binary string). The Terwilliger algebra is defined as the set of matrices of the form

$$\sum_{i,j,t=0}^{n} x_{ij}^t M_{ij}^t, \qquad x_{ij}^t \in \mathbb{C}, \qquad (\text{C.21})$$

which is closed under matrix multiplication. To the state $|\xi(\widetilde{\mathbf{E}})\rangle$, we can associate the element $X \in P(H_n)$ by writing $\widetilde{\mathbf{E}}$ as a binary string and identifying it with its support $X$. Then we have $|X| = n(\widetilde{\mathbf{E}})$. The Gram matrix of the set of vectors $\{|\xi(\widetilde{\mathbf{E}})\rangle\}$ can be represented by an $P(H_n) \times P(H_n)$ matrix $G$ such that

$$G_{X_1 X_2} = \langle \xi(\widetilde{\mathbf{E}}_1) | \xi(\widetilde{\mathbf{E}}_2) \rangle, \qquad (\text{C.22})$$

where $X_1$ and $X_2$ are the elements of $P(H_n)$ associated to $\widetilde{\mathbf{E}}_1$ and $\widetilde{\mathbf{E}}_2$. Let's denote

$$|X_1| = i, \qquad |X_2| = j, \qquad |X_1 \cap X_2| = t. \qquad (\text{C.23})$$

We have

$$n_*(\widetilde{\mathbf{E}}_1, \widetilde{\mathbf{E}}_2) = \max(i, j), \qquad n(\widetilde{\mathbf{E}}_1 + \widetilde{\mathbf{E}}_2) = i + j - 2t, \qquad (\text{C.24})$$

so that the Gram matrix element is

$$G_{X_1 X_2} = \frac{1}{2^n} \sum_{k=n-\max(i,j)}^{n} (-1)^k \mathcal{K}_k(i + j - 2t; n). \qquad (\text{C.25})$$

Because this coefficient depends only on $i$, $j$ and $t$, we can write the Gram matrix as an element of the Terwilliger algebra

$$G = \sum_{i,j,t=0}^{n} x_{ij}^t M_{ij}^t, \qquad x_{ij}^t = \frac{1}{2^n} \sum_{k=n-\max(i,j)}^{n} (-1)^k \mathcal{K}_k(i + j - 2t; n). \qquad (\text{C.26})$$

From this observation, we could attempt to use the techniques of [63] to diagonalize the matrix $G$, and construct the optimal measurement.

# D Overlaps in fermion chains

The purpose of this Appendix is to review the tools used in the computation of overlaps in section 6.2.1. We review Bogoliubov transformations, generalized Wick's theorem and the computation of correlators that contain insertions of Bogoliubov transformations. Then we show how the results lead to the overlaps presented in the main text.

### D.1 Bogoliubov transformations

Let $c = (c_1, \ldots, c_\ell)^\intercal$ and $c^\dagger = (c_1^\dagger, \ldots, c_\ell^\dagger)^\intercal$ and similar definitions of $\psi, \psi^\dagger$. Define the $2\ell$-dimensional vectors

$$\alpha = \begin{pmatrix} c \\ c^\dagger \end{pmatrix}, \quad \Psi = \begin{pmatrix} \psi \\ \psi^\dagger \end{pmatrix} \tag{D.1}$$

whose elements $\alpha_\mu$ are denoted by Greek indices.

We assume that both $\alpha$ and $\Psi$ obey the canonical anticommutation relations:

$$\{\alpha_\mu, \alpha_\nu\} = \Omega_{\mu\nu}, \quad \{\Psi_\mu, \Psi_\nu\} = \Omega_{\mu\nu}, \tag{D.2}$$

where

$$\Omega = \begin{pmatrix} 0 & \mathbf{1}_{\ell \times \ell} \\ \mathbf{1}_{\ell \times \ell} & 0 \end{pmatrix}. \tag{D.3}$$

Consider a linear transformation $W$ between these sets of operators

$$\alpha = W\Psi. \tag{D.4}$$

This transformation is called a Bogoliubov transformation if it preserves the canonical anticommutation relations (D.2) which requires

$$W\Omega W^\intercal = \Omega. \tag{D.5}$$

In addition, since $c^\dagger, \psi^\dagger$ are the Hermitian conjugates of $c, \psi$, we must have (here $(\alpha^\dagger)_\mu = \alpha_\mu^\dagger$ and $*$ is complex conjugation)

$$\alpha^\dagger = W^* \Psi^\dagger. \tag{D.6}$$

Since $\alpha^\dagger = \Omega\alpha$ and $\Psi^\dagger = \Omega\Psi$, we get the condition

$$\Omega W \Omega = W^*. \tag{D.7}$$

The set of Bogoliubov transformations form a group and for real transformations $W^* = W$, it is simply the orthogonal group:

$$W^\intercal W = \Omega W^{-1} \Omega W = \Omega W^{-1} W \Omega = \Omega^2 = \mathbf{1}_{2\ell \times 2\ell} \tag{D.8}$$

with $W^\intercal W = \mathbf{1}_{2\ell \times 2\ell}$ following similarly. Restricting to the component that includes the identity transformation, we get the special orthogonal group.

### D.2 Generalized Wick's theorem as a limit of generalized Gaudin's theorem

Let $\sigma$ be a density operator that satisfies

$$\alpha_\mu \sigma = \sigma \sum_\nu M_{\mu\nu} \alpha_\nu \tag{D.9}$$

for some matrix $M$. Operators of the exponential type (such as reduced density matrices of subregions of spinless fermion chains)

$$\sigma = \frac{1}{Z} \exp\left(\frac{1}{2} \alpha^\intercal S \alpha\right), \quad Z = \mathrm{Tr}\,\sigma, \tag{D.10}$$

belong to this family with $M$ given by [38,39]

$$M = e^{-\Omega S_A}, \tag{D.11}$$

where $S_A$ is the antisymmetric part of $S$. However, not all $\sigma$ that satisfy (D.9) can be written as exponentials (D.10).

Let $\mathcal{T}$ be the operator that implements a real Bogoliubov transformation $T$ on the Hilbert space:

$$\mathcal{T}\alpha\mathcal{T}^{-1} = T\alpha. \tag{D.12}$$

Since $T$ is real, this equation implies that $\mathcal{T}^{-1} = \mathcal{T}^{\dagger}$ is unitary. In addition, we do not assume that $\mathcal{T}$ can be written as an exponential of one-body operators.

The generalized Gaudin's theorem states that [39]

$$\frac{\langle \alpha_{\mu_1} \cdots \alpha_{\mu_n} \mathcal{T} \alpha_{\nu_1} \cdots \alpha_{\nu_n} \rangle_\sigma}{\langle \mathcal{T} \rangle_\sigma} = \sum_{\text{pairings}} (-1)^P \prod_{\text{pairs}} (\text{contraction of a pair}). \tag{D.13}$$

There are three different types of contractions that can appear on the right hand side:

$$G^{(1)}_{\mu\nu} = \frac{\langle \alpha_\mu \alpha_\nu \mathcal{T} \rangle_\sigma}{\langle \mathcal{T} \rangle_\sigma}, \quad G^{(2)}_{\mu\nu} = \frac{\langle \alpha_\mu \mathcal{T} \alpha_\nu \rangle_\sigma}{\langle \mathcal{T} \rangle_\sigma}, \quad G^{(3)}_{\mu\nu} = \frac{\langle \mathcal{T} \alpha_\mu \alpha_\nu \rangle_\sigma}{\langle \mathcal{T} \rangle_\sigma} \tag{D.14}$$

and they are categorized based on the location of the pairs. Equation (D.13) generalizes Gaudin's theorem [67] by including insertions of $\mathcal{T}_i$ in the expectation value.[37]

Generalized Wick's theorem is analogous to equation (D.13), but with the expectation values in the quasi-particle vacuum state $|E_{\text{vac}}\rangle$ which is a pure state. It is obtained as a limit of (D.20) by sending $\sigma$ to $|E_{\text{vac}}\rangle\langle E_{\text{vac}}|$. For this, we take $\sigma$ to be of the exponential type (D.10) with (this would correspond to a free fermion Hamiltonian)

$$S = \begin{pmatrix} 0 & s \\ -s & 0 \end{pmatrix}, \tag{D.15}$$

where $s = \text{diag}(s_i)$ and $S$ is antisymmetric so that

$$M = e^{-\Omega S} = \begin{pmatrix} e^s & 0 \\ 0 & e^{-s} \end{pmatrix}. \tag{D.16}$$

The exact form of $S$ is not important and we have chosen it in such a way that the $\{s_i\} \to \infty$ gives the quasi-particle vacuum state. To see this, write

$$\sigma = \frac{1}{Z} \exp\left(-\sum_i s_i c_i^{\dagger} c_i\right), \quad Z = \prod_i (1 + e^{-s_i}). \tag{D.17}$$

It has eigenstates $|E_{i_1 \dots i_n}\rangle$ and eigenvalues $(1/Z)\exp[-(s_{i_1} + \dots + s_{i_n})]$ generated by acting on the quasi-particle vacuum $|E_{\text{vac}}\rangle$ with creation operators. Hence it is

$$\sigma = \frac{1}{Z}\left(|E_{\text{vac}}\rangle\langle E_{\text{vac}}| + \sum_i e^{-s_i}|E_i\rangle\langle E_i| + \dots\right) \tag{D.18}$$

and the limit $\{s_i\} \to \infty$ produces a pure state

$$\lim_{\{s_i\}\to\infty} \sigma = |E_{\text{vac}}\rangle\langle E_{\text{vac}}|. \tag{D.19}$$

The generalized Wick's theorem is then

$$\frac{\langle E_{\text{vac}}|\alpha_{\mu_1} \cdots \alpha_{\mu_n} \mathcal{T} \alpha_{\nu_1} \cdots \alpha_{\nu_n}|E_{\text{vac}}\rangle}{\langle E_{\text{vac}}|\mathcal{T}|E_{\text{vac}}\rangle} = \sum_{\text{pairings}} (-1)^P \prod_{\text{pairs}} (\text{contraction of a pair}) \tag{D.20}$$

and the three types of contractions appearing on the right hand side are the $\lim_{\{s_i\}\to\infty} G^{(1,2,3)}_{\mu\nu}$. We will next compute the contractions.

---

[37]Gaudin's theorem is a generalization of Wick's theorem to expectation values in mixed states. Its proof is based on the cyclicity of the trace and the identity (D.9).

### D.3 Computation of contractions

We start with the simple 2-point function $\langle \alpha_\mu \alpha_\nu \rangle_\sigma = \text{Tr}(\sigma \alpha_\mu \alpha_\nu)$ in a mixed state $\sigma$ that obeys the relation (D.9). Using the canonical anticommutation relations and (D.9), we can write

$$\langle \alpha_\mu \alpha_\nu \rangle_\sigma = \Omega_{\mu\nu} \text{Tr}\,\sigma - \langle \alpha_\nu \alpha_\mu \rangle_\sigma = \Omega_{\mu\nu} \text{Tr}\,\sigma - \sum_\lambda M_{\mu\lambda} \langle \alpha_\lambda \alpha_\nu \rangle_\sigma. \tag{D.21}$$

From this the 2-point function is solved

$$\langle \alpha_\mu \alpha_\nu \rangle_\sigma = \text{Tr}\,\sigma \, [(1+M)^{-1}\Omega]_{\mu\nu}. \tag{D.22}$$

Let $\mathcal{T}_{i=1,2,3}$ be operators that implement three different Bogoliubov transformations $T_i$:

$$\mathcal{T}_i \, \alpha \, \mathcal{T}_i^{-1} = T_i \, \alpha. \tag{D.23}$$

Thus the operators $\mathcal{T}_i$ obey the relation (D.9) with $M = T_i^{-1}$. We consider real Bogoliubov transformations that are orthogonal $T_i^{\mathsf{T}} = T_i^{-1}$ and for which $\mathcal{T}^\dagger = \mathcal{T}^{-1}$ is unitary.

Consider the expectation value

$$\langle \mathcal{T}_1^{-1} \alpha_\mu \mathcal{T}_3 \, \alpha_\nu \mathcal{T}_2 \rangle_\sigma = \langle \alpha_\mu \mathcal{T}_3 \, \alpha_\nu \rangle_{\mathcal{T}_2 \sigma \mathcal{T}_1^{-1}}, \tag{D.24}$$

where we used cyclicity of the trace. Using

$$\mathcal{T}_3 \, \alpha_\nu = \Big( \sum_\lambda (T_3)_{\nu\lambda} \alpha_\lambda \Big) \mathcal{T}_3, \tag{D.25}$$

we get

$$\langle \mathcal{T}_1^{-1} \alpha_\mu \mathcal{T}_3 \, \alpha_\nu \mathcal{T}_2 \rangle_\sigma = \sum_\lambda (T_3)_{\nu\lambda} \langle \alpha_\mu \alpha_\lambda \rangle_{\widehat{\sigma}}, \tag{D.26}$$

where we have defined $\widehat{\sigma} \equiv \mathcal{T}_3 \mathcal{T}_2 \sigma \, \mathcal{T}_1^{-1}$ which obeys the relation

$$\alpha_\mu \widehat{\sigma} = \widehat{\sigma} \sum_\nu (T_3^{-1} T_2^{-1} M T_1)_{\mu\nu} \, \alpha_\nu, \tag{D.27}$$

so that

$$\langle \alpha_\mu \alpha_\lambda \rangle_{\widehat{\sigma}} = \text{Tr}\,\widehat{\sigma} \, [(\mathbf{1} + T_3^{-1} T_2^{-1} M T_1)^{-1}\Omega]_{\mu\nu}. \tag{D.28}$$

Noting that

$$\text{Tr}\,\widehat{\sigma} = \langle \mathcal{T}_1^{-1} \mathcal{T}_3 \mathcal{T}_2 \rangle_\sigma, \tag{D.29}$$

we get

$$\frac{\langle \mathcal{T}_1^{-1} \alpha_\mu \mathcal{T}_3 \, \alpha_\nu \mathcal{T}_2 \rangle_\sigma}{\langle \mathcal{T}_1^{-1} \mathcal{T}_3 \mathcal{T}_2 \rangle_\sigma} = [(1 + T_3^{-1} T_2^{-1} M T_1)^{-1} \Omega T_3^{\mathsf{T}}]_{\mu\nu}. \tag{D.30}$$

The quasi-particle vacuum expectation values are obtained by focusing on exponential $\sigma$ with $M = e^{-\Omega S}$ and taking the limit $\{s_i\} \to \infty$:

$$\frac{\langle E_{\text{vac}}| \mathcal{T}_1^{-1} \alpha_\mu \mathcal{T}_3 \, \alpha_\nu \mathcal{T}_2 | E_{\text{vac}} \rangle}{\langle E_{\text{vac}}| \mathcal{T}_1^{-1} \mathcal{T}_3 \mathcal{T}_2 | E_{\text{vac}} \rangle} = \lim_{\{s_i\} \to \infty} \frac{\langle \mathcal{T}_1^{-1} \alpha_\mu \mathcal{T}_3 \, \alpha_\nu \mathcal{T}_2 \rangle_\sigma}{\langle \mathcal{T}_1^{-1} \mathcal{T}_3 \mathcal{T}_2 \rangle_\sigma}. \tag{D.31}$$

We focus our attention to the following 2-point functions that appear in the computation of the overlaps:

$$\frac{\langle E_{\text{vac}}| \mathcal{T} \alpha_\mu \alpha_\nu | E_{\text{vac}} \rangle}{\langle E_{\text{vac}}| \mathcal{T} | E_{\text{vac}} \rangle} = \lim_{\{s_i\} \to \infty} [(1 + M T^{-1})^{-1}\Omega]_{\mu\nu} \tag{D.32}$$

$$\frac{\langle E_{\text{vac}}| \alpha_\mu \mathcal{T} \alpha_\nu | E_{\text{vac}} \rangle}{\langle E_{\text{vac}}| \mathcal{T} | E_{\text{vac}} \rangle} = \lim_{\{s_i\} \to \infty} [(1 + T^{-1} M)^{-1}\Omega T^{-1}]_{\mu\nu} \tag{D.33}$$

$$\frac{\langle E_{\text{vac}}| \alpha_\mu \alpha_\nu \mathcal{T} | E_{\text{vac}} \rangle}{\langle E_{\text{vac}}| \mathcal{T} | E_{\text{vac}} \rangle} = \lim_{\{s_i\} \to \infty} [(1 + T^{-1} M)^{-1}\Omega]_{\mu\nu}. \tag{D.34}$$

The other limits were not given in [39], but we can compute them using the identity

$$\lim_{\{s_i\}\to\infty} [(1+Q^{-1}MP)^{-1}\Omega]_{\mu\nu} = \begin{pmatrix} P_{22}^{\mathsf{T}} & P_{12}^{\mathsf{T}} \\ P_{21}^{\mathsf{T}} & P_{11}^{\mathsf{T}} \end{pmatrix} \begin{pmatrix} (Q_{11}P_{22}^{\mathsf{T}}+Q_{12}P_{21}^{\mathsf{T}})^{-1} & 0 \\ 0 & 0 \end{pmatrix} \begin{pmatrix} Q_{12} & Q_{11} \\ Q_{22} & Q_{21} \end{pmatrix}. \quad (D.35)$$

The results are

$$\frac{\langle E_{\text{vac}}|\mathcal{T}\alpha_\mu\alpha_\nu|E_{\text{vac}}\rangle}{\langle E_{\text{vac}}|\mathcal{T}|E_{\text{vac}}\rangle} = \begin{pmatrix} 0 & 1 \\ 0 & T_{21}T_{11}^{-1} \end{pmatrix} \quad (D.36)$$

$$\frac{\langle E_{\text{vac}}|\alpha_\mu\mathcal{T}\alpha_\nu|E_{\text{vac}}\rangle}{\langle E_{\text{vac}}|\mathcal{T}|E_{\text{vac}}\rangle} = \begin{pmatrix} 0 & T_{11}^{-1} \\ 0 & 0 \end{pmatrix} \quad (D.37)$$

$$\frac{\langle E_{\text{vac}}|\alpha_\mu\alpha_\nu\mathcal{T}|E_{\text{vac}}\rangle}{\langle E_{\text{vac}}|\mathcal{T}|E_{\text{vac}}\rangle} = \begin{pmatrix} T_{11}^{-1}T_{12} & 1 \\ 0 & 0 \end{pmatrix}. \quad (D.38)$$

The normalization factor is computed in [38,39]:

$$\langle E_{\text{vac}}|\mathcal{T}|E_{\text{vac}}\rangle = \lim_{\{s_i\}\to\infty} \langle\mathcal{T}\rangle_\sigma = (\det T_{22})^{1/2}. \quad (D.39)$$

### D.4 Overlaps of eigenstates

Overlaps of eigenstates of two modular Hamiltonians are

$$\langle E_{i_1\ldots i_n}|\widetilde{E}_{j_1\ldots j_m}\rangle = \langle E_{i_1\ldots i_n}|\mathcal{T}|E_{j_1\ldots j_m}\rangle = \langle E_{\text{vac}}|c_{i_n}\cdots c_{i_1}\mathcal{T}c_{j_1}^\dagger\cdots c_{j_m}^\dagger|E_{\text{vac}}\rangle. \quad (D.40)$$

Generalized Wick's theorem states that

$$\frac{\langle E_{i_1\ldots i_n}|\widetilde{E}_{j_1\ldots j_m}\rangle}{\langle E_{\text{vac}}|\widetilde{E}_{\text{vac}}\rangle} = \frac{\langle E_{\text{vac}}|c_{i_n}\cdots c_{i_1}\mathcal{T}c_{j_1}^\dagger\cdots c_{j_m}^\dagger|E_{\text{vac}}\rangle}{\langle E_{\text{vac}}|\mathcal{T}|E_{\text{vac}}\rangle} \quad (D.41)$$

expands to a sum over products of contractions. The contractions are obtained from the general formulae above:

$$\frac{\langle E_{\text{vac}}|\mathcal{T}c_i^\dagger c_j^\dagger|E_{\text{vac}}\rangle}{\langle E_{\text{vac}}|\mathcal{T}|E_{\text{vac}}\rangle} = (T_{21}T_{11}^{-1})_{ij} \quad (D.42)$$

$$\frac{\langle E_{\text{vac}}|c_i\mathcal{T}c_j^\dagger|E_{\text{vac}}\rangle}{\langle E_{\text{vac}}|\mathcal{T}|E_{\text{vac}}\rangle} = (T_{11}^{-1})_{ij} \quad (D.43)$$

$$\frac{\langle E_{\text{vac}}|c_ic_j\mathcal{T}|E_{\text{vac}}\rangle}{\langle E_{\text{vac}}|\mathcal{T}|E_{\text{vac}}\rangle} = (T_{11}^{-1}T_{12})_{ij} \quad (D.44)$$

with the normalization given in (D.39). This leads to the formula (6.58) presented in the main text.

## E Optimal measurement for the free boson

In this appendix, we consider the free boson CFT and attempt to describe the optimal subsystem measurement that distinguishes between two thermal states, using the setup of section 7.2.

Let $\phi(z)$ be a free boson and define $j(z) = \partial\phi(z)$. We have the modes

$$\alpha_n = \frac{1}{2\pi i}\oint_0 du\, u^n j(u) = \frac{1}{2\pi i}\int_{C_+} du\, u^n j(u) - \frac{1}{2\pi i}\int_{C_+} d\bar{u}\, \bar{u}^n \bar{j}(\bar{u}). \quad (E.1)$$

We obtain

$$\alpha_n = \frac{1}{2\pi i} \int_I dz\, i^n e^{in\pi f(z)/W} j(z) + \text{h.c.} \tag{E.2}$$

In this case, the commutation relations are

$$[j(z), j(z')] = \partial_z [\phi(z), \partial \phi(z')] = -2\pi i\, \partial_z \delta(z - z') \,. \tag{E.3}$$

Using the above formula, we can check that $[\alpha_n, \alpha_m] = n\delta_{m+n}$ as expected. We now consider the state $\rho$ with

$$\widetilde{\alpha}_n = \frac{1}{2\pi i} \int_I du\, i^n e^{in\pi \widetilde{f}(z)/\widetilde{W}} j(z) + \text{h.c.} \,. \tag{E.4}$$

To obtain the overlaps between the eigenstates of $\rho$ and that of $\sigma$, we need to compute the commutator $[\alpha_n, \widetilde{\alpha}_{-m}]$. After some manipulations, we find

$$[\alpha_n, \widetilde{\alpha}_m] = \frac{i^{n+m} n}{2W} \int_I dz\, f'(z) e^{i\pi(nf(z)/W + m\widetilde{f}(z)/\widetilde{W})} + \text{h.c.} \equiv A_{nm} \,, \tag{E.5}$$

which appear difficult to compute explicitly. A basis of normalized eigenstates for $K$ is labeled by $\mathbf{k} = (k_1, k_2, \ldots)$ with

$$|\Delta_{\mathbf{k}}\rangle = \frac{1}{\sqrt{N_{\mathbf{k}}}} \alpha_{-1}^{k_1} \alpha_{-2}^{k_2} \ldots |0\rangle \,, \tag{E.6}$$

where the normalization is $N_{\mathbf{k}} = \prod_{i \geq 1} i^{k_i} k_i!$ and we have

$$L_0 |\Delta_{\mathbf{k}}\rangle = \Delta_{\mathbf{k}} |\mathbf{k}\rangle, \qquad \Delta_{\mathbf{k}} = \sum_{i \geq 1} i k_i \,. \tag{E.7}$$

Similarly, for $\widetilde{K}$, we have $\widetilde{\mathbf{k}} = (\tilde{k}_1, \tilde{k}_2, \ldots)$ and

$$|\widetilde{\Delta}_{\widetilde{\mathbf{k}}}\rangle = \frac{1}{\sqrt{N_{\widetilde{\mathbf{k}}}}} \widetilde{\alpha}_{-1}^{\tilde{k}_1} \widetilde{\alpha}_{-2}^{\tilde{k}_2} \ldots |0\rangle \,. \tag{E.8}$$

The overlap $\langle \Delta_{\mathbf{k}} | \widetilde{\Delta}_{\widetilde{\mathbf{k}}} \rangle$ is non-zero only if $N = \sum_i k_i = \sum_i \tilde{k}_i$. Is is given as

$$\langle \Delta_{\mathbf{k}} | \widetilde{\Delta}_{\widetilde{\mathbf{k}}} \rangle = \text{perm}(M_{\mathbf{k}\widetilde{\mathbf{k}}}) \,, \tag{E.9}$$

where $M_{\mathbf{k}\widetilde{\mathbf{k}}}$ is the $N \times N$ matrix constructed by starting with the matrix $A_{ij}$ and replacing each entry $(i,j)$ by a $k_i \times \tilde{k}_j$ block where all the elements are equal to $A_{ij}$. Here, perm denotes the permanent which is similar to the determinant, but with only plus signs in the sum over permutations.

We will now attempt to describe the optimal measurement for the free boson, where we have two global thermal states as described in section 7.2. To compute the overlaps, it is convenient to change variable to $w = f(z)$ so that

$$A_{nm} = \frac{i^{n-m} n}{2W} \int_{-W/2}^{W/2} dw\, e^{i\pi(nw/W - mF(w)/\widetilde{W})} + \text{h.c.} \,, \tag{E.10}$$

where $F(w) = \widetilde{f}(f^{-1}(w))$. Unfortunately, this quantity is hard to compute analytically. It can be probed in the small $L$ expansion. At first order, we get

$$A_{nm} = \begin{cases} n & m = n \\ \dfrac{\pi L}{\log(\frac{L}{\varepsilon})}(T_2 - T_1)\dfrac{mn}{m-n} + O(L^2) & m+n \text{ is odd} \\ O(L^2) & \text{otherwise} \end{cases} \,. \tag{E.11}$$

As a result, we see that $|\Delta_{\mathbf{k}}\rangle$ and $|\widetilde{\Delta}_{\widetilde{\mathbf{k}}}\rangle$ can have a non-zero overlap at first order only if they differ in less than one place. We can write

$$\mathbf{k} = \mathbf{k}_0 + \delta_a, \qquad \widetilde{\mathbf{k}} = \mathbf{k}_0 + \delta_b\,, \qquad a + b \text{ odd}, \quad a \neq b\,, \tag{E.12}$$

where $\delta_i$ means a one in position $i$. We compute

$$\langle \Delta_{\mathbf{k}} | \widetilde{\Delta}_{\widetilde{\mathbf{k}}} \rangle = \frac{N_{\mathbf{k}_0} A_{ab} k_a \tilde{k}_b}{\sqrt{N_{\mathbf{k}} N_{\widetilde{\mathbf{k}}}}} + O(L^2)\,. \tag{E.13}$$

We have $N_{\mathbf{k}} N_{\widetilde{\mathbf{k}}} = N_{\mathbf{k}_0}^2\, a k_a\, b \tilde{k}_b$ so we get for $a + b$ odd

$$\langle \Delta_{\mathbf{k}} | \widetilde{\Delta}_{\widetilde{\mathbf{k}}} \rangle = \frac{\pi L}{\log(\frac{L}{\varepsilon})} \frac{\sqrt{a\, b\, k_a \tilde{k}_b}}{b - a} (T_2 - T_1) + O(L^2)\,. \tag{E.14}$$

Following section 7.1.2, we can also define perturbatively the states $|\xi(\widetilde{\Delta}_{\mathbf{k}})\rangle$ which span the acceptance subspace $\mathcal{H}_Q$. Although it's possible to write explicit perturbative expressions, this is not enough. Indeed, to understand this subspace and define the measurement, we would need them to do a Gram-Schmidt procedure to orthonormalize these vectors. To do this, we will have to go beyond the perturbation theory in $L$ and we don't expect to be able to obtain analytical results using this approach. In conclusion, the optimal measurement seems to be difficult to describe explicitly, even in simple examples. An alternative is to use the likelihood ratio test following section 7.1.3, which will be more tractable to implement here, because it requires only the knowledge of the overlaps.

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
