# Peer review of "Quantum hypothesis testing in many-body systems"

_SciPost Physics Core, doi:SciPost Phys. Core 4, 019 (2021)_

## Round 1 · Referee Report · Anonymous (Referee 1) · 2021-5-27

Strengths

--

Weaknesses

--

Report

The authors have reasonably addressed my comments and concerns.

---

## Round 1 · Author Response

Dear editor and referee,

Thank you for the detailed and helpful review of the manuscript. We have reviewed all of the referee comments and made changes accordingly. We have also attached a short note "CapacityDegeneracy.pdf" on a technical point raised by the referee. In addition, we are familiar with the interesting work on variance of suprisal and have added it as a reference to the manuscript.

As suggested by the editor, we have decided to resubmit the revised manuscript to SciPost Physics Core instead of the original journal SciPost Physics.

With best regards,
The authors

---

## Round 1 · List of Changes

Changes according to referee remarks:
* * *
--- "The authors study parametrically close many-body states. However, there is a subtle issue of (non-)commuting limits here, namely taking the limit λ->0 and taking the thermodynamic limit. For example, even if two many-body systems have arbitrarily close temperatures (independent of system size) then in the thermodynamic limit their global states will generically be perfectly distinguishable in the single shot by a global energy measurement (their trace-distance will approach 1). I think it would be important to discuss this issue in this paper."

Author response: This problem can be avoided by understanding that the limit λ->0 is taken while keeping N fixed, but large. Then the trace distance between the two global states is less than one (but close to one for large N) and taking the limit λ->0 makes it approach zero. To clarify this point, we have added a footnote at the beginning of section 6.1 where the thermodynamic limit is first mentioned:

"In what follows, there is a possibility of an order of limits issue with the thermodynamic L-> ∞ and the perturbative λ->0 limits. To circumvent the issue, we simply take L to be larger than any scale in the problem and take the perturbative limit λ->0 while keeping L fixed. We thank the referee for pointing out this subtlety."

In general, the definition of quantum information quantities in the thermodynamic/continuum limit is difficult and a detailed analysis is beyond the scope of this paper.
* * *
--- "There are two different kind of expansions in this paper: One with respect to the parameter value, one with respect to the number of copies one takes in the hypothesis test. The expressions "first order" or "second order" are used differently in the two settings. It may therefore be useful to somehow distinguish these (for example by saying "leading order in n" vs. "first order in λ" or so)."

Changes: In all places with "first order" or "second order", we indicated whether the parameter being expanded is n or lambda. For example "first order" -> "first order in n". Also added extra "λ->0" to correct places. The changes are located at:

Section 3.1 at multiple places
Section 3.2.3
Section 4.3 (second order result -> second order correction)
Section 4.4.1
Section 5.2, numerical results subsection
* * *
--- "In secs. 3.2.1--3.2.3 the authors demonstrate that their, rigorously proven, theorem holds in particular simple examples. It's not clear to me what the use of these sections is, since the examples don't seem to deliver additional insight."

Author comment: We think that explicit examples are always valuable especially in this paper whose aim to be more pedagogical. In addition, the examples provide explicit formulae for relative entropy and its variance in simple cases. However, we have made the following changes to add more motivation around sections 3.2.1-3.2.3. In addition, we have improved the qubit example in section 3.2.1.

New paragraph right before section 3.2.1: "An interesting question is whether there exists special classes of density matrices for which there
is also a constant upper bound for the ratio (3.19). Such an upper bound would imply an upper bound for the perturbative variance by perturbative relative entropy. To gain more intuition, it is useful to study the lower bound (3.20) in explicit examples. At least in the simple examples studied next, no upper bound appears.\footnote{An additional example is presented in section 6.1.3, where relative entropy and its variance between reduced density matrices of a spinless fermion chain are computed.}"

New paragraph at the end of section 3.2.3: "Interestingly, non-perturbative relative entropy variance between two thermal states turns
out to be proportional to the capacity of entanglement (3.54). This might have implications for thermodynamics of AdS black holes in the AdS/CFT correspondence where the holographic dual of the capacity of entanglement is known [49, 50]. However, the holographic dual of relative entropy variance is not yet known, but further results in this direction will be reported in upcoming work [51]."

Added the extra reference [51].

Improvements to section 3.2.1:

-Added eigenvalues of rho (p_+-) and the allowed range for the parameter lambda.

-At the end of section 3.2.1, we explained in detail why the variance vanishes at lambda = +-1 when a = 0. This provides extra use for this qubit example.
* * *
--- "As someone not very well versed in CFT techniques, I found the discussion of the CFT computations extremely dense and hard to follow (especially in contrast to the very explicit, detailed and pedagogical calculations in the rest of the paper). It would be very useful to be more explicit in the calculations, in particular if the aim of the paper is (as it seems to be) to bring together different communities."

We have added a few more details and explanations in the CFT section.

%%%%%%%%%%%%%

Changes according to small referee pointers/comments:

%%%%%%%%%%%%%
* * *
--- "In the introduction it says that subsystem reduced density matrices of fermion chains are determined by two-point functions. This is only true for Gaussian states (free fermions) as studied later in the paper and should be specified."

Page 3, original sentence: "For example subsystems of fermion chains have been extensively studied in the context of entanglement, because subsystem reduced density matrices are determined analytically by two-point
functions [26–29]."

Change: fermion chains -> free fermion chains
* * *
--- "After eq. (2.6) it says that "[...] -log Q_s(ρ,σ) are the relative Rényi entropies [...]".
The quantity -log Q_s(ρ,σ) is only proportional to the relative Rényi entropy of order s?!"

Page 5, below (2.6) old sentence: "We can see that − log Qs(ρ, σ) are the relative Rényi entropies defined by Petz [43]."

Small correction: "... are proportional to the relative Rényi entropies ..."
* * *
--- "At the bottom of page 5 it says that Q satisfies the data-processing inequality. Given the pedagogical aim of the paper, it may be useful to state what the data-processing inequality is."

Page 5, original sentence: "$Q$ also satisfies the data-processing inequality."

Sentence changed to: "$Q$ also satisfies the data-processing inequality (B.17)."

Explanation: The data processing inequality is described in equation (B.17).
* * *
--- "Footnote 9 on p. 12: ΔK is not yet defined."

Old footnote, page 12: "This follows directly from the definition since $ \langle \Delta K \rangle_\rho^2 = S(\rho \lVert \sigma)^2 = \mathcal{O}(\lambda^4) $ and $ \Delta K^2 = \lambda^2 \mathcal{L}^2 + \mathcal{O}(\lambda^3) $."

Addition: ... where $ \Delta K = \log{\rho} - \log{\sigma} $.
* * *
--- "What the authors call "capacity of entanglement" is known under various names in information theory, such as variance of surprisal, varentropy etc."

Old footnote 10 (now it is footnote 12): "... which for a reduced density matrix is known as the capacity of entanglement"

Addition to the footnote: "... which for a reduced density matrix is known as the capacity of entanglement (other names include for example variance of surprisal and varentropy)"
* * *
--- "After (3.49) the authors say that if "β_2 -> ∞, ρ_2 reduces to the ground state, and the relative entropy variance vanishes (along with C(β_2)->0)." This is only true if the ground state of the Hamiltonian is non-degenerate."

The capacity of entanglement of a thermal state does vanish in the zero temperature limit even if the ground state is degenerate, because the density matrix reduces to a flat state. We have attached a short note "CapacityDegeneracy.pdf" that proves this claim explicitly.

We also added footnote 14 in the manuscript with the following sentence: "In a system with a degenerate ground state, at zero temperature the density matrix reduces to a flat state (all non-zero eigenvalues are equal), for which the capacity of entanglement is zero [42, 50]."
* * *
-- "In sec 4.2 shouldn't the acceptance condition be | |E| - |\tilde E| |>= \mathcal E? I.e. with absolute value sign? Similarly for (4.23) etc.?"

No there is no absolute value sign.
* * *
--- "After (4.33) it should probably read "completely positive" instead of "positive"."

Original sentence below (4.33): "since the map ρ → ρ_D is positive and trace preserving [54]."

Change: positive -> (completely) positive

Explanation: The map is completely positive, but only positivity is required for monotonicity as proven in 1512.06117.
* * *
-- "In (6.28) and alter ΔA is not defined."

Below (6.28) added the text: where \Delta A_{ij} = A_{ij} - \widetilde{A}_{ij}
* * *
-- "Why is the sandwiched Rényi relative entropy suddenly mentioned after (6.33)?"

Original paragraph below (6.33): "Sandwiched Rényi relative entropy between RDMs of a free fermion chain was computed in [35] (see also [36]). One can check that the relative entropy (6.26) matches with the first derivative of their expression. As a consistency check, we will see below that the expression for the variance obeys the lower bound (3.20)."

New paragraph: "As far as the authors are aware, the expressions (6.26) and (6.33) for relative entropy and its variance have not appeared in the literature before. However, sandwiched Rényi relative entropy between RDMs of a free fermion chain was computed in [35] (see also [36]) and one can check that the relative entropy (6.26) matches with the first derivative of their expression. Unfortunately, we did not manage to compute the second derivative to see whether the result matches with the variance. As an independent consistency check of (6.33), we will see below that it obeys the lower bound (3.20)."
* * *
--- "In sec. 6.3.3. it says that the XY spin chain can be mapped to a free fermion chain in the thermodynamic limit. This is also possible for finite chains (care must be taken with regard to the boundary condition) and indeed done in (6.85)."

We decided to clarify some details here. List of changes in section 6.3.3:

-(6.84): Added sum range from 1 to L

-(6.85): Changed sum range to be from 1 to L-1 (previous was incorrect)

-Sentence: "In the thermodynamic limit, this Hamiltonian can be mapped to a free fermion chain [61]"

Change: free fermion chain -> periodic free fermion chain

-New footnote above (6.85): "Strictly speaking, the Jordan–Wigner transformation also produces an additional interaction term between ψ_1 and ψ_L in the periodic fermion chain (6.85). However, the interaction produces contributions to Λ_k and v_k that are subleading in the thermodynamic limit L → ∞ [61]. Hence we neglect these extra contributions and focus on the periodic fermion chain (6.85) with translation symmetry."

Original Sentence above (6.87): "Due to translation invariance, the thermal two-point function is a function of i − j"

New sentence above (6.87): "Due to translation invariance, the thermal two-point function is a function of i − j only, and in the thermodynamic limit L → ∞, it takes the form"
* * *
--- "I didn't understand what is meant with the last sentence before Sec. 7: "It is interesting that the breaking of translation invariance leads to non-commutativity from the perspective of two fermions""

Original sentence below (6.89): "We see that v is independent of the temperature
β and of the distance r. This is true in any translation invariant fermion chain for which the
thermal two-point function is of the form (6.88)."

New refined sentence: "We see that v is independent of the temperature
β and of the distance r which is true in any translation invariant fermion chain with
thermal two-point function of the form (6.88)."

Original last sentence of 6.3.3: "It is interesting that the breaking of translation invariance leads to non-commutativity
from the perspective of two fermions."

Replacement: "If the fermion chain is not translation invariant this is no longer true, because then the modular Hamiltonians K, tilde K do not generally commute. It is interesting that translation invariance implies commutativity of two-fermion density matrices in global thermal states."
* * *
--- "In (7.3), L_0 and c hasn't been defined."

This has been corrected.
* * *
--- "In (7.9,7.10) are referred to as giving the "entanglement spectrum", but they give states
(the eigenstates not the eigenvalues of the modular Hamiltonians)."

This has been corrected.
* * *
--- "While maybe not directly relevant for the questions studied here (and hence does not need to be cited here), the authors may be interested in the recent preprint arxiv:2009.08391, which discusses mathematical properties of the relative entropy variance from an information theoretic point of view."

Added a sentence with a reference to arxiv:2009.08391 at the end of the introduction: "After the completion of this paper, a related paper also studying relative entropy variance from an information theoretic point of view appeared in [42]."
* * *
Other minor changes and fixes of typos:

- Fixed equation formatting below (B.20) and in (B.21).

---

## Editorial Decision

published